# A Near-Optimal Algorithm for Decentralized Convex-Concave Finite-Sum Minimax Optimization

**Hongxu Chen**[1]     **Ke Wei**[1]     **Haishan Ye**[2,3]     **Luo Luo**[1,4*]

[1]School of Data Science, Fudan University
[2]School of Management, Xi'an Jiaotong University
[3]SGIT AI Lab, State Grid Corporation of China
[4]Shanghai Key Laboratory for Contemporary Applied Mathematics
{hxchen20,kewei}@fudan.edu.cn  yehaishan@xjtu.edu.cn  luoluo@fudan.edu.cn

## Abstract

In this paper, we study the distributed convex-concave finite-sum minimax optimization over the network, and a decentralized variance-reduced optimistic gradient method with stochastic mini-batch sizes (DIVERSE) is proposed. For the strongly-convex-strongly-concave objective, it is shown that DIVERSE can achieve a linear convergence rate that depends on the global smoothness parameters, yielding sharper computation and communication complexity bounds than existing results. Furthermore, we also establish the lower complexity bounds, which show that our upper bounds are optimal up to a logarithmic factor in terms of the local incremental first-order oracle calls, the computation rounds, and the communication rounds. Numerical experiments demonstrate that our algorithm outperforms existing methods in practice.

## 1 Introduction

In this paper, we consider the following distributed minimax optimization problem

$$\min_{\mathbf{x}\in\mathbb{R}^{d_x}} \max_{\mathbf{y}\in\mathbb{R}^{d_y}} f(\mathbf{x},\mathbf{y}) := \frac{1}{m}\sum_{i=1}^{m} f_i(\mathbf{x},\mathbf{y}), \tag{1}$$

where the global objective $f : \mathbb{R}^{d_x} \times \mathbb{R}^{d_y} \to \mathbb{R}$ is $\mu$-strongly-convex-$\mu$-strongly-concave. We assume the local function $f_i : \mathbb{R}^{d_x} \times \mathbb{R}^{d_y} \to \mathbb{R}$ on the $i$-th node has the finite-sum structure of the form

$$f_i(\mathbf{x},\mathbf{y}) := \frac{1}{n}\sum_{j=1}^{n} f_{i,j}(\mathbf{x},\mathbf{y}), \tag{2}$$

where each $f_{i,j}$ is smooth. This formulation appears in many fields, including game theory [9, 21], robust optimization [10, 29], and control theory [45]. In particular, it has received increasing attention recently from the machine learning community, with the rise of adversarial generative networks [6, 25], adversarial training [7, 49, 68, 71, 77], and reinforcement learning [18, 75].

The first-order minimax optimization has been studied extensively over the past decades. Gradient descent ascent (GDA) is a natural extension of gradient descent in minimization problem, which is a cornerstone for many minimax optimization algorithms [11, 51]. In the extragradient (EG) method, [24, 34, 73] an intermediate prediction step is introduced to improve the convergence of GDA, which exhibits the optimal convergence rate under the convex-concave assumption [59, 89].

---

*Corresponding author

39th Conference on Neural Information Processing Systems (NeurIPS 2025).

Additionally, the optimistic gradient descent ascent (OGDA) method [61, 64] can also achieve the optimal convergence rate, by incorporating the momentum-like term. In the more general variational inequality framework, Kotsalis et al. [35] proposed an optimal operator extrapolation method.

For large-scale optimization arising from machine learning, it is desirable to design efficient stochastic algorithms by exploiting the finite-sum structure in the objective since computing the full gradient is usually expensive. The variance reduction ideas used to achieve the optimal incremental first-order oracle (IFO) complexity bounds in the minimization problem [1, 4, 20, 32, 37, 67, 78, 90] have been extended to minimax optimization, though the details are quite involved. Palaniappan and Bach [60] incorporated variance reduction into GDA iteration and provided an catalyst acceleration framework. Chavdarova et al. [16] and Alacaoglu et al. [3] studied the EG method with variance reduction for a specific minimax problem. Alacaoglu and Malitsky [2] introduced a retracted term into the iteration of EG and OGDA, achieving the optimal IFO complexity for the finite-sum minimax problem under the convex-concave setting [26].

The decentralized optimization have been widely studied in recent years. Compared with the centralized scenario, it can avoid the communication and computation bottlenecks for problems over networks [57, 82, 87], while the algorithm design and analysis are more challenging since each node in a network can only directly share the information with its neighbors. For the minimax optimization, Mukherjee and Chakraborty [55], Beznosikov et al. [13], and Luo and Ye [46] developed EG methods for the decentralized setting and provided the linear convergence rates. Rogozin et al. [65] extended the results to the non-Euclidean mirror prox framework. Kovalev et al. [39] combined variance reduction and the optimistic gradient method [61] within the ADOM framework [38], achieving the best-known upper complexity bound on the computation rounds and the communication rounds.

It is worth noting that exiting decentralized minimax optimization methods require identical mini-batch size for all the nodes when constructing the local gradient estimator, which is sample inefficient. Moreover, both the computation complexity and communication complexity of previous works depend on the local smoothness parameters. It remains an open question on how to develop the decentralized minimax optimization algorithm that depends on the global smoothness parameters.

In this paper, we propose a decentralized variance-reduced optimistic gradient method with stochastic mini-batch sizes (DIVERSE) for the minimax problem (1), which can find an $\epsilon$-suboptimal solution with $\mathcal{O}((mn + \min\{mnL, \sqrt{mn}\bar{L}\}/\mu) \log(1/\epsilon))$ local incremental first-order oracle (LIFO) calls, $\tilde{\mathcal{O}}((n + L/\mu + \min\{nL, \sqrt{n/m}\bar{L}\}/\mu) \log(1/\epsilon)$ computation rounds, and $\tilde{\mathcal{O}}(\sqrt{\chi}L/\mu \log(1/\epsilon))$ communication rounds. Here, $L$ is the smoothness parameter of the objective $f$, $\bar{L}$ is the mean-squared smoothness parameter of the function set $\{f_{i,j}\}_{i,j=1}^{m,n}$, and $\chi$ is the characteristic number of the mixing matrix associated with the network. The corresponding lower bounds have also been established which demonstrate that all the above results are (nearly) optimal. We would like to emphasize that all of our complexity bounds have the global smoothness dependency, which are tighter than existing results that only rely on the local smoothness [39, 46, 55]. Moreover, the linear convergence guarantee in this paper only requires the global objective function $f$ to be strongly-convex-strongly-concave, and the local component function $f_{i,j}$ (also the local function $f_i$) can even be nonconvex-nonconcave. This relaxes the assumption for the state-of-the-art linear convergent decentralized algorithm in Kovalev et al. [39] that requires each $f_i$ to be strongly-convex-strongly-concave.

## 2 Preliminaries

In this section, we introduce the problem setup, followed by a review of the related work.

### 2.1 Problem Setup

We use the bold lowercase letters to represent vectors, e.g., $\mathbf{x} \in \mathbb{R}^{d_x}$ and $\mathbf{y} \in \mathbb{R}^{d_y}$, use $\mathbf{x}_i \in \mathbb{R}^{1 \times d_x}$ and $\mathbf{y}_i \in \mathbb{R}^{1 \times d_y}$ to denote the local variables on the $i$-th node. The bold uppercase letters are used to denote the matrices aggregating the corresponding vectors, such as $\mathbf{X} = [\mathbf{x}_1; \dots; \mathbf{x}_m] \in \mathbb{R}^{m \times d_x}$ and $\mathbf{Y} = [\mathbf{y}_1; \dots; \mathbf{y}_m] \in \mathbb{R}^{m \times d_y}$. The bold lowercase letter with a bar presents the average of local variables, e.g., $\bar{\mathbf{x}} = \frac{1}{m} \sum_{i=1}^m \mathbf{x}_i$ and $\bar{\mathbf{y}} = \frac{1}{m} \sum_{i=1}^m \mathbf{y}_i$. The notations $\mathbf{1}$ and $\mathbf{0}$ are vectors (or matrices) whose entries are all one and zero, respectively. We let $\mathbf{I}$ be the identity matrix. The notation $\| \cdot \|$ represents the Euclidean norm of a vector or the Frobenius norm of a matrix.

For the minimax optimization problem (1), we stack the variables $\mathbf{x} \in \mathbb{R}^{d_x}$ and $\mathbf{y} \in \mathbb{R}^{d_y}$ as $\mathbf{z} = [\mathbf{x}; \mathbf{y}] \in \mathbb{R}^{d_z}$, where $d_z = d_x + d_y$. We further define the gradient operators as

$$\mathbf{g}_{i,j}(\mathbf{z}) = \begin{bmatrix} \nabla_{\mathbf{x}} f_{i,j}(\mathbf{x}, \mathbf{y}) \\ -\nabla_{\mathbf{y}} f_{i,j}(\mathbf{x}, \mathbf{y}) \end{bmatrix}, \quad \mathbf{g}_i(\mathbf{z}) = \frac{1}{n} \sum_{j=1}^{n} \mathbf{g}_{i,j}(\mathbf{z}), \quad \text{and} \quad \mathbf{g}(\mathbf{z}) = \frac{1}{m} \sum_{i=1}^{m} \mathbf{g}_i(\mathbf{z}) \in \mathbb{R}^{d_z}.$$

We make the following assumptions for the decentralized minimax optimization problem (1).

**Assumption 2.1.** The global function $f(\mathbf{x}, \mathbf{y})$ is $\mu$-strongly-convex-$\mu$-strongly-concave, i.e., the function $f(\cdot, \mathbf{y})$ is $\mu$-strongly convex for all given $\mathbf{x} \in \mathbb{R}^{d_x}$ and the function $f(\mathbf{x}, \cdot)$ is $\mu$-strongly concave for all given $\mathbf{y} \in \mathbb{R}^{d_y}$.

**Assumption 2.2** (global smoothness). The global function $f$ is $L$-smooth, i.e., for all $\mathbf{z}, \mathbf{z}' \in \mathbb{R}^{d_z}$, there exists a constant $L > 0$ such that

$$\|\mathbf{g}(\mathbf{z}) - \mathbf{g}(\mathbf{z}')\|^2 \leq L^2 \|\mathbf{z} - \mathbf{z}'\|^2.$$

**Assumption 2.3** (mean-squared smoothness). The function set $\{f_{i,j}\}_{i,j=1}^{m,n}$ is $\bar{L}$-mean-squared smooth, i.e., for all $\mathbf{z}, \mathbf{z}' \in \mathbb{R}^{d_z}$, there exists a constant $\bar{L} > 0$ such that

$$\frac{1}{mn} \sum_{i=1}^{m} \sum_{j=1}^{n} \|\mathbf{g}_{i,j}(\mathbf{z}) - \mathbf{g}_{i,j}(\mathbf{z}')\|^2 \leq \bar{L}^2 \|\mathbf{z} - \mathbf{z}'\|^2.$$

Assumption 2.1 is equivalent to the strong monotonicity of the gradient operator $\mathbf{g}$, i.e., for all $\mathbf{z}, \mathbf{z}' \in \mathbb{R}^{d_z}$, it holds $\langle \mathbf{g}(\mathbf{z}) - \mathbf{g}(\mathbf{z}'), \mathbf{z} - \mathbf{z}' \rangle \geq \mu \|\mathbf{z} - \mathbf{z}'\|^2$. Note that we have

$$\|\mathbf{g}(\mathbf{z}) - \mathbf{g}(\mathbf{z}')\|^2 \leq \frac{1}{mn} \sum_{i=1}^{m} \sum_{j=1}^{n} \|\mathbf{g}_{i,j}(\mathbf{z}) - \mathbf{g}_{i,j}(\mathbf{z}')\|^2 \leq \bar{L}^2 \|\mathbf{z} - \mathbf{z}'\|^2$$

for all $\mathbf{z}, \mathbf{z}' \in \mathbb{R}^{d_z}$. Therefore, it holds that $L \leq \bar{L}$ for the tight parameters $L$ and $\bar{L}$ satisfying Assumptions 2.2 and 2.3. In fact, $\bar{L}$ can be arbitrarily larger than $L$ if there are no convexity/concavity assumption for the local functions [48].

Our complexity analysis considers the upper and lower bounds with respect to the global smoothness parameter $L$ and the mean-squared smoothness parameter $\bar{L}$. In contrast, existing works for decentralized minimax optimization only consider the local smoothness assumptions. For example, Mukherjee and Chakraborty [55] and Luo and Ye [46] assume there exists a constant $L_{\max} > 0$ such that

$$\|\mathbf{g}_{i,j}(\mathbf{z}) - \mathbf{g}_{i,j}(\mathbf{z}')\|^2 \leq L_{\max}^2 \|\mathbf{z} - \mathbf{z}'\|^2$$

for all $\mathbf{z}, \mathbf{z}' \in \mathbb{R}^{d_z}$, $i \in [m]$, and $j \in [n]$; Kovalev et al. [39] assume there exist constants $L_l > 0$ and $\bar{L}_l > 0$ such that

$$\|\mathbf{g}_i(\mathbf{z}) - \mathbf{g}_i(\mathbf{z}')\|^2 \leq L_l^2 \|\mathbf{z} - \mathbf{z}'\|^2 \qquad \text{and} \qquad \frac{1}{n} \sum_{j=1}^{n} \|\mathbf{g}_{i,j}(\mathbf{z}) - \mathbf{g}_{i,j}(\mathbf{z}')\|^2 \leq \bar{L}_l^2 \|\mathbf{z} - \mathbf{z}'\|^2$$

for all $\mathbf{z}, \mathbf{z}' \in \mathbb{R}^{d_z}$ and $i \in [m]$. Noting that the constants $L_l$ and $\bar{L}_l$ are determined by the "worst" local (component) function so that we can verify that $L \leq L_l \leq L_{\max}$ and $\bar{L} \leq \bar{L}_l \leq L_{\max}$ for the tight smoothness parameters that satisfy the above assumptions [48]. The examples in Appendix A demonstrate that the magnitude of these smoothness parameters can differ significantly under data heterogeneity.

In decentralized optimization, each node can only directly communicate with its neighbors. The communication step is usually expressed based on a mixing matrix $\mathbf{W} \in \mathbb{R}^{m \times m}$, which satisfies the following standard assumption [28, 66, 87].

**Assumption 2.4.** Let $\mathbf{W} \in \mathbb{R}^{m \times m}$ be a mixing matrix associated with a network. We assume
  (a) $\mathbf{W}$ is symmetric with $w_{i,j} \geq 0$ for all $i, j$, and $w_{i,j} \neq 0$ if and only if nodes $i$ and $j$ are connected or $i = j$;
  (b) $\mathbf{0} \preceq \mathbf{W} \preceq \mathbf{I}$, $\mathbf{W}^\top \mathbf{1} = \mathbf{W} \mathbf{1} = \mathbf{1}$, and $\mathrm{null}(\mathbf{I} - \mathbf{W}) = \mathrm{span}(\mathbf{1})$.

Table 1: We summarize the complexity for finding the $\epsilon$-suboptimal solution of problem (1). We use the notation $\tilde{\mathcal{O}}(\cdot)$ to hide the logarithmic terms with respect to $m$, $n$, $\mu$, and the smoothness parameters. Note that the computation rounds may not be proportional to the LIFO calls, since distributed algorithms include the scheme of partial participated computation.

| Algorithms | LIFO Calls | Computation Rounds | Communication Rounds |
|---|---|---|---|
| GT-EG [55] | $\mathcal{O}\left(mn\left(\frac{\chi L_{\max}}{\mu}\right)^{4/3}\log\left(\frac{1}{\epsilon}\right)\right)$ | $\mathcal{O}\left(n\left(\frac{\chi L_{\max}}{\mu}\right)^{4/3}\log\left(\frac{1}{\epsilon}\right)\right)$ | $\mathcal{O}\left(\left(\frac{\chi L_{\max}}{\mu}\right)^{4/3}\log\left(\frac{1}{\epsilon}\right)\right)$ |
| MC-SVRE [46] | $\mathcal{O}\left(\left(mn+\frac{m\sqrt{n}L_{\max}}{\mu}\right)\log\left(\frac{1}{\epsilon}\right)\right)$ | $\mathcal{O}\left(\left(n+\frac{\sqrt{n}L_{\max}}{\mu}\right)\log\left(\frac{1}{\epsilon}\right)\right)$ | $\tilde{\mathcal{O}}\left(\sqrt{\chi}\left(n+\frac{\sqrt{n}L_{\max}}{\mu}\right)\log\left(\frac{1}{\epsilon}\right)\right)$ |
| OADSVI [39] | $\mathcal{O}\left(\left(mn+\frac{m\sqrt{n}\bar{L}_l}{\mu}\right)\log\left(\frac{1}{\epsilon}\right)\right)$ | $\mathcal{O}\left(\left(n+\frac{\sqrt{n}\bar{L}_l}{\mu}\right)\log\left(\frac{1}{\epsilon}\right)\right)$ | $\mathcal{O}\left(\frac{\sqrt{\chi}\bar{L}_l}{\mu}\log\left(\frac{1}{\epsilon}\right)\right)$ |
| DIVERSE Theorem 3.7 | $\mathcal{O}\left(\left(mn+\frac{\min\{mnL,\sqrt{mn}\bar{L}\}}{\mu}\right)\log\left(\frac{1}{\epsilon}\right)\right)$ | $\tilde{\mathcal{O}}\left(\left(n+\frac{L}{\mu}+\frac{\min\{nL,\sqrt{n/m}\bar{L}\}}{\mu}\right)\log\left(\frac{1}{\epsilon}\right)\right)$ | $\tilde{\mathcal{O}}\left(\frac{\sqrt{\chi}L}{\mu}\log\left(\frac{1}{\epsilon}\right)\right)$ |
| Lower Bounds Theorem 4.2–4.4 | $\Omega\left(mn+\frac{\min\{mnL,\sqrt{mn}\bar{L}\}}{\mu}\log\left(\frac{1}{\epsilon}\right)\right)$ | $\Omega\left(n+\left(\frac{L}{\mu}+\frac{\min\{nL,\sqrt{n/m}\bar{L}\}}{\mu}\right)\log\left(\frac{1}{\epsilon}\right)\right)$ | $\Omega\left(\frac{\sqrt{\chi}L}{\mu}\log\left(\frac{1}{\epsilon}\right)\right)$ |

Assumption 2.4 indicates that $1 - \lambda_2(\mathbf{W}) > 0$, where $\lambda_2(\mathbf{W})$ is the second largest eigenvalue of $\mathbf{W} \in \mathbb{R}^{m \times m}$. Thus, we can define the characteristic number $\chi := 1/(1 - \lambda_2(\mathbf{W}))$.

In this paper we consider the $\epsilon$-suboptimal solution of problem (1), i.e., the point $\mathbf{z} = (\mathbf{x}, \mathbf{y})$ such that

$$\|\mathbf{x} - \mathbf{x}^*\|^2 + \|\mathbf{y} - \mathbf{y}^*\|^2 \leq \epsilon,$$

where $(\mathbf{x}^*, \mathbf{y}^*)$ is the solution of problem (1) that satisfies $f(\mathbf{x}^*, \mathbf{y}') \leq f(\mathbf{x}^*, \mathbf{y}^*) \leq f(\mathbf{x}', \mathbf{y}^*)$ for all $\mathbf{x}' \in \mathbb{R}^{d_x}$ and $\mathbf{y}' \in \mathbb{R}^{d_y}$. Noting that the solution $(\mathbf{x}^*, \mathbf{y}^*)$ is unique under the strongly-convex-strongly-concave assumption.

## 2.2 Related Work

Significant advancement has been made for decentralized optimization over the last decade. For the minimization problem, the convergence of decentralized gradient descent (DGD) with decaying step sizes has been established by Duchi et al. [22],Tsianos and Rabbat [74], and Jakovetić et al. [31]. The gradient tracking technique was introduced in Nedic et al. [56], Qu and Li [63], and Song et al. [69], so that constant step size can be utilized and linear convergence was achieved for strongly-convex objective. Pu and Nedić [62], Koloskova et al. [33], Ye and Chang [85] further investigated the convergence of gradient tracking under stochastic setting. Scaman et al. [66], Kovalev et al. [38], and Ye et al. [87] introduced the multi-consensus steps by Chebyshev acceleration [5, 43] to further improve the communication complexity. For the objective with the finite-sum structure, Xin et al. [81], Ye et al. [86], Hendrikx et al. [28], and Li et al. [41] integrated the variance reduction techniques to improve the computational efficiency of the algorithms. In recent works [44, 48, 52], different types of smoothness parameters have been considered and sharper complexity bounds have been established for the decentralized finite-sum minimization problems.

For decentralized minimax optimization, Mukherjee and Chakraborty [55] proposed the GT-EG method by combining gradient tracking with EG, proving its linear convergence under the strongly-convex–strongly-concave assumption. Later, Luo and Ye [46] improved the decentralized EG method by incorporating variance reduction [2] and multi-consensus steps [5, 43], achieving better complexity bound on the LIFO calls. It is worth noting that the convergence for both of these methods require the assumption that each component function $f_{i,j}$ is $L_{\max}$-smooth. In a seminal work, Kovalev et al. [39] considered the relaxed conditions that only assume each local function $f_i$ is $L_l$-smooth and each local function set $\{f_{i,j}\}_{j=1}^n$ is $\bar{L}_l$-mean-squared smooth. The authors introduced an extra momentum term into the variance-reduced OGDA method [2], so that they could take the advantage of mini-batch sampling to construct an accurate stochastic gradient estimator, leading to improved computation complexity and communication complexity. The lower bounds were also established therein to justify the optimality of their algorithm with respect to the local smoothness parameters $L_l$ and $\bar{L}_l$. However, none of the previous works on decentralized minimax optimization [39, 46, 55] has considered the potentially tighter complexity bounds with respect to the global smoothness in Assumptions 2.2 and 2.3, which will be well-addressed in this paper. We compare our theoretical results with related work in Table 1.

**Algorithm 1** $\texttt{FastMix}(\mathbf{U}^0, \mathbf{W}, R)$

---

1: **Initialize:** $\mathbf{U}^{-1} = \mathbf{U}^0, \quad \eta_U = \frac{1 - \sqrt{1 - \lambda_2^2(\mathbf{W})}}{1 + \sqrt{1 - \lambda_2^2(\mathbf{W})}}$

2: **for** $r = 0, 1, \ldots, R - 1$ **do**

3: $\quad \mathbf{U}^{r+1} = (1 + \eta_U)\mathbf{W}\mathbf{U}^r - \eta_U \mathbf{U}^{r-1}$

4: **end for**

5: **Output:** $\mathbf{U}^R$

---

**Algorithm 2** DIVERSE

---

1: **Input:** initial point $\mathbf{z}^0$, step size $\eta$, mini-batch size $b$, probability $p \in [0, 1]$, parameters $\alpha, \beta \in [0, 1]$, mixing matrix $\mathbf{W}$, iteration numbers $K$, communication rounds $R$

2: $\mathbf{V}^{-1} = \mathbf{V}^0 = \mathbf{Z}^{-1} = \mathbf{Z}^0 = \mathbf{1}\mathbf{z}^0, \mathbf{S}^{-1} = \mathbf{\Delta}^{-1} = \mathbf{0}$

3: **for** $k = 0, 1, 2, \ldots, K - 1$ **do**

4: $\quad$ **for** $i = 1, 2, \ldots, m$ **in parallel**

5: $\quad\quad \xi_{i,j}^k \overset{\text{i.i.d}}{\sim} \text{Bernoulli}(q)$ with $q = b/(mn)$

6: $\quad\quad \boldsymbol{\delta}_i^k = \mathbf{g}_i(\mathbf{v}_i^{k-1}) + \frac{1}{n}\sum_{j=1}^n \frac{\xi_{i,j}^k}{q}\Big(\mathbf{g}_{i,j}(\mathbf{z}_i^k) - \mathbf{g}_{i,j}(\mathbf{v}_i^{k-1}) + \alpha\big(\mathbf{g}_{i,j}(\mathbf{z}_i^k) - \mathbf{g}_{i,j}(\mathbf{z}_i^{k-1})\big)\Big)$

7: $\quad$ **end for**

8: $\quad \mathbf{S}^k = \texttt{FastMix}(\mathbf{S}^{k-1} + \mathbf{\Delta}^k - \mathbf{\Delta}^{k-1}, \mathbf{W}, R)$

9: $\quad \mathbf{Z}^{k+1} = \texttt{FastMix}((1 - \beta)\mathbf{Z}^k + \beta\mathbf{V}^k - \eta\mathbf{S}^k, \mathbf{W}, R)$

10: $\quad \mathbf{V}^{k+1} = \begin{cases} \texttt{FastMix}(\mathbf{Z}^k, \mathbf{W}, R) & \text{with probability } p, \\ \mathbf{V}^k & \text{with probability } 1 - p \end{cases}$

11: **end for**

12: **Output:** $\mathbf{z}_i^{\text{out}} = \mathbf{z}_i^K$

---

## 3 Algorithm and Complexity Analysis

The proposed decentralized variance-reduced optimistic gradient method with stochastic mini-batch sizes (DIVERSE) is described in Algorithm 2, which is based on a novel sampling strategy and the subroutine of multi-consensus steps (Algorithm 1). The details of the algorithm and its complexity analysis are presented in Sections 3.1 and 3.2, respectively.

### 3.1 Algorithm Design

Recall that the standard OGDA update [19, 54, 61] is given by

$$\mathbf{z}^{k+1} = \mathbf{z}^k - \eta(\underbrace{\mathbf{g}(\mathbf{z}^k) + \mathbf{g}(\mathbf{z}^k) - \mathbf{g}(\mathbf{z}^{k-1})}_{\text{optimistic gradient}}),$$

where $\eta > 0$ is the step size. To improve the computational efficiency by using the finite-sum structure in the local function, DIVERSE constructs the variance-reduced optimistic gradient estimator at node $i$ as follows

$$\boldsymbol{\delta}_i^k = \mathbf{g}_i(\mathbf{v}_i^{k-1}) + \frac{1}{n}\sum_{j=1}^n \frac{\xi_{i,j}^k}{q}\Big(\mathbf{g}_{i,j}(\mathbf{z}_i^k) - \mathbf{g}_{i,j}(\mathbf{v}_i^{k-1}) + \alpha\big(\mathbf{g}_{i,j}(\mathbf{z}_i^k) - \mathbf{g}_{i,j}(\mathbf{z}_i^{k-1})\big)\Big), \quad (3)$$

where $\xi_{i,j}^k \overset{\text{i.i.d}}{\sim} \text{Bernoulli}(q)$ with $q = b/(mn)$ and $\mathbf{v}_i^{k-1}$ is the snapshot point, and $\alpha > 0$ is the momentum parameter. The distribution of $\xi_{i,j}^k$ means we only need to compute the gradient operator $\mathbf{g}_{i,j}$ in equation (3) when $\xi_{i,j}^k = 1$. Note that the snapshot point $\mathbf{v}_i^{k+1}$ is updated with probability $p$ in

each iteration (see Line 10 of Algorithm 2), which implies the term $\mathbf{g}_i(\mathbf{v}_i^{k-1})$ in equation (3) can be reused with probability $1-p$. Therefore, the expected LIFO calls of the algorithm in each iteration is $\mathcal{O}(mnp + (1-p)b)$, which is much more efficient than the cost of accessing the full gradient if we take $p \ll 1$ and $b \ll mn$.

The main difference between DIVERSE and exiting decentralized minimax optimization methods [39, 46, 55] is that the mini-batch size for the local gradient estimator $\boldsymbol{\delta}_i^k$ in equation (3) is not required to be fixed since the variables $\{\xi_{i,j}^k\}_{i,j=1}^{m,n}$ are random. Therefore, the behaviors of all $m$ nodes are similar to the large mini-batch sampling on a single machine. Besides, the steps of gradient tracking and multi-consensus in Lines 8 and 9 of Algorithm 2 ensures that the local variables are sufficiently close to each other, resulting in the sharper complexity bounds with respect to the global smoothness.

## 3.2 Complexity Analysis

For the convergence analysis of DIVERSE (Algorithm 2), define the following Lyapunov function based on the mean vectors as follows

$$\Phi^k := \left(\frac{1}{\eta} + \frac{3\mu}{2}\right)\|\bar{\mathbf{z}}^k - \mathbf{z}^*\|^2 + \frac{\beta}{\eta}\|\bar{\mathbf{z}}^k - \bar{\mathbf{v}}^{k-1}\|^2 + \frac{1}{8\eta}\|\bar{\mathbf{z}}^k - \bar{\mathbf{z}}^{k-1}\|^2$$
$$+ 2\langle \mathbf{g}(\bar{\mathbf{z}}^{k-1}) - \mathbf{g}(\bar{\mathbf{z}}^k), \bar{\mathbf{z}}^k - \mathbf{z}^*\rangle + \frac{\beta + \eta\mu}{p\eta}\|\bar{\mathbf{v}}^k - \mathbf{z}^*\|^2.$$

It is not hard to verify that the Lyapunov function $\Phi^k$ is always non-negative for all $\eta \le 1/(4L)$ (see Appendix B.1).

We first consider the case of $\bar{L} \le \sqrt{mn}L$, in which the Lyapunov function satisfies the following relation.

**Lemma 3.1.** *Under Assumptions 2.1, 2.2, 2.3, and 2.4 with $0 < \mu < L \le \bar{L} \le \sqrt{mn}L$, we run Algorithm 2 with*

$$\eta = \frac{1}{16L}, \quad \beta = p = \frac{\bar{L}}{8L}\max\left\{\frac{\mu}{\bar{L}}, \frac{1}{\sqrt{mn}}\right\}, \quad \alpha = \max\left\{1 - \frac{\mu\eta}{4}, 1 - \frac{p\eta\mu}{\beta + \eta\mu}\right\},$$
$$b = \left\lceil \frac{\bar{L}}{L}\min\left\{\frac{\bar{L}}{\mu}, \sqrt{mn}\right\}\right\rceil, \quad \text{and} \quad R = \mathcal{O}\left(\sqrt{\chi}\log(mn\bar{L}/\mu)\right).$$

*Then it holds that*

$$\mathbb{E}\left[\Phi^{k+1}\right] \le \alpha\mathbb{E}\left[\Phi^k\right] + C_1\left(\mathbb{E}\left[\|\mathbf{Z}^k - \mathbf{1}\bar{\mathbf{z}}^k\|^2\right] + \mathbb{E}\left[\|\mathbf{Z}^{k-1} - \mathbf{1}\bar{\mathbf{z}}^{k-1}\|^2\right] + \mathbb{E}\left[\|\mathbf{V}^{k-1} - \mathbf{1}\bar{\mathbf{v}}^{k-1}\|^2\right]\right),$$

*where $C_1 = (1 + \alpha^2)\left(12n\eta\bar{L}^2/b + 6n\bar{L}^2/\mu\right)$.*

To characterize the convergence of the multi-consensus steps (Algorithm 1), define

$$\rho := \sqrt{14}(1 - (1 - 1/\sqrt{2})\sqrt{1 - \lambda_2(\mathbf{W})})^R.$$

We have $\rho < 1$ if $R$ is sufficient large. See more properties of Algorithm 1 in Appendix B.2.

We then bound the consensus error as follows.

**Lemma 3.2.** *Under the settings of Lemma 3.1, we have*

$$\mathbb{E}\left[\|\mathbf{Z}^{k+1} - \mathbf{1}\bar{\mathbf{z}}^{k+1}\|^2\right] \le 3\rho^2(1 - \beta)^2\mathbb{E}\left[\|\mathbf{Z}^k - \mathbf{1}\bar{\mathbf{z}}^k\|^2\right]$$
$$+ 3\rho^2\beta^2\mathbb{E}\left[\|\mathbf{V}^k - \mathbf{1}\bar{\mathbf{v}}^k\|^2\right] + 3\rho^2\eta^2\mathbb{E}\left[\|\mathbf{S}^k - \mathbf{1}\bar{\mathbf{s}}^k\|^2\right]$$

*and*

$$\mathbb{E}\left[\|\mathbf{V}^{k+1} - \mathbf{1}\bar{\mathbf{v}}^{k+1}\|^2\right] \le p\rho^2\mathbb{E}\left[\|\mathbf{Z}^k - \mathbf{1}\bar{\mathbf{z}}^k\|^2\right] + (1 - p)\mathbb{E}\left[\|\mathbf{V}^k - \mathbf{1}\bar{\mathbf{v}}^k\|^2\right].$$

Noting that the upper bound in Lemma 3.2 depends on the term $\|\mathbf{S}^k - \mathbf{1}\bar{\mathbf{s}}^k\|^2$, the consensus error for $\mathbf{S}$ can be bounded as follows.

**Lemma 3.3.** *Under the settings of Lemma 3.1, we have*

$$\mathbb{E}\left[\|\mathbf{S}^{k+1} - \mathbf{1}\bar{\mathbf{s}}^{k+1}\|^2\right] \leq C_2\rho^2\Big(\mathbb{E}\left[\|\mathbf{Z}^k - \mathbf{1}\bar{\mathbf{z}}^k\|^2\right] + \mathbb{E}\left[\|\mathbf{Z}^{k-1} - \mathbf{1}\bar{\mathbf{z}}^{k-1}\|^2\right] + \mathbb{E}\left[\|\mathbf{V}^k - \mathbf{1}\bar{\mathbf{v}}^k\|^2\right]$$

$$+ \mathbb{E}\left[\|\mathbf{V}^{k-1} - \mathbf{1}\bar{\mathbf{v}}^{k-1}\|^2\right]\Big) + C_3\rho^2\mathbb{E}\left[\|\mathbf{S}^k - \mathbf{1}\bar{\mathbf{s}}^k\|^2\right] + C_4\rho^2\left(\mathbb{E}\left[\Phi^{k+1}\right] + \mathbb{E}\left[\Phi^k\right] + \mathbb{E}\left[\Phi^{k-1}\right]\right),$$

*where $C_2 = 270m^2n^2\bar{L}^2$, $C_3 = 180m^2n^2\bar{L}^2\eta^2 + 2$, $C_4 = 60(16\eta + \eta/\beta)m^3n^2\bar{L}^2$, and $\Phi^{-1} = 0$.*

*Remark* 3.4. Lemma 3.3 shows that the upper bound of $\mathbb{E}\left[\|\mathbf{S}^{k+1} - \mathbf{1}\bar{\mathbf{s}}^{k+1}\|^2\right]$ does not only depend on the consensus error at the $k$-th iteration, but also on that of the $(k-1)$-th iteration. In contrast, the consensus error in the decentralized minimization problem only depends on the term related to the previous iteration [40, 44, 48, 87]. The difference poses a challenge in the analysis, which requires us to develop a novel inductive proof technique.

Applying Lemmas 3.1–3.3, we obtain the linear convergence for the Lyapunov function and consensus errors.

**Lemma 3.5.** *Under the settings of Lemma 3.1, we have*

$$\mathbb{E}\left[\Phi^k\right] \leq \tilde{\alpha}^k\Phi^0, \qquad \mathbb{E}\left[\|\mathbf{Z}^k - \mathbf{1}\bar{\mathbf{z}}^k\|^2\right] \leq \frac{1-\tilde{\alpha}}{4C_1}\tilde{\alpha}^{k+1}\Phi^0,$$

$$\mathbb{E}\left[\|\mathbf{V}^k - \mathbf{1}\bar{\mathbf{v}}^k\|^2\right] \leq \frac{1-\tilde{\alpha}}{4C_1}\tilde{\alpha}^{k+1}\Phi^0, \qquad and \qquad \mathbb{E}\left[\|\mathbf{S}^k - \mathbf{1}\bar{\mathbf{s}}^k\|^2\right] \leq \frac{1-\tilde{\alpha}}{4\eta^2C_1}\tilde{\alpha}^{k+1}\Phi^0,$$

*where $\tilde{\alpha} = \max\left\{1 - \mu\eta/8, 1 - p\eta\mu/(2(\beta + \eta\mu))\right\}$.*

According to the parameter settings in Lemma 3.1, the linear convergence rate $\tilde{\alpha}$ achieved by Lemma 3.5 has the order of $\Theta(1 - \mu/L)$, which depends on the global smoothness. The expected overall LIFO complexity to achieve the $\epsilon$-suboptimal solution is $\mathcal{O}((mn + \sqrt{mn}\bar{L}/\mu)\log(1/\epsilon))$, matching the complexity of variance-reduced EG/OGDA on a single machine [2].

We then consider the case of $L \leq \bar{L}/\sqrt{mn}$. Note that under the setting of Lemma 3.1, one has $b \geq mn$ in this case, which motivates us to use the exact local gradients. That is, we set $p = 0$ and $b = mn$ in Algorithm 2 when $L \leq \bar{L}/\sqrt{mn}$, which leads to $\xi_{i,j}^k = q = 1$ and

$$\boldsymbol{\delta}_i^k = \mathbf{g}_i(\mathbf{z}_i^k) + \alpha\big(\mathbf{g}_i(\mathbf{z}_i^k) - \mathbf{g}_i(\mathbf{z}_i^{k-1})\big).$$

Hence, the snapshot $\mathbf{v}_i^k$ is unnecessary, so we set $\beta = 0$ and define the simplified Lyapunov function

$$\Psi^k := \left(\frac{1}{\eta} + \frac{3\mu}{2}\right)\|\bar{\mathbf{z}}^k - \mathbf{z}^*\|^2 + \frac{3}{4\eta}\|\bar{\mathbf{z}}^k - \bar{\mathbf{z}}^{k-1}\|^2 + 2\langle\mathbf{g}(\bar{\mathbf{z}}^{k-1}) - \mathbf{g}(\bar{\mathbf{z}}^k), \bar{\mathbf{z}}^k - \mathbf{z}^*\rangle.$$

Similar to the analysis of Lemma 3.5, the linear convergence can also be achieved with respect to the global smoothness.

**Lemma 3.6.** *Under Assumptions 2.1, 2.2, 2.3 and 2.4 with $0 < \mu < L \leq \bar{L}/\sqrt{mn}$, we run Algorithm 2 with $\eta = 1/(16L)$, $\beta = p = 0$, $b = mn$, $\alpha = 1 - \mu\eta$, and $R = \mathcal{O}(\sqrt{\chi}\log(mn\bar{L}/\mu))$. Then it holds that*

$$\Psi^k \leq \left(1 - \frac{\mu\eta}{2}\right)^k\Psi^0, \quad \|\mathbf{Z}^k - \mathbf{1}\bar{\mathbf{z}}^k\|^2 \leq \frac{\mu^2\eta}{48n\bar{L}^2}\left(1 - \frac{\mu\eta}{2}\right)^{k+1}\Psi^0,$$

$$and \quad \|\mathbf{S}^k - \mathbf{1}\bar{\mathbf{s}}^k\|^2 \leq \frac{\mu^2}{48\eta n\bar{L}^2}\left(1 - \frac{\mu\eta}{2}\right)^{k+1}\Psi^0.$$

According to Lemma 3.6, we achieve the LIFO complexity of $\mathcal{O}((mnL/\mu)\log(1/\epsilon))$. It is worth noting that the expected overall LIFO complexity of $\mathcal{O}((mn + \sqrt{mn}\bar{L}/\mu)\log(1/\epsilon))$ achieved by variance reduction (under the parameter settings in Lemma 3.1) is worse than the LIFO complexity achieved by iterations with exact local gradients in the case of $L \leq \bar{L}/\sqrt{mn}$. Similar phenomenon is also observed by Luo et al. [48] in nonconvex minimization. Our result implies that the trade-off between variance-reduced gradient estimator and the exact gradient is also necessary in minimax optimization.

Combining the results of Lemmas 3.5 and 3.6 yields the following upper complexity bounds.

**Theorem 3.7.** *Under Assumptions 2.1, 2.2, 2.3 and 2.4 with $0 < \mu < L \leq \bar{L}$, running DIVERSE (Algorithm 2) with appropriate parameter settings can find an $\epsilon$-suboptimal solution at each node, with the expected LIFO complexity of $\mathcal{O}((mn + \min\{mnL, \sqrt{mn}\bar{L}\}/\mu)\log(1/\epsilon))$, the expected computation rounds of $\tilde{\mathcal{O}}((n + L/\mu + \min\{nL, \sqrt{n/m}\bar{L}\}/\mu)\log(1/\epsilon))$, and the communication rounds of $\tilde{\mathcal{O}}(\sqrt{\chi}L/\mu\log(1/\epsilon))$.*

As demonstrated in Table 1, all of our upper bounds in Theorem 3.7 are sharper than state-of-the-art results since we have $L \leq L_l \leq L_{\max}$ and $\bar{L} \leq \bar{L}_l \leq L_{\max}$ for the tight smoothness parameters. Additionally, our LIFO complexity depends on $\sqrt{m}$ in the case of $\bar{L} \leq \sqrt{mn}L$, which is better than existing results that always depends on $m$. These improvements essentially rely on the sampling strategy that does not fix the mini-batch size on different nodes, thereby allowing partial participation to reduce computational costs.

*Remark* 3.8. The computation rounds in Algorithm 2 depend on $\mathbb{E}[\max_{i \in [m]} \sum_{j=1}^{n} \xi_{i,j}^k]$, which may not be proportional to the LIFO calls. We upper bound this quantity by using the locally sub-Gaussian property, which simplifies the analysis in Liu et al. [44] and Luo et al. [48], see Lemma C.1 for details.

# 4 The Lower Complexity Bounds

In this section, we establish the lower complexity bounds of the first-order methods for the decentralized finite-sum minimax optimization. Specifically, we consider the local incremental first-order oracle algorithms as follows.

**Definition 4.1.** A local incremental first-order oracle (LIFO) algorithm over a network of $m$ nodes satisfies the following constraints:

- **Local memory**: Each node $i$ stores vectors in local memories $\mathcal{M}_{i,t}^{\mathbf{x}}$ and $\mathcal{M}_{i,t}^{\mathbf{y}}$ at time $t > 0$. The local memories are updated through local computation or local communication, i.e., for all $i \in [m]$, it holds $\mathcal{M}_{i,t}^{\mathbf{x}} \subseteq \mathcal{M}_{i,t}^{\text{comp},\mathbf{x}} \cup \mathcal{M}_{i,t}^{\text{comm},\mathbf{x}}$ and $\mathcal{M}_{i,t}^{\mathbf{y}} \subseteq \mathcal{M}_{i,t}^{\text{comp},\mathbf{y}} \cup \mathcal{M}_{i,t}^{\text{comm},\mathbf{y}}$.

- **Local computation**: At time $t$, each node $i$ can query the local first-order oracles $\nabla_{\mathbf{x}} f_{i,j}(\mathbf{x}, \mathbf{y})$ and $\nabla_{\mathbf{y}} f_{i,j}(\mathbf{x}, \mathbf{y})$ for any $\mathbf{x} \in \mathcal{M}_{i,t-1}^{\mathbf{x}}$ and $\mathbf{y} \in \mathcal{M}_{i,t-1}^{\mathbf{y}}$. Additionally, the local computational memories $\mathcal{M}_{i,t}^{\text{comp},\mathbf{x}}$ and $\mathcal{M}_{i,t}^{\text{comp},\mathbf{y}}$ satisfy $\mathcal{M}_{i,t}^{\text{comp},\mathbf{x}} = \text{Span}\left(\{\mathbf{x}, \nabla_{\mathbf{x}} f_{i,j}(\mathbf{x}, \mathbf{y}) : \mathbf{x} \in \mathcal{M}_{i,t-1}^{\mathbf{x}}\}\right)$ and $\mathcal{M}_{i,t}^{\text{comp},\mathbf{y}} = \text{Span}\left(\{\mathbf{y}, \nabla_{\mathbf{y}} f_{i,j}(\mathbf{x}, \mathbf{y}) : \mathbf{y} \in \mathcal{M}_{i,t-1}^{\mathbf{y}}\}\right)$.

- **Local communication**: At time $t$, each node $i$ can communicate with its neighbours $\mathcal{N}(i)$. For all $i \in [m]$, the communication memories are defined as $\mathcal{M}_{i,t}^{\text{comm},\mathbf{x}} = \text{Span}(\bigcup_{j \in \mathcal{N}(i),\tau} \mathcal{M}_{j,t-\tau}^{\mathbf{x}})$, and $\mathcal{M}_{i,t}^{\text{comm},\mathbf{y}} = \text{Span}(\bigcup_{j \in \mathcal{N}(i),\tau} \mathcal{M}_{j,t-\tau}^{\mathbf{y}})$, where $\tau$ is a delay parameter satisfying $\tau < t$.

- **Output value**: Each node $i$ specifies local outputs from its memory at time $t$, that is, for all $i \in [m]$, we have $\mathbf{x}_i^t \in \mathcal{M}_{i,t}^{\mathbf{x}}$ and $\mathbf{y}_i^t \in \mathcal{M}_{i,t}^{\mathbf{y}}$.

The definition of the above algorithm class follows the standard settings in the studies of decentralized optimization [8, 28, 44, 48, 66]. Compared with the algorithm classes defined by Kovalev et al. [39], we remove the requirement that all nodes must access their stochastic local gradients with the identical mini-batch size per iteration, so our algorithm class also contains the partial participated computation schemes.

The lower complexity bounds on the LIFO calls, the computation rounds, and the communication rounds are presented in the following three theorems.

**Theorem 4.2.** *For the parameters $\bar{L} \geq L$, $L/\mu > 2$, and $\epsilon < 0.003$, there exists hard instances satisfying Assumptions 2.1–2.4. In order to find an $\epsilon$-suboptimal solution, the LIFO calls of any LIFO algorithm is lower bounded by $\Omega(mn + \min\{mnL, \sqrt{mn}\bar{L}\}/\mu\log(1/\epsilon))$.*

**Theorem 4.3.** *For the parameters $\bar{L} \geq L$, $L/\mu > 2$, and $\epsilon < 0.003$, there exists hard instances satisfying Assumptions 2.1–2.4. In order to find an $\epsilon$-suboptimal solution, the computation rounds of any LIFO algorithm is lower bounded by $\Omega(n + (L/\mu + \min\{nL, \sqrt{n/m}\bar{L}\}/\mu)\log(1/\epsilon))$.*

**Theorem 4.4.** *For the parameters $\bar{L} \geq L \geq 2\mu > 0$, and $m \geq 2, n \in \mathbb{N}$, there exists a hard instance satisfying Assumptions 2.1–2.4 with $\lambda_2(\mathbf{W}) \in [0, \cos(\pi/m)]$. In order to find an $\epsilon$-suboptimal solution, the communication rounds of any LIFO algorithm is lower bounded by $\Omega(\sqrt{\chi}L/\mu\log(1/\epsilon))$.*

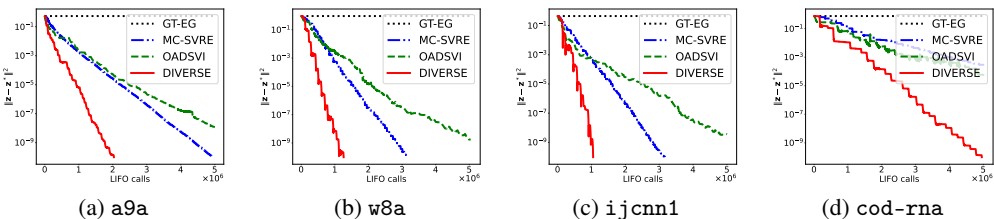

(a) `a9a`  (b) `w8a`  (c) `ijcnn1`  (d) `cod-rna`

Figure 1: Performance comparison with respect to LIFO calls across different datasets.

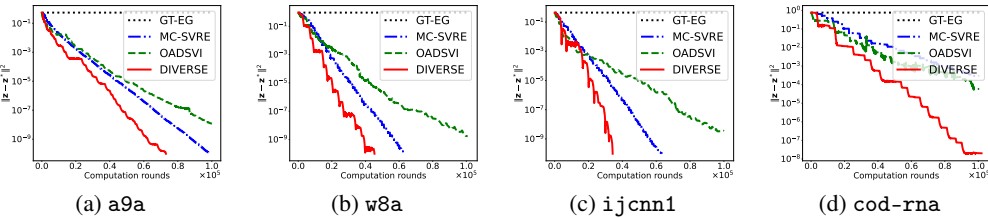

(a) `a9a`  (b) `w8a`  (c) `ijcnn1`  (d) `cod-rna`

Figure 2: Performance comparison with respect to computation rounds across different datasets.

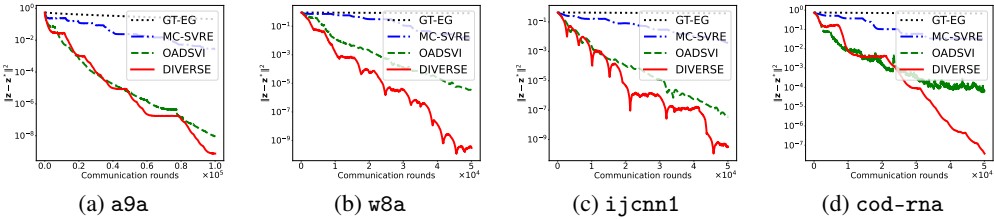

(a) `a9a`  (b) `w8a`  (c) `ijcnn1`  (d) `cod-rna`

Figure 3: Performance comparison with respect to communication rounds across different datasets.

The above theorems indicate that the upper complexity bounds provided in Theorem 3.7 are optimal (up to a logarithmic factor). Our lower bounds hold for the decentralized finite-sum minimax optimization under the general smoothness settings. Specifically, the results in Theorems 4.2–4.4 hold for all $L$ and $\bar{L}$ such that $0 < L \leq \bar{L}$. In contrast, the existing lower bounds [39] consider the local smoothness parameters $L_l$ and $\bar{L}_l$ (see Section 2.1), and their analysis requires the additional condition $\sqrt{n}L_l = \bar{L}_l$.

## 5 Numerical Experiments

In this section, numerical experiments are conducted to evaluate the performance of Algorithm 2. We consider the problem of robust regularized linear regression [12, 27, 39, 50], which is formulated as

$$\min_{\mathbf{x}\in\mathbb{R}^d} \max_{\mathbf{y}\in\mathbb{R}^d} \frac{1}{2N} \sum_{i=1}^{N} \left(\mathbf{x}^\top(\mathbf{a}_i + \mathbf{y}) - b_i\right)^2 + \frac{r_1}{2}\|\mathbf{x}\|^2 - \frac{r_2}{2}\|\mathbf{y}\|^2,$$

where $\mathbf{x}$ is the weight of the model, $\mathbf{y}$ is the adversarial noise, $\{(\mathbf{a}_i, b_i)\}_{i=1}^{N}$ is the training dataset, and $r_1, r_2$ are regularization parameters. We consider the undirected ring network with $m = 50$ nodes and each node has $n$ training samples. Therefore, the total number of samples is $N = mn$. The mixing matrix with Metropolis–Hastings weights [80] is used for communication steps. The regularization parameters are set to be $r_1 = r_2 = 0.2$. The numerical experiments are conducted on datasets `a9a`, `w8a`, `ijcnn1`, and `cod-rna`, from the LIBSVM repository [14]. We compare the proposed DIVERSE (Algorithm 2) with the baseline methods including GT-EG [55], MC-SVRE [46], and OADSVI [39, Algorithm 1]. The parameters of these algorithms are set according to the theoretical analysis or the recommended settings by the authors [39, 46, 55]. Specifically, the parameter $b$ in the DIVERSE is set to be 128 and the fixed batch size for each node in OADSVI is set to be 3. The best performance step sizes from $\{0.1, 0.05, 0.01\}$ are used, up to the algorithms and the datasets.

The experimental results are shown in Figures 1-3. It can be observed that the proposed DIVERSE outperforms all the tested methods in terms of LIFO calls, the computation rounds, and the communication rounds, which validates our theoretical results. The deterministic method GT-EG [55] and the

stochastic method MC-SVRE [46] require much more communication rounds than other methods. This is because GT-EG does not include Chebyshev acceleration in its communication protocol and MC-SVRE cannot benefit from the communication efficiency by the mini-batch sampling. Additionally, the LIFO complexity of DIVERSE is significantly superior to all the baselines, since it is the only one that uses stochastic mini-batch sizes, benefiting from partially participated computations.

## 6    Conclusion

This paper proposes variance-reduced optimistic gradient method with stochastic mini-batch sizes for decentralized convex-concave finite-sum minimax problem. We establish the linear convergence rate with global smoothness parameters dependency for the strongly-convex-strongly-concave objective, which is shaper than existing results that only consider the local smoothness. Lower complexity bounds are constructed to show the near optimality of our method. The efficiency of the proposed method is also validated through numerical experiments. For future direction, we would like to extend the ideas to solve the decentralized minimax problem with different constants of strong convexity and strong concavity [26, 36, 42, 47, 53, 70, 76, 84]. We can also study the global smoothness dependency in decentralized nonconvex minimax optimization [17, 23, 30, 72, 79, 83, 91–93].

## Acknowledgments and Disclosure of Funding

Luo is supported by the Major Key Project of Pengcheng Laboratory (No. PCL2024A06), National Natural Science Foundation of China (No. 12571557), National Natural Science Foundation of China (No. 62206058), and Shanghai Basic Research Program (23JC1401000). Chen and Wei were partially supported by the National Key R&D Program of China (No. 2023YFA1009300).

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

## Appendix

In Appendix A, we construct examples that smoothness parameters are significantly different. In Appendix B, we present some basic results, including the non-negativity of the Lyapunov functions and several useful lemmas. The proofs of the lemmas and Theorem 3.7 in Section 3 are provided in Appendix C, and Appendix D contains the proofs of Theorems 4.2, 4.3 and 4.4 in Section 4.

## A  Examples of Differences among Smoothness Parameters

In this section, we provide two specific examples to show that the smoothness parameters can differ significantly. We first construct an instance where the local smoothness parameters $L_l$, $\bar{L}_l$, and $L_{\max}$ are a factor of $\Theta(\sqrt{m})$ larger than the global parameters $L$ and $\bar{L}$.

*Example* A.1. For simplicity, assume $d_x = d_y$ and $n = 1$. Define the function

$$h(\mathbf{x}, \mathbf{y}) = \sqrt{(L^2 - \mu^2)}\mathbf{x}^\top \mathbf{y} + \frac{\mu}{2}\|\mathbf{x}\|^2 - \frac{\mu}{2}\|\mathbf{y}\|^2, \tag{4}$$

and denote its gradient operator by $\mathbf{g}_h(\mathbf{z})$. It can be verified that for any $\mathbf{z}, \mathbf{z}' \in \mathbb{R}^{d_z}$,

$$\begin{aligned}
&\|\mathbf{g}_h(\mathbf{z}) - \mathbf{g}_h(\mathbf{z}')\|^2 \\
&= \|\nabla_\mathbf{x} h(\mathbf{x}, \mathbf{y}) - \nabla_\mathbf{x} h(\mathbf{x}', \mathbf{y}')\|^2 + \|\nabla_\mathbf{y} h(\mathbf{x}, \mathbf{y}) - \nabla_\mathbf{y} h(\mathbf{x}', \mathbf{y}')\|^2 \\
&= \|\sqrt{(L^2 - \mu^2)}(\mathbf{y} - \mathbf{y}') + \mu(\mathbf{x} - \mathbf{x}')\|^2 + \|\sqrt{(L^2 - \mu^2)}(\mathbf{x} - \mathbf{x}') - \mu(\mathbf{y} - \mathbf{y}')\|^2 \\
&= L^2 \left(\|\mathbf{x} - \mathbf{x}'\|^2 + \|\mathbf{y} - \mathbf{y}'\|^2\right) \\
&= L^2\|\mathbf{z} - \mathbf{z}'\|^2,
\end{aligned}$$

indicating that $h(\mathbf{x}, \mathbf{y})$ is $L$-smooth, and the constant $L$ is tight.

We now define the local objective functions as

$$f_i(\mathbf{x}, \mathbf{y}) = \begin{cases} (1 + \sqrt{m})h(\mathbf{x}, \mathbf{y}), & \text{if } i = 1, \\ (1 - \sqrt{m})h(\mathbf{x}, \mathbf{y}), & \text{if } i = 2, \\ h(\mathbf{x}, \mathbf{y}), & \text{otherwise.} \end{cases}$$

Then the global objective $f(\mathbf{x}, \mathbf{y}) = \frac{1}{m}\sum_{i=1}^m f_i(\mathbf{x}, \mathbf{y}) = h(\mathbf{x}, \mathbf{y})$ remains $L$-smooth. The global mean-squared smoothness parameter $\bar{L}$ can be computed as

$$\begin{aligned}
\frac{1}{m}\sum_{i=1}^m \|\mathbf{g}_i(\mathbf{z}) - \mathbf{g}_i(\mathbf{z}')\|^2 &= \frac{1}{m}\left((1 + \sqrt{m})^2 + (1 - \sqrt{m})^2 + m - 2\right)\|\mathbf{g}_h(\mathbf{z}) - \mathbf{g}_h(\mathbf{z}')\|^2 \\
&= 3L^2\|\mathbf{z} - \mathbf{z}'\|^2.
\end{aligned}$$

Meanwhile, the local smoothness parameters $L_l$, $\bar{L}_l$, and $L_{\max}$ are all determined by $f_1$, since for any $i \in [m]$,

$$\begin{aligned}
\|\mathbf{g}_i(\mathbf{z}) - \mathbf{g}_i(\mathbf{z}')\|^2 &\leq \|\mathbf{g}_1(\mathbf{z}) - \mathbf{g}_1(\mathbf{z}')\|^2 \\
&= (1 + \sqrt{m})^2\|\mathbf{g}_h(\mathbf{z}) - \mathbf{g}_h(\mathbf{z}')\|^2 \\
&= (1 + \sqrt{m})^2 L^2\|\mathbf{z} - \mathbf{z}'\|^2.
\end{aligned}$$

Thus, we conclude that $\bar{L} = \sqrt{3}L$ and $L_l = \bar{L}_l = L_{\max} = (1 + \sqrt{m})L$.

Based on Example A.1 and Table 1 (ignoring all log term), DIVERSE achieves the LIFO calls complexity of $\tilde{\mathcal{O}}(m + \sqrt{m}L/\mu)$, computation rounds complexity of $\tilde{\mathcal{O}}(L/\mu)$, and communication complexity of $\tilde{\mathcal{O}}(\sqrt{\chi}L/\mu)$. In contrast, the existing state-of-the-art method OADSVI requires the LIFO calls complexity of $\tilde{\mathcal{O}}(m + m^{3/2}L/\mu)$, computation rounds complexity of $\tilde{\mathcal{O}}(\sqrt{m}L/\mu)$, and communication complexity of $\tilde{\mathcal{O}}(\sqrt{m\chi}L/\mu)$.

If the heterogeneity among nodes further increases, the local smoothness parameters may exceed $\Omega(\sqrt{m})$ relative to $L$, as demonstrated by the following example. (In fact, the local smoothness parameters can be arbitrarily large relative to $L$.)

*Example* A.2. Using the same definition of $h(\mathbf{x}, \mathbf{y})$ as in equation (4), define the local functions as

$$f_i(\mathbf{x}, \mathbf{y}) = \begin{cases} (1 + m^2)h(\mathbf{x}, \mathbf{y}), & \text{if } i = 1, \\ (1 - m^2)h(\mathbf{x}, \mathbf{y}), & \text{if } i = 2, \\ h(\mathbf{x}, \mathbf{y}), & \text{otherwise.} \end{cases}$$

Then the global objective remains $f(\mathbf{x}, \mathbf{y}) = h(\mathbf{x}, \mathbf{y})$, which is $L$-smooth. Following the calculation as in Example A.1, it is straightforward to verify that $\bar{L} = \sqrt{1 + m^3}L$ and $L_l = \bar{L}_l = L_{\max} = (1 + m^2)L$.

For Example A.2, DIVERSE achieves the LIFO calls complexity of $\tilde{\mathcal{O}}(m + mL/\mu)$, computation rounds complexity of $\tilde{\mathcal{O}}(L/\mu)$, and communication complexity of $\tilde{\mathcal{O}}(\sqrt{\chi}L/\mu)$, while existing methods require the LIFO calls complexity of $\tilde{\mathcal{O}}(m + m^3L/\mu)$, computation rounds complexity of $\tilde{\mathcal{O}}(m^2L/\mu)$, and communication complexity of $\tilde{\mathcal{O}}(m^2\sqrt{\chi}L/\mu)$.

# B    Some Basic Results

In this section, we establish the non-negativity of the Lyapunov functions, and then present some useful lemmas.

## B.1    The Non-Negativity of Lyapunov Functions

In this section, we prove that the defined Lyapunov functions $\Phi$ and $\Psi$ are non-negative when $\eta \leq 1/(4L)$. Recalling the definition of $\Phi^k$, we have

$$
\begin{aligned}
\Phi^k &= \left(\frac{1}{\eta} + \frac{3\mu}{2}\right)\|\bar{\mathbf{z}}^k - \mathbf{z}^*\|^2 + \frac{\beta}{\eta}\|\bar{\mathbf{z}}^k - \bar{\mathbf{v}}^{k-1}\|^2 + \frac{1}{8\eta}\|\bar{\mathbf{z}}^k - \bar{\mathbf{z}}^{k-1}\|^2 \\
&\quad + 2\langle \mathbf{g}(\bar{\mathbf{z}}^{k-1}) - \mathbf{g}(\bar{\mathbf{z}}^k), \bar{\mathbf{z}}^k - \mathbf{z}^* \rangle + \frac{\beta + \eta\mu}{p\eta}\|\bar{\mathbf{v}}^k - \mathbf{z}^*\|^2 \\
&\geq \frac{1}{2\eta}\|\bar{\mathbf{z}}^k - \mathbf{z}^*\|^2 + 2\langle \mathbf{g}(\bar{\mathbf{z}}^{k-1}) - \mathbf{g}(\bar{\mathbf{z}}^k), \bar{\mathbf{z}}^k - \mathbf{z}^* \rangle + \frac{1}{8\eta}\|\bar{\mathbf{z}}^k - \bar{\mathbf{z}}^{k-1}\|^2 \\
&\geq \frac{1}{2\eta}\|\bar{\mathbf{z}}^k - \mathbf{z}^*\|^2 - \frac{1}{8\eta L^2}\|\mathbf{g}(\bar{\mathbf{z}}^k) - \mathbf{g}(\bar{\mathbf{z}}^{k-1})\|^2 - 8\eta L^2\|\bar{\mathbf{z}}^k - \mathbf{z}^*\|^2 + \frac{1}{8\eta}\|\bar{\mathbf{z}}^k - \bar{\mathbf{z}}^{k-1}\|^2 \\
&\geq \frac{1}{2\eta}\|\bar{\mathbf{z}}^k - \mathbf{z}^*\|^2 - \frac{1}{8\eta}\|\bar{\mathbf{z}}^k - \bar{\mathbf{z}}^{k-1}\|^2 - 8\eta L^2\|\bar{\mathbf{z}}^k - \mathbf{z}^*\|^2 + \frac{1}{8\eta}\|\bar{\mathbf{z}}^k - \bar{\mathbf{z}}^{k-1}\|^2 \\
&= \left(\frac{1}{2\eta} - 8\eta L^2\right)\|\bar{\mathbf{z}}^k - \mathbf{z}^*\|^2 \\
&\geq 0,
\end{aligned}
$$

where the third inequality holds by Assumption 2.2 and the last inequality is based on $\eta \leq 1/(4L)$.

Similarly, for $\Psi^k$, we have

$$
\begin{aligned}
\Psi^k &= \left(\frac{1}{\eta} + \frac{3\mu}{2}\right)\|\bar{\mathbf{z}}^k - \mathbf{z}^*\|^2 + \frac{3}{4\eta}\|\bar{\mathbf{z}}^k - \bar{\mathbf{z}}^{k-1}\|^2 + 2\langle \mathbf{g}(\bar{\mathbf{z}}^{k-1}) - \mathbf{g}(\bar{\mathbf{z}}^k), \bar{\mathbf{z}}^k - \mathbf{z}^* \rangle \\
&\geq \frac{1}{2\eta}\|\bar{\mathbf{z}}^k - \mathbf{z}^*\|^2 + 2\langle \mathbf{g}(\bar{\mathbf{z}}^{k-1}) - \mathbf{g}(\bar{\mathbf{z}}^k), \bar{\mathbf{z}}^k - \mathbf{z}^* \rangle + \frac{1}{8\eta}\|\bar{\mathbf{z}}^k - \bar{\mathbf{z}}^{k-1}\|^2 \\
&\geq 0.
\end{aligned}
$$

## B.2    Useful Lemmas

In this section, we present basic lemmas that will be used in the subsequent proofs. Firstly, based on Assumption 2.3, we can derive the following lemma.

**Lemma B.1.** *Under Assumption 2.3, we have*

$$\|\mathbf{g}_{i,j}(\mathbf{z}) - \mathbf{g}_{i,j}(\mathbf{z}')\|^2 \leq mn\bar{L}^2\|\mathbf{z} - \mathbf{z}'\|^2, \tag{5}$$

*and*

$$\|\mathbf{g}_i(\mathbf{z}) - \mathbf{g}_i(\mathbf{z}')\|^2 \le m\bar{L}^2 \|\mathbf{z} - \mathbf{z}'\|^2, \tag{6}$$

*for all* $\mathbf{z}, \mathbf{z}' \in \mathbb{R}^{d_z}$, $i \in [m]$, *and* $j \in [n]$.

*Proof.* For equation (5), it holds because

$$\|\mathbf{g}_{i,j}(\mathbf{z}) - \mathbf{g}_{i,j}(\mathbf{z}')\|^2 \le \sum_{i=1}^m \sum_{j=1}^n \|\mathbf{g}_{i,j}(\mathbf{z}) - \mathbf{g}_{i,j}(\mathbf{z}')\|^2 \le mn\bar{L}^2 \|\mathbf{z} - \mathbf{z}'\|^2,$$

where the last inequality is based on Assumption 2.3. Equation (6) is due to the fact that

$$
\begin{aligned}
\|\mathbf{g}_i(\mathbf{z}) - \mathbf{g}_i(\mathbf{z}')\|^2 &= \|\frac{1}{n} \sum_{j=1}^n \left( \mathbf{g}_{i,j}(\mathbf{z}) - \mathbf{g}_{i,j}(\mathbf{z}') \right) \|^2 \\
&\le \frac{1}{n} \sum_{j=1}^n \|\mathbf{g}_{i,j}(\mathbf{z}) - \mathbf{g}_{i,j}(\mathbf{z}')\|^2 \\
&\le \frac{1}{n} \sum_{i=1}^m \sum_{j=1}^n \|\mathbf{g}_{i,j}(\mathbf{z}) - \mathbf{g}_{i,j}(\mathbf{z}')\|^2 \\
&\le m\bar{L}^2 \|\mathbf{z} - \mathbf{z}'\|^2.
\end{aligned}
$$

This completes the proof. $\square$

The following proposition [87] characterizes the convergence of Algorithm 1.

**Proposition B.2** (Ye et al. [87, Proposition 1]). *Under Assumption 2.4, Algorithm 1 holds that*

$$\frac{1}{m} \mathbf{1}^\top \mathbf{U}^R = \bar{\mathbf{u}}^0$$

*and*

$$\|\mathbf{U}^R - \mathbf{1}\bar{\mathbf{u}}^0\| \le c_1 \left(1 - c_2 \sqrt{1 - \lambda_2(\mathbf{W})}\right)^R \|\mathbf{U}^0 - \mathbf{1}\bar{\mathbf{u}}^0\|,$$

*where* $\bar{\mathbf{u}}^0 = \frac{1}{m} \mathbf{1}^\top \mathbf{U}^0, c_1 = \sqrt{14}$, *and* $c_2 = 1 - 1/\sqrt{2}$.

The following lemma shows a crucial property of gradient tracking.

**Lemma B.3.** *For Algorithm 2, it holds that*

$$\bar{\mathbf{s}}^k = \bar{\boldsymbol{\delta}}^k. \tag{7}$$

*Proof.* From Proposition B.2, combined with the update of $\mathbf{S}^k$ in Algorithm 2 (Line 8), we have

$$
\begin{aligned}
\bar{\mathbf{s}}^k &= \frac{1}{m} \mathbf{1}^\top \mathbf{S}^k \\
&= \frac{1}{m} \mathbf{1}^\top \texttt{FastMix}(\mathbf{S}^{k-1} + \boldsymbol{\Delta}^k - \boldsymbol{\Delta}^{k-1}, \mathbf{W}, R) \\
&= \bar{\mathbf{s}}^{k-1} + \bar{\boldsymbol{\delta}}^k - \bar{\boldsymbol{\delta}}^{k-1}.
\end{aligned}
$$

Note that $\bar{\mathbf{s}}^0 = \bar{\boldsymbol{\delta}}^0$. By induction, it is straightforward to verify that $\bar{\mathbf{s}}^k = \bar{\boldsymbol{\delta}}^k$ holds for $k \ge 0$. $\square$

## C  The Proofs for Section 3

In this section, we prove the lemmas and Theorem 3.7 in Section 3. Specifically, it is organized as follows:

- In Appendix C.1, we prove Lemma 3.1, which provides an upper bound for the Lyapunov function $\Phi$.

- In Appendix C.2, we provide the proof of Lemma 3.2, which bounds the consensus errors of $\mathbf{Z}$ and $\mathbf{V}$.
- In Appendix C.3, Lemma 3.3 is established, bounding the consensus error of $\mathbf{S}$.
- In Appendix C.4, we prove Lemma 3.5 by applying Lemmas 3.1–3.3 and induction.
- In Appendix C.5, Lemma 3.6 is established using a similar approach to Lemma 3.5.
- Theorem 3.7 is proved in Appendix C.6.

## C.1 The Proof of Lemma 3.1

It follows from Line 9 of Algorithm 2 and equation (7) that

$$
\begin{aligned}
\bar{\mathbf{z}}^{k+1} &= \frac{1}{m}\mathbf{1}^\top \mathbf{Z}^{k+1} \\
&= \frac{1}{m}\mathbf{1}^\top \texttt{FastMix}\left((1-\beta)\mathbf{Z}^k + \beta\mathbf{V}^k - \eta\mathbf{S}^k, \mathbf{W}, R\right) \\
&= (1-\beta)\bar{\mathbf{z}}^k + \beta\bar{\mathbf{v}}^k - \eta\bar{\mathbf{s}}^k \\
&= (1-\beta)\bar{\mathbf{z}}^k + \beta\bar{\mathbf{v}}^k - \eta\bar{\boldsymbol{\delta}}^k,
\end{aligned}
$$

which implies

$$
\bar{\mathbf{z}}^{k+1} - \bar{\mathbf{z}}^k = \beta\left(\bar{\mathbf{v}}^k - \bar{\mathbf{z}}^k\right) - \eta\bar{\boldsymbol{\delta}}^k.
$$

We have

$$
\begin{aligned}
\|\bar{\mathbf{z}}^{k+1} - \mathbf{z}^*\|^2 =& \|\bar{\mathbf{z}}^{k+1} - \bar{\mathbf{z}}^k + \bar{\mathbf{z}}^k - \mathbf{z}^*\|^2 \\
=& \|\bar{\mathbf{z}}^k - \mathbf{z}^*\|^2 + 2\langle\bar{\mathbf{z}}^{k+1} - \bar{\mathbf{z}}^k, \bar{\mathbf{z}}^k - \mathbf{z}^*\rangle + \|\bar{\mathbf{z}}^{k+1} - \bar{\mathbf{z}}^k\|^2 \\
=& \|\bar{\mathbf{z}}^k - \mathbf{z}^*\|^2 + 2\langle\bar{\mathbf{z}}^{k+1} - \bar{\mathbf{z}}^k, \bar{\mathbf{z}}^{k+1} - \mathbf{z}^*\rangle - \|\bar{\mathbf{z}}^{k+1} - \bar{\mathbf{z}}^k\|^2 \\
=& \|\bar{\mathbf{z}}^k - \mathbf{z}^*\|^2 + 2\beta\langle\bar{\mathbf{v}}^k - \bar{\mathbf{z}}^k, \bar{\mathbf{z}}^{k+1} - \mathbf{z}^*\rangle \\
& - 2\eta\langle\bar{\boldsymbol{\delta}}^k - \mathbf{g}(\mathbf{z}^*), \bar{\mathbf{z}}^{k+1} - \mathbf{z}^*\rangle - \|\bar{\mathbf{z}}^{k+1} - \bar{\mathbf{z}}^k\|^2 \\
=& \|\bar{\mathbf{z}}^k - \mathbf{z}^*\|^2 + 2\beta\langle\bar{\mathbf{v}}^k - \mathbf{z}^*, \bar{\mathbf{z}}^{k+1} - \mathbf{z}^*\rangle - 2\beta\langle\bar{\mathbf{z}}^k - \mathbf{z}^*, \bar{\mathbf{z}}^{k+1} - \mathbf{z}^*\rangle \\
& - 2\eta\langle\bar{\boldsymbol{\delta}}^k - \mathbf{g}(\mathbf{z}^*), \bar{\mathbf{z}}^{k+1} - \mathbf{z}^*\rangle - \|\bar{\mathbf{z}}^{k+1} - \bar{\mathbf{z}}^k\|^2 \\
=& \|\bar{\mathbf{z}}^k - \mathbf{z}^*\|^2 - \beta\left(\|\bar{\mathbf{z}}^{k+1} - \bar{\mathbf{v}}^k\|^2 - \|\bar{\mathbf{v}}^k - \mathbf{z}^*\|^2 - \|\bar{\mathbf{z}}^{k+1} - \mathbf{z}^*\|^2\right) \\
& + \beta\left(\|\bar{\mathbf{z}}^{k+1} - \bar{\mathbf{z}}^k\|^2 - \|\bar{\mathbf{z}}^k - \mathbf{z}^*\|^2 - \|\bar{\mathbf{z}}^{k+1} - \mathbf{z}^*\|^2\right) \\
& - \|\bar{\mathbf{z}}^{k+1} - \bar{\mathbf{z}}^k\|^2 - 2\eta\langle\bar{\boldsymbol{\delta}}^k - \mathbf{g}(\mathbf{z}^*), \bar{\mathbf{z}}^{k+1} - \mathbf{z}^*\rangle \\
=& (1-\beta)\|\bar{\mathbf{z}}^k - \mathbf{z}^*\|^2 - \beta\|\bar{\mathbf{z}}^{k+1} - \bar{\mathbf{v}}^k\|^2 + \beta\|\bar{\mathbf{v}}^k - \mathbf{z}^*\|^2 - (1-\beta)\|\bar{\mathbf{z}}^{k+1} - \bar{\mathbf{z}}^k\|^2 \\
& - 2\eta\langle\bar{\boldsymbol{\delta}}^k - \mathbf{g}(\mathbf{z}^*), \bar{\mathbf{z}}^{k+1} - \mathbf{z}^*\rangle. \tag{8}
\end{aligned}
$$

Then we bound the term $\mathbb{E}\left[\langle\bar{\boldsymbol{\delta}}^k - \mathbf{g}(\mathbf{z}^*), \bar{\mathbf{z}}^{k+1} - \mathbf{z}^*\rangle\right]$. Note that

$$
\begin{aligned}
\mathbb{E}\left[\langle\bar{\boldsymbol{\delta}}^k - \mathbf{g}(\mathbf{z}^*), \bar{\mathbf{z}}^{k+1} - \mathbf{z}^*\rangle\right] =& \mathbb{E}\left[\langle\bar{\boldsymbol{\delta}}^k - \mathbb{E}_{\xi^k}[\bar{\boldsymbol{\delta}}^k], \bar{\mathbf{z}}^k - \mathbf{z}^*\rangle\right] + \mathbb{E}\left[\langle\bar{\boldsymbol{\delta}}^k - \mathbb{E}_{\xi^k}[\bar{\boldsymbol{\delta}}^k], \bar{\mathbf{z}}^{k+1} - \bar{\mathbf{z}}^k\rangle\right] \\
& + \mathbb{E}\left[\langle\mathbb{E}_{\xi^k}[\bar{\boldsymbol{\delta}}^k] - \mathbf{g}(\mathbf{z}^*), \bar{\mathbf{z}}^{k+1} - \mathbf{z}^*\rangle\right],
\end{aligned} \tag{9}
$$

where $\mathbb{E}_{\xi^k}$ denotes the expectation taken over random variables $\{\xi^k_{i,j}\}_{i,j=1}^{m,n}$.

For the first term of equation (9), we have

$$
\mathbb{E}\left[\langle\bar{\boldsymbol{\delta}}^k - \mathbb{E}_{\xi^k}[\bar{\boldsymbol{\delta}}^k], \bar{\mathbf{z}}^k - \mathbf{z}^*\rangle\right] = \mathbb{E}\left[\mathbb{E}_{\xi^k}\left[\langle\bar{\boldsymbol{\delta}}^k - \mathbb{E}_{\xi^k}[\bar{\boldsymbol{\delta}}^k], \bar{\mathbf{z}}^k - \mathbf{z}^*\rangle\right]\right] = 0. \tag{10}
$$

For the second term of equation (9), we aim to obtain its lower bound, which is equivalent to finding an upper bound for $\mathbb{E}\left[\langle\bar{\boldsymbol{\delta}}^k - \mathbb{E}_{\xi^k}[\bar{\boldsymbol{\delta}}^k], \bar{\mathbf{z}}^k - \bar{\mathbf{z}}^{k+1}\rangle\right]$. By Young's inequality, we have

$$
\mathbb{E}\left[\langle\bar{\boldsymbol{\delta}}^k - \mathbb{E}_{\xi^k}[\bar{\boldsymbol{\delta}}^k], \bar{\mathbf{z}}^k - \bar{\mathbf{z}}^{k+1}\rangle\right] \leq \eta\mathbb{E}\left[\|\bar{\boldsymbol{\delta}}^k - \mathbb{E}_{\xi^k}[\bar{\boldsymbol{\delta}}^k]\|^2\right] + \frac{1}{4\eta}\mathbb{E}\left[\|\bar{\mathbf{z}}^{k+1} - \bar{\mathbf{z}}^k\|^2\right]. \tag{11}
$$

Recalling the definition that

$$\bar{\boldsymbol{\delta}}^k = \frac{1}{m}\sum_{i=1}^m \mathbf{g}_i(\mathbf{v}_i^{k-1}) + \frac{1}{mn}\sum_{i=1}^m \sum_{j=1}^n \frac{\xi_{i,j}^k}{q}\left(\mathbf{g}_{i,j}(\mathbf{z}_i^k) - \mathbf{g}_{i,j}(\mathbf{v}_i^{k-1}) + \alpha(\mathbf{g}_{i,j}(\mathbf{z}_i^k) - \mathbf{g}_{i,j}(\mathbf{z}_i^{k-1}))\right),$$

we have

$$\mathbb{E}_{\xi^k}\left[\|\bar{\boldsymbol{\delta}}^k - \mathbb{E}_{\xi^k}[\bar{\boldsymbol{\delta}}^k]\|^2\right] \le 2\mathbb{E}_{\xi^k}\left[\left\|\frac{1}{mn}\sum_{i=1}^m \sum_{j=1}^n \left(\frac{\xi_{i,j}^k}{q} - 1\right)\left(\mathbf{g}_{i,j}(\mathbf{z}_i^k) - \mathbf{g}_{i,j}(\mathbf{v}_i^{k-1})\right)\right\|^2\right]$$

$$+ 2\mathbb{E}_{\xi^k}\left[\left\|\frac{\alpha}{mn}\sum_{i=1}^m \sum_{j=1}^n \left(\frac{\xi_{i,j}^k}{q} - 1\right)\left(\mathbf{g}_{i,j}(\mathbf{z}_i^k) - \mathbf{g}_{i,j}(\mathbf{z}_i^{k-1})\right)\right\|^2\right]. \tag{12}$$

By Yong's inequality, the first term of equation (12) can be bounded by

$$2\mathbb{E}_{\xi^k}\left[\left\|\frac{1}{mn}\sum_{i=1}^m \sum_{j=1}^n \left(\frac{\xi_{i,j}^k}{q} - 1\right)\left(\mathbf{g}_{i,j}(\mathbf{z}_i^k) - \mathbf{g}_{i,j}(\mathbf{v}_i^{k-1})\right)\right\|^2\right]$$

$$= 2\mathbb{E}_{\xi^k}\left[\left\|\frac{1}{mn}\sum_{i=1}^m \sum_{j=1}^n \left(\frac{\xi_{i,j}^k}{q} - 1\right)\left(\mathbf{g}_{i,j}(\mathbf{z}_i^k) - \mathbf{g}_{i,j}(\bar{\mathbf{z}}^k)\right)\right.\right.$$

$$- \frac{1}{mn}\sum_{i=1}^m \sum_{j=1}^n \left(\frac{\xi_{i,j}^k}{q} - 1\right)\left(\mathbf{g}_{i,j}(\mathbf{v}_i^{k-1}) - \mathbf{g}_{i,j}(\bar{\mathbf{v}}^{k-1})\right)$$

$$\left.\left.+ \frac{1}{mn}\sum_{i=1}^m \sum_{j=1}^n \left(\frac{\xi_{i,j}^k}{q} - 1\right)\left(\mathbf{g}_{i,j}(\bar{\mathbf{z}}^k) - \mathbf{g}_{i,j}(\bar{\mathbf{v}}^{k-1})\right)\right\|^2\right]$$

$$\le 6\mathbb{E}_{\xi^k}\left[\left\|\frac{1}{mn}\sum_{i=1}^m \sum_{j=1}^n \left(\frac{\xi_{i,j}^k}{q} - 1\right)\left(\mathbf{g}_{i,j}(\mathbf{z}_i^k) - \mathbf{g}_{i,j}(\bar{\mathbf{z}}^k)\right)\right\|^2\right]$$

$$+ 6\mathbb{E}_{\xi^k}\left[\left\|\frac{1}{mn}\sum_{i=1}^m \sum_{j=1}^n \left(\frac{\xi_{i,j}^k}{q} - 1\right)\left(\mathbf{g}_{i,j}(\mathbf{v}_i^{k-1}) - \mathbf{g}_{i,j}(\bar{\mathbf{v}}^{k-1})\right)\right\|^2\right] \tag{13}$$

$$+ 6\mathbb{E}_{\xi^k}\left[\left\|\frac{1}{mn}\sum_{i=1}^m \sum_{j=1}^n \left(\frac{\xi_{i,j}^k}{q} - 1\right)\left(\mathbf{g}_{i,j}(\bar{\mathbf{z}}^k) - \mathbf{g}_{i,j}(\bar{\mathbf{v}}^{k-1})\right)\right\|^2\right].$$

For the first term of equation (13), we have

$$\mathbb{E}_{\xi^k}\left[\left\|\frac{1}{mn}\sum_{i=1}^m \sum_{j=1}^n \left(\frac{\xi_{i,j}^k}{q} - 1\right)\left(\mathbf{g}_{i,j}(\mathbf{z}_i^k) - \mathbf{g}_{i,j}(\bar{\mathbf{z}}^k)\right)\right\|^2\right]$$

$$= \frac{1}{m^2 n^2}\sum_{i=1}^m \sum_{j=1}^n \frac{1-q}{q}\left\|\left(\mathbf{g}_{i,j}(\mathbf{z}_i^k) - \mathbf{g}_{i,j}(\bar{\mathbf{z}}^k)\right)\right\|^2$$

$$\le \frac{1}{m^2 n^2 q}\sum_{i=1}^m \sum_{j=1}^n \left\|\left(\mathbf{g}_{i,j}(\mathbf{z}_i^k) - \mathbf{g}_{i,j}(\bar{\mathbf{z}}^k)\right)\right\|^2$$

$$\le \frac{\bar{L}^2}{mnq}\sum_{i=1}^m \sum_{j=1}^n \left\|\mathbf{z}_i^k - \bar{\mathbf{z}}^k\right\|^2$$

$$= \frac{n\bar{L}^2}{b}\left\|\mathbf{Z}^k - \mathbf{1}\bar{\mathbf{z}}^k\right\|^2,$$

where the second inequality holds by equation (5).

Similarly, it holds that

$$\mathbb{E}_{\xi^k}\left[\left\|\frac{1}{mn}\sum_{i=1}^{m}\sum_{j=1}^{n}\left(\frac{\xi_{i,j}^k}{q}-1\right)\left(\mathbf{g}_{i,j}(\mathbf{v}_i^{k-1})-\mathbf{g}_{i,j}(\bar{\mathbf{v}}^{k-1})\right)\right\|^2\right]\leq\frac{n\bar{L}^2}{b}\left\|\mathbf{V}^{k-1}-\mathbf{1}\bar{\mathbf{v}}^{k-1}\right\|^2.$$

For the third term of equation (13),

$$\mathbb{E}_{\xi^k}\left[\left\|\frac{1}{mn}\sum_{i=1}^{m}\sum_{j=1}^{n}\left(\frac{\xi_{i,j}^k}{q}-1\right)\left(\mathbf{g}_{i,j}(\bar{\mathbf{z}}^k)-\mathbf{g}_{i,j}(\bar{\mathbf{v}}^{k-1})\right)\right\|^2\right]$$

$$=\frac{1}{m^2n^2}\sum_{i=1}^{m}\sum_{j=1}^{n}\frac{1-q}{q}\left\|\left(\mathbf{g}_{i,j}(\bar{\mathbf{z}}^k)-\mathbf{g}_{i,j}(\bar{\mathbf{v}}^{k-1})\right)\right\|^2$$

$$\leq\frac{1}{m^2n^2q}\sum_{i=1}^{m}\sum_{j=1}^{n}\left\|\left(\mathbf{g}_{i,j}(\bar{\mathbf{z}}^k)-\mathbf{g}_{i,j}(\bar{\mathbf{v}}^{k-1})\right)\right\|^2$$

$$\leq\frac{\bar{L}^2}{mnq}\left\|\bar{\mathbf{z}}^k-\bar{\mathbf{v}}^{k-1}\right\|^2$$

$$=\frac{\bar{L}^2}{b}\left\|\bar{\mathbf{z}}^k-\bar{\mathbf{v}}^{k-1}\right\|^2,$$

where the second inequality holds by Assumption 2.3.

Thus, an upper bound for equation (13) is given by

$$2\mathbb{E}_{\xi^k}\left[\left\|\frac{1}{mn}\sum_{i=1}^{m}\sum_{j=1}^{n}\left(\frac{\xi_{i,j}^k}{q}-1\right)\left(\mathbf{g}_{i,j}(\mathbf{z}_i^k)-\mathbf{g}_{i,j}(\mathbf{v}_i^{k-1})\right)\right\|^2\right]$$

$$\leq\frac{6n\bar{L}^2}{b}\left\|\mathbf{Z}^k-\mathbf{1}\bar{\mathbf{z}}^k\right\|^2+\frac{6n\bar{L}^2}{b}\left\|\mathbf{V}^{k-1}-\mathbf{1}\bar{\mathbf{v}}^{k-1}\right\|^2+\frac{6\bar{L}^2}{b}\left\|\bar{\mathbf{z}}^k-\bar{\mathbf{v}}^{k-1}\right\|^2.$$

Similarly, the second term of equation (12) can be bounded by

$$\mathbb{E}_{\xi^k}\left[\left\|\frac{\alpha}{mn}\sum_{i=1}^{m}\sum_{j=1}^{n}\left(\frac{\xi_{i,j}^k}{q}-1\right)\left(\mathbf{g}_{i,j}(\mathbf{z}_i^k)-\mathbf{g}_{i,j}(\mathbf{z}_i^{k-1})\right)\right\|^2\right]$$

$$\leq\alpha^2\left(\frac{6n\bar{L}^2}{b}\|\mathbf{Z}^k-\mathbf{1}\bar{\mathbf{z}}^k\|^2+\frac{6n\bar{L}^2}{b}\|\mathbf{Z}^{k-1}-\mathbf{1}\bar{\mathbf{z}}^{k-1}\|^2+\frac{6\bar{L}^2}{b}\|\bar{\mathbf{z}}^k-\bar{\mathbf{z}}^{k-1}\|^2\right)$$

$$=\frac{6n\bar{L}^2\alpha^2}{b}\|\mathbf{Z}^k-\mathbf{1}\bar{\mathbf{z}}^k\|^2+\frac{6n\bar{L}^2\alpha^2}{b}\|\mathbf{Z}^{k-1}-\mathbf{1}\bar{\mathbf{z}}^{k-1}\|^2+\frac{6\bar{L}^2\alpha^2}{b}\|\bar{\mathbf{z}}^k-\bar{\mathbf{z}}^{k-1}\|^2.$$

Consequently, an upper bound for equation (11) is

$$\mathbb{E}\left[\langle\bar{\boldsymbol{\delta}}^k-\mathbb{E}_{\xi^k}[\bar{\boldsymbol{\delta}}^k],\bar{\mathbf{z}}^k-\bar{\mathbf{z}}^{k+1}\rangle\right]$$

$$\leq\eta\mathbb{E}\left[\|\bar{\boldsymbol{\delta}}^k-\mathbb{E}_{\xi^k}[\bar{\boldsymbol{\delta}}^k]\|^2\right]+\frac{1}{4\eta}\mathbb{E}\left[\|\bar{\mathbf{z}}^{k+1}-\bar{\mathbf{z}}^k\|^2\right]$$

$$\leq\left(\frac{6\eta n\bar{L}^2}{b}+\frac{6\eta n\bar{L}^2\alpha^2}{b}\right)\mathbb{E}\left[\|\mathbf{Z}^k-\mathbf{1}\bar{\mathbf{z}}^k\|^2\right]+\frac{6\eta n\bar{L}^2}{b}\mathbb{E}\left[\|\mathbf{V}^{k-1}-\mathbf{1}\bar{\mathbf{v}}^{k-1}\|^2\right]$$

$$+\frac{6\eta n\bar{L}^2\alpha^2}{b}\mathbb{E}\left[\|\mathbf{Z}^{k-1}-\mathbf{1}\bar{\mathbf{z}}^{k-1}\|^2\right]+\frac{6\eta\bar{L}^2}{b}\mathbb{E}\left[\|\bar{\mathbf{z}}^k-\bar{\mathbf{v}}^{k-1}\|^2\right]$$

$$+\frac{6\eta\bar{L}^2\alpha^2}{b}\mathbb{E}\left[\|\bar{\mathbf{z}}^k-\bar{\mathbf{z}}^{k-1}\|^2\right]+\frac{1}{4\eta}\mathbb{E}\left[\|\bar{\mathbf{z}}^{k+1}-\bar{\mathbf{z}}^k\|^2\right]. \tag{14}$$

For the third term of equation (9), we have

$$\mathbb{E}\left[\left\langle \mathbb{E}_{\xi^k}[\bar{\boldsymbol{\delta}}^k] - \mathbf{g}(\mathbf{z}^*), \bar{\mathbf{z}}^{k+1} - \mathbf{z}^*\right\rangle\right]$$

$$= \mathbb{E}\left[\left\langle \frac{1}{m}\sum_{i=1}^m \mathbf{g}_i(\mathbf{z}_i^k) + \alpha\left(\frac{1}{m}\sum_{i=1}^m \mathbf{g}_i(\mathbf{z}_i^k) - \frac{1}{m}\sum_{i=1}^m \mathbf{g}_i(\mathbf{z}_i^{k-1})\right) - \mathbf{g}(\mathbf{z}^*), \bar{\mathbf{z}}^{k+1} - \mathbf{z}^*\right\rangle\right]$$

$$= \mathbb{E}\left[\left\langle \frac{1}{m}\sum_{i=1}^m \mathbf{g}_i(\mathbf{z}_i^k) + \alpha\left(\frac{1}{m}\sum_{i=1}^m \mathbf{g}_i(\mathbf{z}_i^k) - \frac{1}{m}\sum_{i=1}^m \mathbf{g}_i(\mathbf{z}_i^{k-1})\right) - \mathbf{g}(\bar{\mathbf{z}}^k) - \alpha\left(\mathbf{g}(\bar{\mathbf{z}}^k) - \mathbf{g}(\bar{\mathbf{z}}^{k-1})\right), \bar{\mathbf{z}}^{k+1} - \mathbf{z}^*\right\rangle\right]$$

$$\quad + \mathbb{E}\left[\left\langle \mathbf{g}(\bar{\mathbf{z}}^k) + \alpha\left(\mathbf{g}(\bar{\mathbf{z}}^k) - \mathbf{g}(\bar{\mathbf{z}}^{k-1})\right) - \mathbf{g}(\mathbf{z}^*), \bar{\mathbf{z}}^{k+1} - \mathbf{z}^*\right\rangle\right]$$

$$\geq -\frac{1}{\mu}\mathbb{E}\left[\left\|\frac{1}{m}\sum_{i=1}^m \mathbf{g}_i(\mathbf{z}_i^k) + \alpha\left(\frac{1}{m}\sum_{i=1}^m \mathbf{g}_i(\mathbf{z}_i^k) - \frac{1}{m}\sum_{i=1}^m \mathbf{g}_i(\mathbf{z}_i^{k-1})\right) - \mathbf{g}(\bar{\mathbf{z}}^k) - \alpha\left(\mathbf{g}(\bar{\mathbf{z}}^k) - \mathbf{g}(\bar{\mathbf{z}}^{k-1})\right)\right\|^2\right]$$

$$\quad - \frac{\mu}{4}\mathbb{E}\left[\|\bar{\mathbf{z}}^{k+1} - \mathbf{z}^*\|^2\right] + \mathbb{E}\left[\left\langle \mathbf{g}(\bar{\mathbf{z}}^k) + \alpha\left(\mathbf{g}(\bar{\mathbf{z}}^k) - \mathbf{g}(\bar{\mathbf{z}}^{k-1})\right) - \mathbf{g}(\mathbf{z}^*), \bar{\mathbf{z}}^{k+1} - \mathbf{z}^*\right\rangle\right]$$

$$\geq -\frac{3}{\mu}\mathbb{E}\left[\left\|\frac{1}{m}\sum_{i=1}^m \mathbf{g}_i(\mathbf{z}_i^k) - \mathbf{g}(\bar{\mathbf{z}}^k)\right\|^2\right] - \frac{3}{\mu}\mathbb{E}\left[\left\|\frac{\alpha}{m}\sum_{i=1}^m \mathbf{g}_i(\mathbf{z}_i^k) - \mathbf{g}(\bar{\mathbf{z}}^k)\right\|^2\right]$$

$$\quad - \frac{3}{\mu}\mathbb{E}\left[\left\|\frac{\alpha}{m}\sum_{i=1}^m \mathbf{g}_i(\mathbf{z}_i^{k-1}) - \mathbf{g}(\bar{\mathbf{z}}^{k-1})\right\|^2\right] - \frac{\mu}{4}\mathbb{E}\left[\|\bar{\mathbf{z}}^{k+1} - \mathbf{z}^*\|^2\right]$$

$$\quad + \mathbb{E}\left[\left\langle \mathbf{g}(\bar{\mathbf{z}}^k) + \alpha\left(\mathbf{g}(\bar{\mathbf{z}}^k) - \mathbf{g}(\bar{\mathbf{z}}^{k-1})\right) - \mathbf{g}(\mathbf{z}^*), \bar{\mathbf{z}}^{k+1} - \mathbf{z}^*\right\rangle\right], \tag{15}$$

where the inequalities hold by Young's inequality.

Note that

$$\left\|\frac{1}{m}\sum_{i=1}^m \mathbf{g}_i(\mathbf{z}_i^k) - \mathbf{g}_i(\bar{\mathbf{z}}^k)\right\|^2 \leq \frac{1}{m}\sum_{i=1}^m \left\|\mathbf{g}_i(\mathbf{z}_i^k) - \mathbf{g}_i(\bar{\mathbf{z}}^k)\right\|^2$$

$$\leq \frac{1}{mn}\sum_{i=1}^m \sum_{j=1}^n \left\|\mathbf{g}_{i,j}(\mathbf{z}_i^k) - \mathbf{g}_{i,j}(\bar{\mathbf{z}}^k)\right\|^2$$

$$\leq \frac{1}{mn}\sum_{i=1}^m \sum_{j=1}^n mn\bar{L}^2 \left\|\mathbf{z}_i^k - \bar{\mathbf{z}}^k\right\|^2$$

$$= n\bar{L}^2 \left\|\mathbf{Z}^k - \mathbf{1}\bar{\mathbf{z}}^k\right\|^2,$$

where the first inequality is based on the fact $\left\|\frac{1}{m}\sum_{i=1}^m \mathbf{a}_i\right\|^2 \leq \frac{1}{m}\sum_{i=1}^m \|\mathbf{a}_i\|^2$; the second inequality follows similarly; the third inequality is based on equation (5), and

$$\left\langle \mathbf{g}(\bar{\mathbf{z}}^k) + \alpha\left(\mathbf{g}(\bar{\mathbf{z}}^k) - \mathbf{g}(\bar{\mathbf{z}}^{k-1})\right) - \mathbf{g}(\mathbf{z}^*), \bar{\mathbf{z}}^{k+1} - \mathbf{z}^*\right\rangle$$

$$= \left\langle \mathbf{g}(\bar{\mathbf{z}}^k) - \mathbf{g}(\bar{\mathbf{z}}^{k+1}), \bar{\mathbf{z}}^{k+1} - \mathbf{z}^*\right\rangle - \alpha\left\langle \mathbf{g}(\bar{\mathbf{z}}^{k-1}) - \mathbf{g}(\bar{\mathbf{z}}^k), \bar{\mathbf{z}}^{k+1} - \bar{\mathbf{z}}^k\right\rangle$$

$$\quad - \alpha\left\langle \mathbf{g}(\bar{\mathbf{z}}^{k-1}) - \mathbf{g}(\bar{\mathbf{z}}^k), \bar{\mathbf{z}}^k - \mathbf{z}^*\right\rangle + \left\langle \mathbf{g}(\bar{\mathbf{z}}^{k+1}) - \mathbf{g}(\mathbf{z}^*), \bar{\mathbf{z}}^{k+1} - \mathbf{z}^*\right\rangle$$

$$\geq \left\langle \mathbf{g}(\bar{\mathbf{z}}^k) - \mathbf{g}(\bar{\mathbf{z}}^{k+1}), \bar{\mathbf{z}}^{k+1} - \mathbf{z}^*\right\rangle - \frac{\alpha}{2}\left(4\eta\alpha\|\mathbf{g}(\bar{\mathbf{z}}^k) - \mathbf{g}(\bar{\mathbf{z}}^{k-1})\|^2 + \frac{1}{4\eta\alpha}\|\bar{\mathbf{z}}^{k+1} - \bar{\mathbf{z}}^k\|^2\right)$$

$$\quad - \alpha\left\langle \mathbf{g}(\bar{\mathbf{z}}^{k-1}) - \mathbf{g}(\bar{\mathbf{z}}^k), \bar{\mathbf{z}}^k - \mathbf{z}^*\right\rangle + \mu\|\bar{\mathbf{z}}^{k+1} - \mathbf{z}^*\|^2$$

$$\geq \left\langle \mathbf{g}(\bar{\mathbf{z}}^k) - \mathbf{g}(\bar{\mathbf{z}}^{k+1}), \bar{\mathbf{z}}^{k+1} - \mathbf{z}^*\right\rangle - 2\eta\alpha^2 L^2\|\bar{\mathbf{z}}^k - \bar{\mathbf{z}}^{k-1}\|^2 - \frac{1}{8\eta}\|\bar{\mathbf{z}}^{k+1} - \bar{\mathbf{z}}^k\|^2$$

$$\quad - \alpha\left\langle \mathbf{g}(\bar{\mathbf{z}}^{k-1}) - \mathbf{g}(\bar{\mathbf{z}}^k), \bar{\mathbf{z}}^k - \mathbf{z}^*\right\rangle + \mu\|\bar{\mathbf{z}}^{k+1} - \mathbf{z}^*\|^2,$$

where the first inequality holds due to Young's inequality and Assumption 2.1 (equivalently, the strong monotonicity), while the second inequality follows from Assumption 2.2. The lower bound of

equation (15) can be further expressed as

$$
\mathbb{E}\left[\langle \mathbb{E}_{\xi^k}[\bar{\boldsymbol{\delta}}^k] - \mathbf{g}(\mathbf{z}^*), \bar{\mathbf{z}}^{k+1} - \mathbf{z}^* \rangle\right]
$$
$$
\geq -\frac{3n\bar{L}^2}{\mu}(1+\alpha^2)\mathbb{E}\left[\|\mathbf{Z}^k - \mathbf{1}\bar{\mathbf{z}}^k\|^2\right] - \frac{3n\bar{L}^2\alpha^2}{\mu}\mathbb{E}\left[\|\mathbf{Z}^{k-1} - \mathbf{1}\bar{\mathbf{z}}^{k-1}\|^2\right]
$$
$$
+ \mathbb{E}\left[\langle \mathbf{g}(\bar{\mathbf{z}}^k) - \mathbf{g}(\bar{\mathbf{z}}^{k+1}), \bar{\mathbf{z}}^{k+1} - \mathbf{z}^* \rangle\right] - 2\eta\alpha^2 L^2 \mathbb{E}\left[\|\bar{\mathbf{z}}^k - \bar{\mathbf{z}}^{k-1}\|^2\right] - \frac{1}{8\eta}\mathbb{E}\left[\|\bar{\mathbf{z}}^{k+1} - \bar{\mathbf{z}}^k\|^2\right]
$$
$$
- \alpha\mathbb{E}\left[\langle \mathbf{g}(\bar{\mathbf{z}}^{k-1}) - \mathbf{g}(\bar{\mathbf{z}}^k), \bar{\mathbf{z}}^k - \mathbf{z}^* \rangle\right] + \frac{3\mu}{4}\mathbb{E}\left[\|\bar{\mathbf{z}}^{k+1} - \mathbf{z}^*\|^2\right].
$$
$$(16)$$

Combining the results of equations (8), (9), (10), (14), and (16), we obtain the upper bound of $\mathbb{E}\left[\|\bar{\mathbf{z}}^{k+1} - \mathbf{z}^*\|^2\right]$:

$$
\mathbb{E}\left[\|\bar{\mathbf{z}}^{k+1} - \mathbf{z}^*\|^2\right]
$$
$$
= (1-\beta)\mathbb{E}\left[\|\bar{\mathbf{z}}^k - \mathbf{z}^*\|^2\right] - \beta\mathbb{E}\left[\|\bar{\mathbf{z}}^{k+1} - \bar{\mathbf{v}}^k\|^2\right] + \beta\mathbb{E}\left[\|\bar{\mathbf{v}}^k - \mathbf{z}^*\|^2\right]
$$
$$
- (1-\beta)\mathbb{E}\left[\|\bar{\mathbf{z}}^{k+1} - \bar{\mathbf{z}}^k\|^2\right] - 2\eta\mathbb{E}\left[\langle \bar{\boldsymbol{\delta}}^k - \mathbf{g}(\mathbf{z}^*), \bar{\mathbf{z}}^{k+1} - \mathbf{z}^* \rangle\right]
$$
$$
\leq (1-\beta)\mathbb{E}\left[\|\bar{\mathbf{z}}^k - \mathbf{z}^*\|^2\right] - \beta\mathbb{E}\left[\|\bar{\mathbf{z}}^{k+1} - \bar{\mathbf{v}}^k\|^2\right] + \beta\mathbb{E}\left[\|\bar{\mathbf{v}}^k - \mathbf{z}^*\|^2\right]
$$
$$
- (1-\beta)\mathbb{E}\left[\|\bar{\mathbf{z}}^{k+1} - \bar{\mathbf{z}}^k\|^2\right] + \frac{1}{2}\mathbb{E}\left[\|\bar{\mathbf{z}}^{k+1} - \bar{\mathbf{z}}^k\|^2\right]
$$
$$
+ \frac{12\eta^2 n\bar{L}^2(1+\alpha^2)}{b}\mathbb{E}\left[\|\mathbf{Z}^k - \mathbf{1}\bar{\mathbf{z}}^k\|^2\right] + \frac{12\eta^2 n\bar{L}^2}{b}\mathbb{E}\left[\|\mathbf{V}^{k-1} - \mathbf{1}\bar{\mathbf{v}}^{k-1}\|^2\right]
$$
$$
+ \frac{12\eta^2 n\bar{L}^2\alpha^2}{b}\mathbb{E}\left[\|\mathbf{Z}^{k-1} - \mathbf{1}\bar{\mathbf{z}}^{k-1}\|^2\right] + \frac{12\eta^2 \bar{L}^2}{b}\mathbb{E}\left[\|\bar{\mathbf{z}}^k - \bar{\mathbf{v}}^{k-1}\|^2\right]
$$
$$
+ \frac{12\eta^2 \bar{L}^2\alpha^2}{b}\mathbb{E}\left[\|\bar{\mathbf{z}}^k - \bar{\mathbf{z}}^{k-1}\|^2\right] + \frac{6\eta n\bar{L}^2}{\mu}(1+\alpha^2)\mathbb{E}\left[\|\mathbf{Z}^k - \mathbf{1}\bar{\mathbf{z}}^k\|^2\right]
$$
$$
+ \frac{6\eta n\bar{L}^2\alpha^2}{\mu}\mathbb{E}\left[\|\mathbf{Z}^{k-1} - \mathbf{1}\bar{\mathbf{z}}^{k-1}\|^2\right] - 2\eta\mathbb{E}\left[\langle \mathbf{g}(\bar{\mathbf{z}}^k) - \mathbf{g}(\bar{\mathbf{z}}^{k+1}), \bar{\mathbf{z}}^{k+1} - \mathbf{z}^* \rangle\right]
$$
$$
+ 2\eta\alpha\mathbb{E}\left[\langle \mathbf{g}(\bar{\mathbf{z}}^{k-1}) - \mathbf{g}(\bar{\mathbf{z}}^k), \bar{\mathbf{z}}^k - \mathbf{z}^* \rangle\right] + 4\eta^2\alpha^2 L^2\mathbb{E}\left[\|\bar{\mathbf{z}}^k - \bar{\mathbf{z}}^{k-1}\|^2\right]
$$
$$
+ \frac{1}{4}\mathbb{E}\left[\|\bar{\mathbf{z}}^{k+1} - \bar{\mathbf{z}}^k\|^2\right] - \frac{3}{2}\mu\eta\mathbb{E}\left[\|\bar{\mathbf{z}}^{k+1} - \mathbf{z}^*\|^2\right].
$$

Rearranging the above equations yields

$$
\left(\frac{1}{\eta} + \frac{3}{2}\mu\right)\mathbb{E}\left[\|\bar{\mathbf{z}}^{k+1} - \mathbf{z}^*\|^2\right] + \frac{\beta}{\eta}\mathbb{E}\left[\|\bar{\mathbf{z}}^{k+1} - \bar{\mathbf{v}}^k\|^2\right] + \left(\frac{1}{4\eta} - \frac{\beta}{\eta}\right)\mathbb{E}\left[\|\bar{\mathbf{z}}^{k+1} - \bar{\mathbf{z}}^k\|^2\right]
$$
$$
+ 2\mathbb{E}\left[\langle \mathbf{g}(\bar{\mathbf{z}}^k) - \mathbf{g}(\bar{\mathbf{z}}^{k+1}), \bar{\mathbf{z}}^{k+1} - \mathbf{z}^* \rangle\right]
$$
$$
\leq \left(\frac{1}{\eta} - \frac{\beta}{\eta}\right)\mathbb{E}\left[\|\bar{\mathbf{z}}^k - \mathbf{z}^*\|^2\right] + \frac{\beta}{\eta}\mathbb{E}\left[\|\bar{\mathbf{v}}^k - \mathbf{z}^*\|^2\right] + \frac{12\eta\bar{L}^2}{b}\mathbb{E}\left[\|\bar{\mathbf{z}}^k - \bar{\mathbf{v}}^{k-1}\|^2\right]
$$
$$
+ \left(\frac{12\eta\bar{L}^2\alpha^2}{b} + 4\eta\alpha^2 L^2\right)\mathbb{E}\left[\|\bar{\mathbf{z}}^k - \bar{\mathbf{z}}^{k-1}\|^2\right] + 2\alpha\mathbb{E}\left[\langle \mathbf{g}(\bar{\mathbf{z}}^{k-1}) - \mathbf{g}(\bar{\mathbf{z}}^k), \bar{\mathbf{z}}^k - \mathbf{z}^* \rangle\right] \quad (17)
$$
$$
+ \left(\frac{12\eta n\bar{L}^2}{b} + \frac{6n\bar{L}^2}{\mu}\right)(1+\alpha^2)\mathbb{E}\left[\|\mathbf{Z}^k - \mathbf{1}\bar{\mathbf{z}}^k\|^2\right] + \frac{12\eta n\bar{L}^2}{b}\mathbb{E}\left[\|\mathbf{V}^{k-1} - \mathbf{1}\bar{\mathbf{v}}^{k-1}\|^2\right]
$$
$$
+ \left(\frac{12\eta n\bar{L}^2\alpha^2}{b} + \frac{6n\bar{L}^2\alpha^2}{\mu}\right)\mathbb{E}\left[\|\mathbf{Z}^{k-1} - \mathbf{1}\bar{\mathbf{z}}^{k-1}\|^2\right].
$$

According to the update rule (Line 10) in Algorithm 2, we have

$$
\frac{\beta + \eta\mu}{p\eta}\mathbb{E}\left[\|\bar{\mathbf{v}}^{k+1} - \mathbf{z}^*\|^2\right] = \frac{\beta + \eta\mu}{\eta}\mathbb{E}\left[\|\bar{\mathbf{z}}^k - \mathbf{z}^*\|^2\right] + (1-p)\frac{\beta + \eta\mu}{p\eta}\mathbb{E}\left[\|\bar{\mathbf{v}}^k - \mathbf{z}^*\|^2\right]. \quad (18)
$$

By adding equation (18) to both sides of equation (17) and rearranging the terms, we obtain

$$
\left(\frac{1}{\eta} + \frac{3}{2}\mu\right) \mathbb{E}\left[\|\bar{\mathbf{z}}^{k+1} - \mathbf{z}^*\|^2\right] + \frac{\beta}{\eta}\mathbb{E}\left[\|\bar{\mathbf{z}}^{k+1} - \bar{\mathbf{v}}^k\|^2\right] + \left(\frac{1}{4\eta} - \frac{\beta}{\eta}\right)\mathbb{E}\left[\|\bar{\mathbf{z}}^{k+1} - \bar{\mathbf{z}}^k\|^2\right]
$$

$$
+ 2\mathbb{E}\left[\langle \mathbf{g}(\bar{\mathbf{z}}^k) - \mathbf{g}(\bar{\mathbf{z}}^{k+1}), \bar{\mathbf{z}}^{k+1} - \mathbf{z}^*\rangle\right] + \frac{\beta + \eta\mu}{p\eta}\mathbb{E}\left[\|\bar{\mathbf{v}}^{k+1} - \mathbf{z}^*\|^2\right]
$$

$$
\leq \left(\frac{1}{\eta} + \mu\right)\mathbb{E}\left[\|\bar{\mathbf{z}}^k - \mathbf{z}^*\|^2\right] + \frac{12\eta\bar{L}^2}{b}\mathbb{E}\left[\|\bar{\mathbf{z}}^k - \bar{\mathbf{v}}^{k-1}\|^2\right]
$$

$$
+ \left(\frac{12\eta\bar{L}^2\alpha^2}{b} + 4\eta\alpha^2 L^2\right)\mathbb{E}\left[\|\bar{\mathbf{z}}^k - \bar{\mathbf{z}}^{k-1}\|^2\right] \tag{19}
$$

$$
+ 2\alpha\mathbb{E}\left[\langle \mathbf{g}(\bar{\mathbf{z}}^{k-1}) - \mathbf{g}(\bar{\mathbf{z}}^k), \bar{\mathbf{z}}^k - \mathbf{z}^*\rangle\right] + \left(1 - \frac{p\eta\mu}{\beta + \eta\mu}\right)\frac{\beta + \eta\mu}{p\eta}\mathbb{E}\left[\|\bar{\mathbf{v}}^k - \mathbf{z}^*\|^2\right]
$$

$$
+ \left(\frac{12\eta n\bar{L}^2}{b} + \frac{6n\bar{L}^2}{\mu}\right)(1 + \alpha^2)\mathbb{E}\left[\|\mathbf{Z}^k - \mathbf{1}\bar{\mathbf{z}}^k\|^2\right] + \frac{12\eta n\bar{L}^2}{b}\mathbb{E}\left[\|\mathbf{V}^{k-1} - \mathbf{1}\bar{\mathbf{v}}^{k-1}\|^2\right]
$$

$$
+ \left(\frac{12\eta n\bar{L}^2\alpha^2}{b} + \frac{6n\bar{L}^2\alpha^2}{\mu}\right)\mathbb{E}\left[\|\mathbf{Z}^{k-1} - \mathbf{1}\bar{\mathbf{z}}^{k-1}\|^2\right].
$$

Based on the choice of parameters, it follows that

$$
\frac{1}{\eta} + \mu \leq \left(1 - \frac{\mu\eta}{4}\right)\left(\frac{1}{\eta} + \frac{3}{2}\mu\right), \quad \frac{12\eta\bar{L}^2}{b} \leq \frac{\alpha\beta}{\eta},
$$

$$
\frac{1}{4\eta} - \frac{\beta}{\eta} \geq \frac{1}{8\eta}, \quad \text{and} \quad \frac{12\eta\bar{L}^2\alpha^2}{b} + 4\eta\alpha^2 L^2 \leq \frac{\alpha}{8\eta}.
$$

Thus, equation (19) implies that

$$
\left(\frac{1}{\eta} + \frac{3}{2}\mu\right)\mathbb{E}\left[\|\bar{\mathbf{z}}^{k+1} - \mathbf{z}^*\|^2\right] + \frac{\beta}{\eta}\mathbb{E}\left[\|\bar{\mathbf{z}}^{k+1} - \bar{\mathbf{v}}^k\|^2\right] + \frac{1}{8\eta}\mathbb{E}\left[\|\bar{\mathbf{z}}^{k+1} - \bar{\mathbf{z}}^k\|^2\right]
$$

$$
+ 2\mathbb{E}\left[\langle \mathbf{g}(\bar{\mathbf{z}}^k) - \mathbf{g}(\bar{\mathbf{z}}^{k+1}), \bar{\mathbf{z}}^{k+1} - \mathbf{z}^*\rangle\right] + \frac{\beta + \eta\mu}{p\eta}\mathbb{E}\left[\|\bar{\mathbf{v}}^{k+1} - \mathbf{z}^*\|^2\right]
$$

$$
\leq \left(1 - \frac{\mu\eta}{4}\right)\left(\frac{1}{\eta} + \frac{3}{2}\mu\right)\mathbb{E}\left[\|\bar{\mathbf{z}}^k - \mathbf{z}^*\|^2\right] + \alpha \cdot \frac{\beta}{\eta}\mathbb{E}\left[\|\bar{\mathbf{z}}^k - \bar{\mathbf{v}}^{k-1}\|^2\right] + \alpha \cdot \frac{1}{8\eta}\mathbb{E}\left[\|\bar{\mathbf{z}}^k - \bar{\mathbf{z}}^{k-1}\|^2\right]
$$

$$
+ 2\alpha\mathbb{E}\left[\langle \mathbf{g}(\bar{\mathbf{z}}^{k-1}) - \mathbf{g}(\bar{\mathbf{z}}^k), \bar{\mathbf{z}}^k - \mathbf{z}^*\rangle\right] + \left(1 - \frac{p\eta\mu}{\beta + \eta\mu}\right)\frac{\beta + \eta\mu}{p\eta}\mathbb{E}\left[\|\bar{\mathbf{v}}^k - \mathbf{z}^*\|^2\right]
$$

$$
+ \left(\frac{12\eta n\bar{L}^2}{b} + \frac{6n\bar{L}^2}{\mu}\right)(1 + \alpha^2)\left(\mathbb{E}\left[\|\mathbf{Z}^k - \mathbf{1}\bar{\mathbf{z}}^k\|^2\right] + \mathbb{E}\left[\|\mathbf{Z}^{k-1} - \mathbf{1}\bar{\mathbf{z}}^{k-1}\|^2\right]\right.
$$

$$
\left. + \mathbb{E}\left[\|\mathbf{V}^{k-1} - \mathbf{1}\bar{\mathbf{v}}^{k-1}\|^2\right]\right).
$$

Recalling

$$
\alpha = \max\left\{1 - \frac{\mu\eta}{4}, \ 1 - \frac{p\eta\mu}{\beta + \eta\mu}\right\}, \quad C_1 = \left(\frac{12\eta n\bar{L}^2}{b} + \frac{6n\bar{L}^2}{\mu}\right)(1 + \alpha^2),
$$

we thereby complete the proof of Lemma 3.1.

## C.2    The proof of Lemma 3.2

Based on the update rule for $\mathbf{Z}^{k+1}$ in Algorithm 2, we obtain

$$
\mathbb{E}\left[\|\mathbf{Z}^{k+1} - \mathbf{1}\bar{\mathbf{z}}^{k+1}\|^2\right]
$$

$$
= \mathbb{E}\left[\left\|\texttt{FastMix}\left((1 - \beta)\mathbf{Z}^k + \beta\mathbf{V}^k - \eta\mathbf{S}^k, \mathbf{W}, R\right)\right.\right.
$$

$$
\left.\left. - \frac{1}{m}\mathbf{1}\mathbf{1}^\top\texttt{FastMix}\left((1 - \beta)\mathbf{Z}^k + \beta\mathbf{V}^k - \eta\mathbf{S}^k, \mathbf{W}, R\right)\right\|^2\right]
$$

$$\leq \rho^2 \mathbb{E}\left[\left\|(1-\beta)\mathbf{Z}^k + \beta\mathbf{V}^k - \eta\mathbf{S}^k - \mathbf{1}\left((1-\beta)\bar{\mathbf{z}}^k + \beta\bar{\mathbf{v}}^k - \eta\bar{\mathbf{s}}^k\right)\right\|^2\right]$$

$$\leq 3\rho^2(1-\beta)^2\mathbb{E}\left[\|\mathbf{Z}^k - \mathbf{1}\bar{\mathbf{z}}^k\|^2\right] + 3\rho^2\beta^2\mathbb{E}\left[\|\mathbf{V}^k - \mathbf{1}\bar{\mathbf{v}}^k\|^2\right] + 3\rho^2\eta^2\mathbb{E}\left[\|\mathbf{S}^k - \mathbf{1}\bar{\mathbf{s}}^k\|^2\right], \quad (20)$$

where the first inequality follows from Proposition B.2, and the second inequality is due to Young's inequality.

For the consensus error of $\mathbf{V}^{k+1}$, we have

$$\mathbb{E}\left[\|\mathbf{V}^{k+1} - \mathbf{1}\bar{\mathbf{v}}^{k+1}\|^2\right]$$

$$= p\mathbb{E}\left[\left\|\mathtt{FastMix}\left(\mathbf{Z}^k, \mathbf{W}, R\right) - \frac{1}{m}\mathbf{1}\mathbf{1}^\top\mathtt{FastMix}\left(\mathbf{Z}^k, \mathbf{W}, R\right)\right\|^2\right] + (1-p)\mathbb{E}\left[\|\mathbf{V}^k - \mathbf{1}\bar{\mathbf{v}}^k\|^2\right]$$

$$\leq p\rho^2\mathbb{E}\left[\|\mathbf{Z}^k - \mathbf{1}\bar{\mathbf{z}}^k\|^2\right] + (1-p)\mathbb{E}\left[\|\mathbf{V}^k - \mathbf{1}\bar{\mathbf{v}}^k\|^2\right],$$

where the inequality holds by Proposition B.2.

### C.3 The proof of Lemma 3.3

According to the update rule for $\mathbf{S}^{k+1}$ in Algorithm 2, we have

$$\|\mathbf{S}^{k+1} - \mathbf{1}\bar{\mathbf{s}}^{k+1}\|^2$$

$$= \left\|\mathtt{FastMix}\left(\mathbf{S}^k + \mathbf{\Delta}^{k+1} - \mathbf{\Delta}^k, \mathbf{W}, R\right) - \frac{1}{m}\mathbf{1}\mathbf{1}^\top\mathtt{FastMix}\left(\mathbf{S}^k + \mathbf{\Delta}^{k+1} - \mathbf{\Delta}^k, \mathbf{W}, R\right)\right\|^2$$

$$\leq \rho^2\left\|\mathbf{S}^k + \mathbf{\Delta}^{k+1} - \mathbf{\Delta}^k - \frac{1}{m}\mathbf{1}\mathbf{1}^\top\left(\mathbf{S}^k + \mathbf{\Delta}^{k+1} - \mathbf{\Delta}^k\right)\right\|^2$$

$$\leq 2\rho^2\|\mathbf{S}^k - \mathbf{1}\bar{\mathbf{s}}^k\|^2 + 2\rho^2\|\mathbf{\Delta}^{k+1} - \mathbf{\Delta}^k - \frac{1}{m}\mathbf{1}\mathbf{1}^\top(\mathbf{\Delta}^{k+1} - \mathbf{\Delta}^k)\|^2$$

$$\leq 2\rho^2\|\mathbf{S}^k - \mathbf{1}\bar{\mathbf{s}}^k\|^2 + 2\rho^2\|\mathbf{\Delta}^{k+1} - \mathbf{\Delta}^k\|^2, \quad (21)$$

where the second inequality holds due to Young's inequality, and the last inequality follows from the fact that $\|\mathbf{A} - \frac{1}{m}\mathbf{1}\mathbf{1}^\top\mathbf{A}\| \leq \|\mathbf{A}\|$ for any $\mathbf{A} \in \mathbb{R}^{m \times d_z}$.

Next, we bound the term $\mathbb{E}\left[\|\mathbf{\Delta}^{k+1} - \mathbf{\Delta}^k\|^2\right]$. It follows that

$$\mathbb{E}\left[\|\mathbf{\Delta}^{k+1} - \mathbf{\Delta}^k\|^2\right]$$

$$= \sum_{i=1}^m \mathbb{E}\left[\|\boldsymbol{\delta}_i^{k+1} - \boldsymbol{\delta}_i^k\|^2\right]$$

$$= \sum_{i=1}^m \mathbb{E}\left[\left\|\mathbf{g}_i(\mathbf{v}_i^k) - \mathbf{g}_i(\mathbf{v}_i^{k-1}) + \frac{1}{n}\sum_{j=1}^n \frac{\xi_{i,j}^k}{q}\left(\left(\mathbf{g}_{i,j}(\mathbf{z}_i^{k+1}) - \mathbf{g}_{i,j}(\mathbf{v}_i^k)\right) + \alpha\left(\mathbf{g}_{i,j}(\mathbf{z}_i^{k+1}) - \mathbf{g}_{i,j}(\mathbf{z}_i^k)\right)\right)\right.\right.$$

$$\left.\left.- \frac{1}{n}\sum_{j=1}^n \frac{\xi_{i,j}^{k-1}}{q}\left(\left(\mathbf{g}_{i,j}(\mathbf{z}_i^k) - \mathbf{g}_{i,j}(\mathbf{v}_i^{k-1})\right) + \alpha\left(\mathbf{g}_{i,j}(\mathbf{z}_i^k) - \mathbf{g}_{i,j}(\mathbf{z}_i^{k-1})\right)\right)\right\|^2\right]$$

$$\leq 5\sum_{i=1}^m \mathbb{E}\left[\|\mathbf{g}_i(\mathbf{v}_i^k) - \mathbf{g}_i(\mathbf{v}_i^{k-1})\|^2\right] + 5\sum_{i=1}^m \mathbb{E}\left[\left\|\frac{1}{n}\sum_{j=1}^n \frac{\xi_{i,j}^k}{q}\left(\mathbf{g}_{i,j}(\mathbf{z}_i^{k+1}) - \mathbf{g}_{i,j}(\mathbf{v}_i^k)\right)\right\|^2\right] \quad (22)$$

$$+ 5\sum_{i=1}^m \mathbb{E}\left[\left\|\frac{\alpha}{n}\sum_{j=1}^n \frac{\xi_{i,j}^k}{q}\left(\mathbf{g}_{i,j}(\mathbf{z}_i^{k+1}) - \mathbf{g}_{i,j}(\mathbf{z}_i^k)\right)\right\|^2\right]$$

$$+ 5\sum_{i=1}^m \mathbb{E}\left[\left\|\frac{1}{n}\sum_{j=1}^n \frac{\xi_{i,j}^{k-1}}{q}\left(\mathbf{g}_{i,j}(\mathbf{z}_i^k) - \mathbf{g}_{i,j}(\mathbf{v}_i^{k-1})\right)\right\|^2\right]$$

$$+ 5\sum_{i=1}^m \mathbb{E}\left[\left\|\frac{\alpha}{n}\sum_{j=1}^n \frac{\xi_{i,j}^{k-1}}{q}\left(\mathbf{g}_{i,j}(\mathbf{z}_i^k) - \mathbf{g}_{i,j}(\mathbf{z}_i^{k-1})\right)\right\|^2\right].$$

For the first term of equation (22), it holds that

$$
\begin{aligned}
&\mathbb{E}\left[\|\mathbf{g}_i(\mathbf{v}_i^k) - \mathbf{g}_i(\mathbf{v}_i^{k-1})\|^2\right] \\
&\leq m\bar{L}^2\mathbb{E}\left[\|\mathbf{v}_i^k - \mathbf{v}_i^{k-1}\|^2\right] \\
&\leq 3m\bar{L}^2\mathbb{E}\left[\|\mathbf{v}_i^k - \bar{\mathbf{v}}^k\|^2\right] + 3m\bar{L}^2\mathbb{E}\left[\|\bar{\mathbf{v}}^k - \bar{\mathbf{v}}^{k-1}\|^2\right] + 3m\bar{L}^2\mathbb{E}\left[\|\mathbf{v}_i^{k-1} - \bar{\mathbf{v}}^{k-1}\|^2\right],
\end{aligned}
\tag{23}
$$

where the first inequality holds by equation (6) and the second inequality is due to Young's inequality.
For the other terms in equation (22), Note that

$$
\begin{aligned}
&\mathbb{E}_{\xi^k}\left[\left\|\frac{1}{n}\sum_{j=1}^n \frac{\xi_{i,j}^k}{q}\left(\mathbf{g}_{i,j}(\mathbf{z}_i^{k+1}) - \mathbf{g}_{i,j}(\mathbf{v}_i^k)\right)\right\|^2\right] \\
&\leq \frac{1}{n}\sum_{j=1}^n \mathbb{E}_{\xi^k}\left[\left\|\frac{\xi_{i,j}^k}{q}\left(\mathbf{g}_{i,j}(\mathbf{z}_i^{k+1}) - \mathbf{g}_{i,j}(\mathbf{v}_i^k)\right)\right\|^2\right] \\
&= \frac{1}{nq}\sum_{j=1}^n \|\mathbf{g}_{i,j}(\mathbf{z}_i^{k+1}) - \mathbf{g}_{i,j}(\mathbf{v}_i^k)\|^2 \\
&\leq \frac{mn\bar{L}^2}{q}\|\mathbf{z}_i^{k+1} - \mathbf{v}_i^k\|^2 \\
&\leq \frac{3m^2n^2\bar{L}^2}{b}\|\mathbf{z}_i^{k+1} - \bar{\mathbf{z}}^{k+1}\|^2 + \frac{3m^2n^2\bar{L}^2}{b}\|\bar{\mathbf{z}}^{k+1} - \bar{\mathbf{v}}^k\|^2 + \frac{3m^2n^2\bar{L}^2}{b}\|\mathbf{v}_i^k - \bar{\mathbf{v}}^k\|^2,
\end{aligned}
$$

where the first inequality is based on the fact that $\left\|\frac{1}{n}\sum_{i=1}^n \mathbf{a}_i\right\|^2 \leq \frac{1}{n}\sum_{i=1}^n \|\mathbf{a}_i\|^2$; the second inequality is based on equation (5); the last inequality holds by Young's inequality. Further, we obtain

$$
\begin{aligned}
&\mathbb{E}\left[\left\|\frac{1}{n}\sum_{j=1}^n \frac{\xi_{i,j}^k}{q}\left(\mathbf{g}_{i,j}(\mathbf{z}_i^{k+1}) - \mathbf{g}_{i,j}(\mathbf{v}_i^k)\right)\right\|^2\right] \\
&\leq \frac{3m^2n^2\bar{L}^2}{b}\mathbb{E}\left[\|\mathbf{z}_i^{k+1} - \bar{\mathbf{z}}^{k+1}\|^2\right] + \frac{3m^2n^2\bar{L}^2}{b}\mathbb{E}\left[\|\bar{\mathbf{z}}^{k+1} - \bar{\mathbf{v}}^k\|^2\right] + \frac{3m^2n^2\bar{L}^2}{b}\mathbb{E}\left[\|\mathbf{v}_i^k - \bar{\mathbf{v}}^k\|^2\right].
\end{aligned}
\tag{24}
$$

Similarly, for the third term in equation (22), we have

$$
\begin{aligned}
&\mathbb{E}\left[\left\|\frac{\alpha}{n}\sum_{j=1}^n \frac{\xi_{i,j}^k}{q}\left(\mathbf{g}_{i,j}(\mathbf{z}_i^{k+1}) - \mathbf{g}_{i,j}(\mathbf{z}_i^k)\right)\right\|^2\right] \\
&\leq \frac{3m^2n^2\bar{L}^2\alpha^2}{b}\mathbb{E}\left[\|\mathbf{z}_i^{k+1} - \bar{\mathbf{z}}^{k+1}\|^2\right] + \frac{3m^2n^2\bar{L}^2\alpha^2}{b}\mathbb{E}\left[\|\bar{\mathbf{z}}^{k+1} - \bar{\mathbf{z}}^k\|^2\right] + \frac{3m^2n^2\bar{L}^2\alpha^2}{b}\mathbb{E}\left[\|\mathbf{z}_i^k - \bar{\mathbf{z}}^k\|^2\right].
\end{aligned}
\tag{25}
$$

For the fourth term, it follows that

$$
\begin{aligned}
&\mathbb{E}\left[\left\|\frac{1}{n}\sum_{j=1}^n \frac{\xi_{i,j}^{k-1}}{q}\left(\mathbf{g}_{i,j}(\mathbf{z}_i^k) - \mathbf{g}_{i,j}(\mathbf{v}_i^{k-1})\right)\right\|^2\right] \\
&\leq \frac{3m^2n^2\bar{L}^2}{b}\mathbb{E}\left[\|\mathbf{z}_i^k - \bar{\mathbf{z}}^k\|^2\right] + \frac{3m^2n^2\bar{L}^2}{b}\mathbb{E}\left[\|\bar{\mathbf{z}}^k - \bar{\mathbf{v}}^{k-1}\|^2\right] + \frac{3m^2n^2\bar{L}^2}{b}\mathbb{E}\left[\|\mathbf{v}_i^{k-1} - \bar{\mathbf{v}}^{k-1}\|^2\right].
\end{aligned}
\tag{26}
$$

For the fifth term, it holds that

$$
\begin{aligned}
&\mathbb{E}\left[\left\|\frac{\alpha}{n}\sum_{j=1}^n \frac{\xi_{i,j}^{k-1}}{q}\left(\mathbf{g}_{i,j}(\mathbf{z}_i^k) - \mathbf{g}_{i,j}(\mathbf{z}_i^{k-1})\right)\right\|^2\right] \\
&\leq \frac{3m^2n^2\bar{L}^2\alpha^2}{b}\mathbb{E}\left[\|\mathbf{z}_i^k - \bar{\mathbf{z}}^k\|^2\right] + \frac{3m^2n^2\bar{L}^2\alpha^2}{b}\mathbb{E}\left[\|\bar{\mathbf{z}}^k - \bar{\mathbf{z}}^{k-1}\|^2\right] + \frac{3m^2n^2\bar{L}^2\alpha^2}{b}\mathbb{E}\left[\|\mathbf{z}_i^{k-1} - \bar{\mathbf{z}}^{k-1}\|^2\right].
\end{aligned}
\tag{27}
$$

Substituting the results of equations (23), (24), (25), (26), and (27) into equation (22), we finally obtain an upper bound for $\mathbb{E}\left[\|\mathbf{\Delta}^{k+1} - \mathbf{\Delta}^k\|^2\right]$ as

$$\mathbb{E}\left[\|\mathbf{\Delta}^{k+1} - \mathbf{\Delta}^k\|^2\right]$$

$$\leq 15m\bar{L}^2\mathbb{E}\left[\|\mathbf{V}^k - \mathbf{1}\bar{\mathbf{v}}^k\|^2\right] + 15m^2\bar{L}^2\mathbb{E}\left[\|\bar{\mathbf{v}}^k - \bar{\mathbf{v}}^{k-1}\|^2\right] + 15m\bar{L}^2\mathbb{E}\left[\|\mathbf{V}^{k-1} - \mathbf{1}\bar{\mathbf{v}}^{k-1}\|^2\right]$$

$$+ \frac{15m^2n^2\bar{L}^2}{b}\mathbb{E}\left[\|\mathbf{Z}^{k+1} - \mathbf{1}\bar{\mathbf{z}}^{k+1}\|^2\right] + \frac{15m^3n^2\bar{L}^2}{b}\mathbb{E}\left[\|\bar{\mathbf{z}}^{k+1} - \bar{\mathbf{v}}^k\|^2\right]$$

$$+ \frac{15m^2n^2\bar{L}^2}{b}\mathbb{E}\left[\|\mathbf{V}^k - \mathbf{1}\bar{\mathbf{v}}^k\|^2\right] + \frac{15m^2n^2\bar{L}^2\alpha^2}{b}\mathbb{E}\left[\|\mathbf{Z}^{k+1} - \mathbf{1}\bar{\mathbf{z}}^{k+1}\|^2\right]$$

$$+ \frac{15m^3n^2\bar{L}^2\alpha^2}{b}\mathbb{E}\left[\|\bar{\mathbf{z}}^{k+1} - \bar{\mathbf{z}}^k\|^2\right] + \frac{15m^2n^2\bar{L}^2\alpha^2}{b}\mathbb{E}\left[\|\mathbf{Z}^k - \mathbf{1}\bar{\mathbf{z}}^k\|^2\right]$$

$$+ \frac{15m^2n^2\bar{L}^2}{b}\mathbb{E}\left[\|\mathbf{Z}^k - \mathbf{1}\bar{\mathbf{z}}^k\|^2\right] + \frac{15m^3n^2\bar{L}^2}{b}\mathbb{E}\left[\|\bar{\mathbf{z}}^k - \bar{\mathbf{v}}^{k-1}\|^2\right]$$

$$+ \frac{15m^2n^2\bar{L}^2}{b}\mathbb{E}\left[\|\mathbf{V}^{k-1} - \mathbf{1}\bar{\mathbf{v}}^{k-1}\|^2\right] + \frac{15m^2n^2\bar{L}^2\alpha^2}{b}\mathbb{E}\left[\|\mathbf{Z}^k - \mathbf{1}\bar{\mathbf{z}}^k\|^2\right]$$

$$+ \frac{15m^3n^2\bar{L}^2\alpha^2}{b}\mathbb{E}\left[\|\bar{\mathbf{z}}^k - \bar{\mathbf{z}}^{k-1}\|^2\right] + \frac{15m^2n^2\bar{L}^2\alpha^2}{b}\mathbb{E}\left[\|\mathbf{Z}^{k-1} - \mathbf{1}\bar{\mathbf{z}}^{k-1}\|^2\right]$$

$$\leq 30m^2n^2\bar{L}^2\mathbb{E}\left[\|\mathbf{V}^k - \mathbf{1}\bar{\mathbf{v}}^k\|^2\right] + 15m^2\bar{L}^2\mathbb{E}\left[\|\bar{\mathbf{v}}^k - \bar{\mathbf{v}}^{k-1}\|^2\right] + 30m^2n^2\bar{L}^2\mathbb{E}\left[\|\mathbf{V}^{k-1} - \mathbf{1}\bar{\mathbf{v}}^{k-1}\|^2\right]$$

$$+ 30m^2n^2\bar{L}^2\mathbb{E}\left[\|\mathbf{Z}^{k+1} - \mathbf{1}\bar{\mathbf{z}}^{k+1}\|^2\right] + 45m^2n^2\bar{L}^2\mathbb{E}\left[\|\mathbf{Z}^k - \mathbf{1}\bar{\mathbf{z}}^k\|^2\right]$$

$$+ 15m^2n^2\bar{L}^2\alpha^2\mathbb{E}\left[\|\mathbf{Z}^{k-1} - \mathbf{1}\bar{\mathbf{z}}^{k-1}\|^2\right] + 15m^3n^2\bar{L}^2\mathbb{E}\left[\|\bar{\mathbf{z}}^{k+1} - \bar{\mathbf{v}}^k\|^2\right]$$

$$+ 15m^3n^2\bar{L}^2\alpha^2\mathbb{E}\left[\|\bar{\mathbf{z}}^{k+1} - \bar{\mathbf{z}}^k\|^2\right] + 15m^3n^2\bar{L}^2\mathbb{E}\left[\|\bar{\mathbf{z}}^k - \bar{\mathbf{v}}^{k-1}\|^2\right]$$

$$+ 15m^3n^2\bar{L}^2\alpha^2\mathbb{E}\left[\|\bar{\mathbf{z}}^k - \bar{\mathbf{z}}^{k-1}\|^2\right]$$

$$\leq 30m^2n^2\bar{L}^2\mathbb{E}\left[\|\mathbf{V}^k - \mathbf{1}\bar{\mathbf{v}}^k\|^2\right] + 15m^2\bar{L}^2\mathbb{E}\left[\|\bar{\mathbf{v}}^k - \bar{\mathbf{v}}^{k-1}\|^2\right] + 30m^2n^2\bar{L}^2\mathbb{E}\left[\|\mathbf{V}^{k-1} - \mathbf{1}\bar{\mathbf{v}}^{k-1}\|^2\right]$$

$$+ 30m^2n^2\bar{L}^2\left(3\rho^2(1-\beta)^2\mathbb{E}\left[\|\mathbf{Z}^k - \mathbf{1}\bar{\mathbf{z}}^k\|^2\right] + 3\rho^2\beta^2\mathbb{E}\left[\|\mathbf{V}^k - \mathbf{1}\bar{\mathbf{v}}^k\|^2\right]\right.$$

$$\left. + 3\rho^2\eta^2\mathbb{E}\left[\|\mathbf{S}^k - \mathbf{1}\bar{\mathbf{s}}^k\|^2\right]\right) + 45m^2n^2\bar{L}^2\mathbb{E}\left[\|\mathbf{Z}^k - \mathbf{1}\bar{\mathbf{z}}^k\|^2\right]$$

$$+ 15m^2n^2\bar{L}^2\alpha^2\mathbb{E}\left[\|\mathbf{Z}^{k-1} - \mathbf{1}\bar{\mathbf{z}}^{k-1}\|^2\right] + 15m^3n^2\bar{L}^2\mathbb{E}\left[\|\bar{\mathbf{z}}^{k+1} - \bar{\mathbf{v}}^k\|^2\right]$$

$$+ 15m^3n^2\bar{L}^2\alpha^2\mathbb{E}\left[\|\bar{\mathbf{z}}^{k+1} - \bar{\mathbf{z}}^k\|^2\right] + 15m^3n^2\bar{L}^2\mathbb{E}\left[\|\bar{\mathbf{z}}^k - \bar{\mathbf{v}}^{k-1}\|^2\right]$$

$$+ 15m^3n^2\bar{L}^2\alpha^2\mathbb{E}\left[\|\bar{\mathbf{z}}^k - \bar{\mathbf{z}}^{k-1}\|^2\right]$$

$$\leq 120m^2n^2\bar{L}^2\mathbb{E}\left[\|\mathbf{V}^k - \mathbf{1}\bar{\mathbf{v}}^k\|^2\right] + 15m^2\bar{L}^2\mathbb{E}\left[\|\bar{\mathbf{v}}^k - \bar{\mathbf{v}}^{k-1}\|^2\right] + 30m^2n^2\bar{L}^2\mathbb{E}\left[\|\mathbf{V}^{k-1} - \mathbf{1}\bar{\mathbf{v}}^{k-1}\|^2\right]$$

$$+ 135m^2n^2\bar{L}^2\mathbb{E}\left[\|\mathbf{Z}^k - \mathbf{1}\bar{\mathbf{z}}^k\|^2\right] + 15m^2n^2\bar{L}^2\alpha^2\mathbb{E}\left[\|\mathbf{Z}^{k-1} - \mathbf{1}\bar{\mathbf{z}}^{k-1}\|^2\right]$$

$$+ 90m^2n^2\bar{L}^2\eta^2\mathbb{E}\left[\|\mathbf{S}^k - \mathbf{1}\bar{\mathbf{s}}^k\|^2\right] + 15m^3n^2\bar{L}^2\mathbb{E}\left[\|\bar{\mathbf{z}}^{k+1} - \bar{\mathbf{v}}^k\|^2\right]$$

$$+ 15m^3n^2\bar{L}^2\alpha^2\mathbb{E}\left[\|\bar{\mathbf{z}}^{k+1} - \bar{\mathbf{z}}^k\|^2\right] + 15m^3n^2\bar{L}^2\mathbb{E}\left[\|\bar{\mathbf{z}}^k - \bar{\mathbf{v}}^{k-1}\|^2\right]$$

$$+ 15m^3n^2\bar{L}^2\alpha^2\mathbb{E}\left[\|\bar{\mathbf{z}}^k - \bar{\mathbf{z}}^{k-1}\|^2\right],$$

where the second inequality holds because $b, m, n \geq 1$ and $\alpha \leq 1$; the third inequality follows from equation (20); the last inequality is due to $\rho < 1$ and $\beta \in [0, 1]$.

Substituting the above results into equation (21), we have

$$\mathbb{E}\left[\|\mathbf{S}^{k+1} - \mathbf{1}\bar{\mathbf{s}}^{k+1}\|^2\right]$$

$$\leq 2\rho^2\mathbb{E}\left[\|\mathbf{S}^k - \mathbf{1}\bar{\mathbf{s}}^k\|^2\right] + 2\rho^2\mathbb{E}\left[\|\mathbf{\Delta}^{k+1} - \mathbf{\Delta}^k\|^2\right]$$

$$\leq 240m^2n^2\bar{L}^2\rho^2\mathbb{E}\left[\|\mathbf{V}^k - \mathbf{1}\bar{\mathbf{v}}^k\|^2\right] + 30m^2\bar{L}^2\rho^2\mathbb{E}\left[\|\bar{\mathbf{v}}^k - \bar{\mathbf{v}}^{k-1}\|^2\right]$$

$$+ 60m^2n^2\bar{L}^2\rho^2\mathbb{E}\left[\|\mathbf{V}^{k-1} - \mathbf{1}\bar{\mathbf{v}}^{k-1}\|^2\right] + 270m^2n^2\bar{L}^2\rho^2\mathbb{E}\left[\|\mathbf{Z}^k - \mathbf{1}\bar{\mathbf{z}}^k\|^2\right]$$

$$+ 30m^2n^2\bar{L}^2\alpha^2\rho^2\mathbb{E}\left[\|\mathbf{Z}^{k-1} - \mathbf{1}\bar{\mathbf{z}}^{k-1}\|^2\right] + (180m^2n^2\bar{L}^2\eta^2\rho^2 + 2\rho^2)\mathbb{E}\left[\|\mathbf{S}^k - \mathbf{1}\bar{\mathbf{s}}^k\|^2\right]$$

$$+ 30m^3n^2\bar{L}^2\rho^2\mathbb{E}\left[\|\bar{\mathbf{z}}^{k+1} - \bar{\mathbf{v}}^k\|^2\right] + 30m^3n^2\bar{L}^2\alpha^2\rho^2\mathbb{E}\left[\|\bar{\mathbf{z}}^{k+1} - \bar{\mathbf{z}}^k\|^2\right]$$

$$+ 30m^3n^2\bar{L}^2\rho^2\mathbb{E}\left[\|\bar{\mathbf{z}}^k - \bar{\mathbf{v}}^{k-1}\|^2\right] + 30m^3n^2\bar{L}^2\alpha^2\rho^2\mathbb{E}\left[\|\bar{\mathbf{z}}^k - \bar{\mathbf{z}}^{k-1}\|^2\right]$$

$$\leq 270m^2n^2\bar{L}^2\rho^2\Big(\mathbb{E}\left[\|\mathbf{Z}^k - \mathbf{1}\bar{\mathbf{z}}^k\|^2\right] + \mathbb{E}\left[\|\mathbf{Z}^{k-1} - \mathbf{1}\bar{\mathbf{z}}^{k-1}\|^2\right] + \mathbb{E}\left[\|\mathbf{V}^k - \mathbf{1}\bar{\mathbf{v}}^k\|^2\right]$$

$$+ \mathbb{E}\left[\|\mathbf{V}^{k-1} - \mathbf{1}\bar{\mathbf{v}}^{k-1}\|^2\right]\Big) + (180m^2n^2\bar{L}^2\eta^2 + 2)\rho^2\mathbb{E}\left[\|\mathbf{S}^k - \mathbf{1}\bar{\mathbf{s}}^k\|^2\right]$$

$$+ 60m^2\bar{L}^2\rho^2\left(\mathbb{E}\left[\|\bar{\mathbf{v}}^k - \bar{\mathbf{v}}^*\|^2\right] + \mathbb{E}\left[\|\bar{\mathbf{v}}^{k-1} - \bar{\mathbf{v}}^*\|^2\right]\right)$$

$$+ 30m^3n^2\bar{L}^2\rho^2\Big(\mathbb{E}\left[\|\bar{\mathbf{z}}^{k+1} - \bar{\mathbf{v}}^k\|^2\right] + \mathbb{E}\left[\|\bar{\mathbf{z}}^{k+1} - \bar{\mathbf{z}}^k\|^2\right] + \mathbb{E}\left[\|\bar{\mathbf{z}}^k - \bar{\mathbf{v}}^{k-1}\|^2\right]$$

$$+ \mathbb{E}\left[\|\bar{\mathbf{z}}^k - \bar{\mathbf{z}}^{k-1}\|^2\right]\Big). \tag{28}$$

With the choice of parameters $\beta = p$, it holds that

$$\Phi^k \geq \frac{\beta}{\eta}\|\bar{\mathbf{z}}^k - \bar{\mathbf{v}}^{k-1}\|^2 + \frac{1}{16\eta}\|\bar{\mathbf{z}}^k - \bar{\mathbf{z}}^{k-1}\|^2 + \frac{1}{\eta}\|\bar{\mathbf{v}}^k - \mathbf{z}^*\|^2.$$

Therefore,

$$\|\bar{\mathbf{z}}^k - \bar{\mathbf{v}}^{k-1}\|^2 + \|\bar{\mathbf{z}}^k - \bar{\mathbf{z}}^{k-1}\|^2 + \|\bar{\mathbf{v}}^k - \mathbf{z}^*\|^2 \leq \left(16\eta + \frac{\eta}{\beta}\right)\Phi^k. \tag{29}$$

Combining the results of equations (28) and (29), we obtain

$$\mathbb{E}\left[\|\mathbf{S}^{k+1} - \mathbf{1}\bar{\mathbf{s}}^{k+1}\|^2\right]$$
$$\leq C_2\rho^2\left(\mathbb{E}\left[\|\mathbf{Z}^k - \mathbf{1}\bar{\mathbf{z}}^k\|^2\right] + \mathbb{E}\left[\|\mathbf{Z}^{k-1} - \mathbf{1}\bar{\mathbf{z}}^{k-1}\|^2\right] + \mathbb{E}\left[\|\mathbf{V}^k - \mathbf{1}\bar{\mathbf{v}}^k\|^2\right] + \mathbb{E}\left[\|\mathbf{V}^{k-1} - \mathbf{1}\bar{\mathbf{v}}^{k-1}\|^2\right]\right)$$
$$+ C_3\rho^2\mathbb{E}\left[\|\mathbf{S}^k - \mathbf{1}\bar{\mathbf{s}}^k\|^2\right] + C_4\rho^2\left(\mathbb{E}\left[\Phi^{k+1}\right] + \mathbb{E}\left[\Phi^k\right] + \mathbb{E}\left[\Phi^{k-1}\right]\right),$$

where $C_2 = 270m^2n^2\bar{L}^2$, $C_3 = 180m^2n^2\bar{L}^2\eta^2 + 2$, $C_4 = 60(16\eta + \eta/\beta)m^3n^2\bar{L}^2$, and $\Phi^{-1} = 0$.

### C.4  The proof of Lemma 3.5

From the definition of $\rho$, it follows that

$$\rho^2 = 14(1 - (1 - 1/\sqrt{2})\sqrt{1 - \lambda_2(\mathbf{W})})^{2R} \leq 14\exp\left(-2(1 - 1/\sqrt{2})\sqrt{1 - \lambda_2(\mathbf{W})}R\right), \tag{30}$$

where the inequality holds by the fact that $1 - x \leq e^{-x}$.

Therefore, when

$$R = \left\lceil \frac{2 + \sqrt{2}}{2\sqrt{1 - \lambda_2(\mathbf{W})}}\log\left(14\max\left\{\frac{3C_3}{\tilde{\alpha}}, \frac{12\eta^2 C_2}{\tilde{\alpha}^2},\right.\right.\right.$$
$$\left.\left.\left.\frac{36\eta^2 C_1 C_4}{(1 - \tilde{\alpha})\tilde{\alpha}^3}, \frac{9}{\tilde{\alpha}}, \frac{p}{1 - \tilde{\alpha}}, \frac{8C_1\eta^2\left(m^2\bar{L}^2\|\mathbf{z}^0 - \mathbf{z}^*\|^2 + \sum_{i=1}^m\|\mathbf{g}_i(\mathbf{z}^*)\|^2\right)}{(1 - \tilde{\alpha})\tilde{\alpha}\Phi^0}\right\}\right)\right\rceil,$$
$$\tag{31}$$

it holds that

$$\rho^2 \leq \min\left\{\frac{\tilde{\alpha}}{3C_3}, \frac{\tilde{\alpha}^2}{12\eta^2 C_2}, \frac{(1 - \tilde{\alpha})\tilde{\alpha}^3}{36\eta^2 C_1 C_4}, \frac{\tilde{\alpha}}{9}, \frac{1 - \tilde{\alpha}}{p}, \frac{(1 - \tilde{\alpha})\tilde{\alpha}\Phi^0}{8C_1\eta^2\left(m^2\bar{L}^2\|\mathbf{z}^0 - \mathbf{z}^*\|^2 + \sum_{i=1}^m\|\mathbf{g}_i(\mathbf{z}^*)\|^2\right)}\right\}.$$
$$\tag{32}$$

We will use induction to prove that the following results hold for $k \geq 0$:

$$\mathbb{E}\left[\Phi^k\right] \leq \tilde{\alpha}^k \Phi^0,$$

$$\mathbb{E}\left[\|\mathbf{Z}^k - \mathbf{1}\bar{\mathbf{z}}^k\|^2\right] \leq \frac{1-\tilde{\alpha}}{4C_1}\tilde{\alpha}^{k+1}\Phi^0,$$

$$\mathbb{E}\left[\|\mathbf{V}^k - \mathbf{1}\bar{\mathbf{v}}^k\|^2\right] \leq \frac{1-\tilde{\alpha}}{4C_1}\tilde{\alpha}^{k+1}\Phi^0, \tag{33}$$

$$\mathbb{E}\left[\|\mathbf{S}^k - \mathbf{1}\bar{\mathbf{s}}^k\|^2\right] \leq \frac{1-\tilde{\alpha}}{4\eta^2 C_1}\tilde{\alpha}^{k+1}\Phi^0,$$

where $\tilde{\alpha} = \max\left\{1 - \frac{\mu\eta}{8}, 1 - \frac{p\eta\mu}{2(\beta+\eta\mu)}\right\}$.

For $k = 0$, since $\|\mathbf{Z}^0 - \mathbf{1}\bar{\mathbf{z}}^0\|^2 = \|\mathbf{V}^0 - \mathbf{1}\bar{\mathbf{v}}^0\|^2 = 0$, it is straightforward to verify that the first three inequalities hold. Next, we verify that $\|\mathbf{S}^0 - \mathbf{1}\bar{\mathbf{s}}^0\|^2$ also satisfies the inequality in (33). It holds that

$$\begin{aligned}
\|\mathbf{S}^0 - \mathbf{1}\bar{\mathbf{s}}^0\|^2 &= \|\texttt{FastMix}(\mathbf{\Delta}^0, \mathbf{W}, R) - \frac{1}{m}\mathbf{1}\mathbf{1}^\top\texttt{FastMix}(\mathbf{\Delta}^0, \mathbf{W}, R)\|^2 \\
&\leq \rho^2\|\mathbf{\Delta}^0 - \frac{1}{m}\mathbf{1}\mathbf{1}^\top\mathbf{\Delta}^0\|^2 \\
&\leq \rho^2\|\mathbf{\Delta}^0\|^2 \\
&= \rho^2\sum_{i=1}^m \|\mathbf{g}_i(\mathbf{z}^0)\|^2 \\
&\leq 2\rho^2\sum_{i=1}^m \|\mathbf{g}_i(\mathbf{z}^0) - \mathbf{g}_i(\mathbf{z}^*)\|^2 + 2\rho^2\sum_{i=1}^m \|\mathbf{g}_i(\mathbf{z}^*)\|^2 \\
&\leq 2\rho^2 m^2\bar{L}^2\|\mathbf{z}^0 - \mathbf{z}^*\|^2 + 2\rho^2\sum_{i=1}^m \|\mathbf{g}_i(\mathbf{z}^*)\|^2,
\end{aligned}$$

where the first inequality holds by Proposition B.2, the second inequality follows from the fact that $\|\mathbf{A} - \frac{1}{m}\mathbf{1}\mathbf{1}^\top\mathbf{A}\| \leq \|\mathbf{A}\|$ for any $\mathbf{A} \in \mathbb{R}^{m \times d_z}$, the third inequality is based on Young's inequality, and the last inequality holds due to Assumption 2.3.

From the upper bound of $\rho^2$ in equation (32), we have

$$\|\mathbf{S}^0 - \mathbf{1}\bar{\mathbf{s}}^0\|^2 \leq \frac{1-\tilde{\alpha}}{4\eta^2 C_1}\tilde{\alpha}\Phi^0.$$

For $k = 1$, according to Lemma 3.1,

$$\begin{aligned}
\mathbb{E}\left[\Phi^1\right] &\leq \max\left\{1 - \frac{\mu\eta}{4}, 1 - \frac{p\eta\mu}{\beta+\eta\mu}\right\}\Phi^0 \\
&\leq \max\left\{1 - \frac{\mu\eta}{8}, 1 - \frac{p\eta\mu}{2(\beta+\eta\mu)}\right\}\Phi^0.
\end{aligned}$$

From Lemmas 3.2 and 3.3, together with equation (32), we have

$$\mathbb{E}\left[\|\mathbf{Z}^1 - \mathbf{1}\bar{\mathbf{z}}^1\|^2\right] \leq 3\rho^2\eta^2\mathbb{E}\left[\|\mathbf{S}^0 - \mathbf{1}\bar{\mathbf{s}}^0\|^2\right] \leq \frac{1-\tilde{\alpha}}{4C_1}\tilde{\alpha}^2\Phi^0,$$

$$\mathbb{E}\left[\|\mathbf{V}^1 - \mathbf{1}\bar{\mathbf{v}}^1\|^2\right] = p\mathbb{E}\left[\|\mathbf{Z}^0 - \mathbf{1}\bar{\mathbf{z}}^0\|^2\right] + (1-p)\mathbb{E}\left[\|\mathbf{V}^0 - \mathbf{1}\bar{\mathbf{v}}^0\|^2\right] = 0,$$

and

$$\begin{aligned}
\mathbb{E}\left[\|\mathbf{S}^1 - \mathbf{1}\bar{\mathbf{s}}^1\|^2\right] &\leq C_3\rho^2\mathbb{E}\left[\|\mathbf{S}^0 - \mathbf{1}\bar{\mathbf{s}}^0\|^2\right] + C_4\rho^2\left(\mathbb{E}\left[\Phi^1\right] + \mathbb{E}\left[\Phi^0\right]\right) \\
&\leq C_3 \cdot \frac{\tilde{\alpha}}{3C_3} \cdot \frac{1-\tilde{\alpha}}{4\eta^2 C_1}\tilde{\alpha}\Phi^0 + C_4 \cdot \frac{(1-\tilde{\alpha})\tilde{\alpha}^3}{36\eta^2 C_1 C_4} \cdot (\tilde{\alpha}+1)\Phi^0 \\
&\leq \frac{1-\tilde{\alpha}}{4\eta^2 C_1}\tilde{\alpha}^2\Phi^0.
\end{aligned}$$

Assume that the conclusion of equation (33) holds for $k \leq t$. Then, for $k = t + 1$, by Lemma 3.1, we have

$$\mathbb{E}\left[\Phi^{t+1}\right]$$

$$\leq \alpha \mathbb{E}\left[\Phi^t\right] + C_1\Big(\mathbb{E}\left[\|\mathbf{Z}^t - \mathbf{1}\bar{\mathbf{z}}^t\|^2\right] + \mathbb{E}\left[\|\mathbf{Z}^{t-1} - \mathbf{1}\bar{\mathbf{z}}^{t-1}\|^2\right] + \mathbb{E}\left[\|\mathbf{V}^{t-1} - \mathbf{1}\bar{\mathbf{v}}^{t-1}\|^2\right]\Big)$$

$$\leq \alpha \cdot \tilde{\alpha}^t \Phi^0 + C_1\left(\frac{1-\tilde{\alpha}}{4C_1}\tilde{\alpha}^{t+1}\Phi^0 + \frac{1-\tilde{\alpha}}{4C_1}\tilde{\alpha}^t\Phi^0 + \frac{1-\tilde{\alpha}}{4C_1}\tilde{\alpha}^t\Phi^0\right)$$

$$\leq \max\left\{1 - \frac{\mu\eta}{4}, 1 - \frac{p\eta\mu}{\beta + \eta\mu}\right\}\left(\max\left\{1 - \frac{\mu\eta}{8}, 1 - \frac{p\eta\mu}{2(\beta + \eta\mu)}\right\}\right)^t \Phi^0$$

$$+ 3\min\left\{\frac{\mu\eta}{32}, \frac{p\eta\mu}{8(\beta + \eta\mu)}\right\}\left(\max\left\{1 - \frac{\mu\eta}{8}, 1 - \frac{p\eta\mu}{2(\beta + \eta\mu)}\right\}\right)^t \Phi^0$$

$$= \left(\max\left\{1 - \frac{\mu\eta}{4}, 1 - \frac{p\eta\mu}{\beta + \eta\mu}\right\} + \min\left\{\frac{3\mu\eta}{32}, \frac{3p\eta\mu}{8(\beta + \eta\mu)}\right\}\right)\left(\max\left\{1 - \frac{\mu\eta}{8}, 1 - \frac{p\eta\mu}{2(\beta + \eta\mu)}\right\}\right)^t \Phi^0$$

$$\leq \left(\max\left\{1 - \frac{\mu\eta}{8}, 1 - \frac{p\eta\mu}{2(\beta + \eta\mu)}\right\}\right)^{t+1} \Phi^0$$

$$= \tilde{\alpha}^{t+1}\Phi^0, \tag{34}$$

where the second inequality is derived using the inductive hypothesis.

For $\mathbb{E}\left[\|\mathbf{Z}^{t+1} - \mathbf{1}\bar{\mathbf{z}}^{t+1}\|^2\right]$, by Lemma 3.2, it holds that

$$\mathbb{E}\left[\|\mathbf{Z}^{t+1} - \mathbf{1}\bar{\mathbf{z}}^{t+1}\|^2\right] \leq 3\rho^2\left(\mathbb{E}\left[\|\mathbf{Z}^t - \mathbf{1}\bar{\mathbf{z}}^t\|^2\right] + \mathbb{E}\left[\|\mathbf{V}^t - \mathbf{1}\bar{\mathbf{v}}^t\|^2\right] + \eta^2\mathbb{E}\left[\|\mathbf{S}^t - \mathbf{1}\bar{\mathbf{s}}^t\|^2\right]\right)$$

$$\leq 3\rho^2\left(\frac{1-\tilde{\alpha}}{4C_1}\tilde{\alpha}^{t+1}\Phi^0 + \frac{1-\tilde{\alpha}}{4C_1}\tilde{\alpha}^{t+1}\Phi^0 + \eta^2 \cdot \frac{1-\tilde{\alpha}}{4\eta^2 C_1}\tilde{\alpha}^{t+1}\Phi^0\right)$$

$$\leq 3 \cdot \frac{\tilde{\alpha}}{9} \cdot 3 \cdot \frac{1-\tilde{\alpha}}{4C_1}\tilde{\alpha}^{t+1}\Phi^0$$

$$= \frac{1-\tilde{\alpha}}{4C_1}\tilde{\alpha}^{t+2}\Phi^0,$$

where the second inequality holds by the inductive hypothesis, and the third inequality follows equation (32).

For $\mathbb{E}\left[\|\mathbf{V}^{t+1} - \mathbf{1}\bar{\mathbf{v}}^{t+1}\|^2\right]$, note that

$$1 - p \leq 1 - \frac{p\eta\mu}{\beta + \eta\mu} \leq \max\left\{1 - \frac{\mu\eta}{4}, 1 - \frac{p\eta\mu}{\beta + \eta\mu}\right\} = \alpha. \tag{35}$$

By Lemma 3.2, we obtain

$$\mathbb{E}\left[\|\mathbf{Z}^{t+1} - \mathbf{1}\bar{\mathbf{z}}^{t+1}\|^2\right] \leq p\rho^2\mathbb{E}\left[\|\mathbf{Z}^k - \mathbf{1}\bar{\mathbf{z}}^k\|^2\right] + (1-p)\mathbb{E}\left[\|\mathbf{V}^k - \mathbf{1}\bar{\mathbf{v}}^k\|^2\right]$$

$$\leq p\rho^2\frac{1-\tilde{\alpha}}{4C_1}\tilde{\alpha}^{t+1}\Phi^0 + (1-p)\frac{1-\tilde{\alpha}}{4C_1}\tilde{\alpha}^{t+1}\Phi^0$$

$$\leq p \cdot \frac{1-\tilde{\alpha}}{p} \cdot \frac{1-\tilde{\alpha}}{4C_1}\tilde{\alpha}^{t+1}\Phi^0 + \alpha \cdot \frac{1-\tilde{\alpha}}{4C_1}\tilde{\alpha}^{t+1}\Phi^0$$

$$= (1 - \tilde{\alpha} + \alpha)\frac{1-\tilde{\alpha}}{4C_1}\tilde{\alpha}^{t+1}\Phi^0$$

$$= \frac{1-\tilde{\alpha}}{4C_1}\tilde{\alpha}^{t+2}\Phi^0,$$

where the second inequality holds by the inductive hypothesis; the third inequality follows equation (32) and equation (35); the last equality is based on the fact that $1 - \tilde{\alpha} + \alpha = \tilde{\alpha}$.

For $\mathbb{E}\left[\|\mathbf{S}^{t+1} - \mathbf{1}\bar{\mathbf{s}}^{t+1}\|^2\right]$, by Lemma 3.3, it follows that

$$\mathbb{E}\left[\|\mathbf{S}^{t+1} - \mathbf{1}\bar{\mathbf{s}}^{t+1}\|^2\right]$$

$$\leq C_2\rho^2\Big(\mathbb{E}\left[\|\mathbf{Z}^t - \mathbf{1}\bar{\mathbf{z}}^t\|^2\right] + \mathbb{E}\left[\|\mathbf{Z}^{t-1} - \mathbf{1}\bar{\mathbf{z}}^{t-1}\|^2\right] + \mathbb{E}\left[\|\mathbf{V}^t - \mathbf{1}\bar{\mathbf{v}}^t\|^2\right] + \mathbb{E}\left[\|\mathbf{V}^{t-1} - \mathbf{1}\bar{\mathbf{v}}^{t-1}\|^2\right]\Big)$$

$$+ C_3\rho^2\mathbb{E}\left[\|\mathbf{S}^t - \mathbf{1}\bar{\mathbf{s}}^t\|^2\right] + C_4\rho^2\left(\mathbb{E}\left[\Phi^{t+1}\right] + \mathbb{E}\left[\Phi^t\right] + \mathbb{E}\left[\Phi^{t-1}\right]\right)$$

$$\leq C_2\rho^2 \cdot 4 \cdot \frac{1-\tilde{\alpha}}{4C_1}\tilde{\alpha}^t\Phi^0 + C_3\rho^2 \cdot \frac{1-\tilde{\alpha}}{4\eta^2 C_1}\tilde{\alpha}^{t+1}\Phi^0 + C_4\rho^2 \cdot 3 \cdot \tilde{\alpha}^{t-1}\Phi^0$$

$$\leq \frac{\tilde{\alpha}^2}{12\eta^2 C_2} \cdot 4C_2 \cdot \frac{1-\tilde{\alpha}}{4C_1}\tilde{\alpha}^t\Phi^0 + \frac{\tilde{\alpha}}{3C_3} \cdot C_3 \cdot \frac{1-\tilde{\alpha}}{4\eta^2 C_1}\tilde{\alpha}^{t+1}\Phi^0 + \frac{(1-\tilde{\alpha})\tilde{\alpha}^3}{36\eta^2 C_1 C_4} \cdot 3C_4 \cdot \tilde{\alpha}^{t-1}\Phi^0$$

$$= \frac{1-\tilde{\alpha}}{4\eta^2 C_1}\tilde{\alpha}^{t+2}\Phi^0,$$

where the second inequality follows from the inductive hypothesis and equation (34), while the third inequality holds due to equation (32).

This completes the proof for the case $k = t + 1$. The conclusion of Lemma 3.5 is established by induction.

### C.5  The proof of Lemma 3.6

Under the parameter settings of Lemma 3.6, it follows from equation (8) that

$$\|\bar{\mathbf{z}}^{k+1} - \mathbf{z}^*\|^2 = \|\bar{\mathbf{z}}^k - \mathbf{z}^*\|^2 - \|\bar{\mathbf{z}}^{k+1} - \bar{\mathbf{z}}^k\|^2 - 2\eta\langle\bar{\boldsymbol{\delta}}^k - \mathbf{g}(\mathbf{z}^*), \bar{\mathbf{z}}^{k+1} - \mathbf{z}^*\rangle.$$

Note that $\bar{\boldsymbol{\delta}}^k = \mathbb{E}_{\xi^k}\left[\boldsymbol{\delta}^k\right]$, by equation (16), we have

$$\langle\bar{\boldsymbol{\delta}}^k - \mathbf{g}(\mathbf{z}^*), \bar{\mathbf{z}}^{k+1} - \mathbf{z}^*\rangle$$

$$\geq -\frac{3n\bar{L}^2}{\mu}(1+\alpha^2)\|\mathbf{Z}^k - \mathbf{1}\bar{\mathbf{z}}^k\|^2 - \frac{3n\bar{L}^2\alpha^2}{\mu}\|\mathbf{Z}^{k-1} - \mathbf{1}\bar{\mathbf{z}}^{k-1}\|^2$$

$$+ \langle\mathbf{g}(\bar{\mathbf{z}}^k) - \mathbf{g}(\bar{\mathbf{z}}^{k+1}), \bar{\mathbf{z}}^{k+1} - \mathbf{z}^*\rangle - 2\eta\alpha^2 L^2\|\bar{\mathbf{z}}^k - \bar{\mathbf{z}}^{k-1}\|^2 - \frac{1}{8\eta}\|\bar{\mathbf{z}}^{k+1} - \bar{\mathbf{z}}^k\|^2$$

$$- \alpha\langle\mathbf{g}(\bar{\mathbf{z}}^{k-1}) - \mathbf{g}(\bar{\mathbf{z}}^k), \bar{\mathbf{z}}^k - \mathbf{z}^*\rangle + \frac{3\mu}{4}\|\bar{\mathbf{z}}^{k+1} - \mathbf{z}^*\|^2.$$

Therefore, it holds that

$$\|\bar{\mathbf{z}}^{k+1} - \mathbf{z}^*\|^2 \leq \|\bar{\mathbf{z}}^k - \mathbf{z}^*\|^2 - \|\bar{\mathbf{z}}^{k+1} - \bar{\mathbf{z}}^k\|^2 + \frac{6\eta n\bar{L}^2}{\mu}(1+\alpha^2)\|\mathbf{Z}^k - \mathbf{1}\bar{\mathbf{z}}^k\|^2$$

$$+ \frac{6\eta n\bar{L}^2\alpha^2}{\mu}\|\mathbf{Z}^{k-1} - \mathbf{1}\bar{\mathbf{z}}^{k-1}\|^2 - 2\eta\langle\mathbf{g}(\bar{\mathbf{z}}^k) - \mathbf{g}(\bar{\mathbf{z}}^{k+1}), \bar{\mathbf{z}}^{k+1} - \mathbf{z}^*\rangle$$

$$+ 2\eta\alpha\langle\mathbf{g}(\bar{\mathbf{z}}^{k-1}) - \mathbf{g}(\bar{\mathbf{z}}^k), \bar{\mathbf{z}}^k - \mathbf{z}^*\rangle + 4\eta^2\alpha^2 L^2\|\bar{\mathbf{z}}^k - \bar{\mathbf{z}}^{k-1}\|^2$$

$$+ \frac{1}{4}\|\bar{\mathbf{z}}^{k+1} - \bar{\mathbf{z}}^k\|^2 - \frac{3}{2}\mu\eta\|\bar{\mathbf{z}}^{k+1} - \mathbf{z}^*\|^2.$$

Rearranging the above equation, we obtain

$$\left(\frac{1}{\eta} + \frac{3\mu}{2}\right)\|\bar{\mathbf{z}}^{k+1} - \mathbf{z}^*\|^2 + \frac{3}{4\eta}\|\bar{\mathbf{z}}^{k+1} - \bar{\mathbf{z}}^k\|^2 + 2\langle\mathbf{g}(\bar{\mathbf{z}}^k) - \mathbf{g}(\bar{\mathbf{z}}^{k+1}), \bar{\mathbf{z}}^{k+1} - \mathbf{z}^*\rangle$$

$$\leq \frac{1}{\eta}\|\bar{\mathbf{z}}^k - \mathbf{z}^*\|^2 + 4\eta\alpha^2 L^2\|\bar{\mathbf{z}}^k - \bar{\mathbf{z}}^{k-1}\|^2 + 2\alpha\langle\mathbf{g}(\bar{\mathbf{z}}^{k-1}) - \mathbf{g}(\bar{\mathbf{z}}^k), \bar{\mathbf{z}}^k - \mathbf{z}^*\rangle$$

$$+ \frac{6n\bar{L}^2}{\mu}(1+\alpha^2)\|\mathbf{Z}^k - \mathbf{1}\bar{\mathbf{z}}^k\|^2 + \frac{6n\bar{L}^2\alpha^2}{\mu}\|\mathbf{Z}^{k-1} - \mathbf{1}\bar{\mathbf{z}}^{k-1}\|^2.$$

Based on the parameter settings, we have

$$\frac{1}{\eta} \leq (1 - \mu\eta)\left(\frac{1}{\eta} + \frac{3\mu}{2}\right) \quad \text{and} \quad 4\eta\alpha^2 L^2 \leq \frac{3}{4\eta}\alpha.$$

Thus, the following inequality holds:

$$\left(\frac{1}{\eta} + \frac{3\mu}{2}\right)\|\bar{\mathbf{z}}^{k+1} - \mathbf{z}^*\|^2 + \frac{3}{4\eta}\|\bar{\mathbf{z}}^{k+1} - \bar{\mathbf{z}}^k\|^2 + 2\langle\mathbf{g}(\bar{\mathbf{z}}^k) - \mathbf{g}(\bar{\mathbf{z}}^{k+1}), \bar{\mathbf{z}}^{k+1} - \mathbf{z}^*\rangle$$

$$\leq (1 - \mu\eta) \left( \frac{1}{\eta} + \frac{3\mu}{2} \right) \|\bar{\mathbf{z}}^k - \mathbf{z}^*\|^2 + \alpha\frac{3}{4\eta}\|\bar{\mathbf{z}}^k - \bar{\mathbf{z}}^{k-1}\|^2 + 2\alpha\langle \mathbf{g}(\bar{\mathbf{z}}^{k-1}) - \mathbf{g}(\bar{\mathbf{z}}^k), \bar{\mathbf{z}}^k - \mathbf{z}^* \rangle$$
$$+ \frac{12n\bar{L}^2}{\mu} \left( \|\mathbf{Z}^k - \mathbf{1}\bar{\mathbf{z}}^k\|^2 + \|\mathbf{Z}^{k-1} - \mathbf{1}\bar{\mathbf{z}}^{k-1}\|^2 \right),$$

that is,

$$\Psi^{k+1} \leq \alpha\Psi^k + \frac{12n\bar{L}^2}{\mu} \left( \|\mathbf{Z}^k - \mathbf{1}\bar{\mathbf{z}}^k\|^2 + \|\mathbf{Z}^{k-1} - \mathbf{1}\bar{\mathbf{z}}^{k-1}\|^2 \right). \tag{36}$$

Next, we bound the consensus errors. For $\|\mathbf{Z}^{k+1} - \mathbf{1}\bar{\mathbf{z}}^{k+1}\|^2$, similar to equation (20), we have

$$\|\mathbf{Z}^{k+1} - \mathbf{1}\bar{\mathbf{z}}^{k+1}\|^2 \leq 2\rho^2\|\mathbf{Z}^k - \mathbf{1}\bar{\mathbf{z}}^k\|^2 + 2\rho^2\eta^2\|\mathbf{S}^k - \mathbf{1}\bar{\mathbf{s}}^k\|^2. \tag{37}$$

For $\|\mathbf{S}^{k+1} - \mathbf{1}\bar{\mathbf{s}}^{k+1}\|^2$, according to equation (21), we have

$$\|\mathbf{S}^{k+1} - \mathbf{1}\bar{\mathbf{s}}^{k+1}\|^2 \leq 2\rho^2\|\mathbf{S}^k - \mathbf{1}\bar{\mathbf{s}}^k\|^2 + 2\rho^2\|\boldsymbol{\Delta}^{k+1} - \boldsymbol{\Delta}^k\|^2.$$

Note that

$$\left\| \mathbf{g}_i(\mathbf{z}_i^{k+1}) - \alpha \left( \mathbf{g}_i(\mathbf{z}_i^{k+1}) - \mathbf{g}_i(\mathbf{z}_i^k) \right) - \left( \mathbf{g}_i(\mathbf{z}_i^k) - \alpha \left( \mathbf{g}_i(\mathbf{z}_i^k) - \mathbf{g}_i(\mathbf{z}_i^{k-1}) \right) \right) \right\|^2$$
$$\leq 3\|\mathbf{g}_i(\mathbf{z}_i^{k+1}) - \mathbf{g}_i(\mathbf{z}_i^k)\|^2 + 3\alpha^2\|\mathbf{g}_i(\mathbf{z}_i^{k+1}) - \mathbf{g}_i(\mathbf{z}_i^k)\|^2 + 3\alpha^2\|\mathbf{g}_i(\mathbf{z}_i^k) - \mathbf{g}_i(\mathbf{z}_i^{k-1})\|^2$$
$$\leq 6m\bar{L}^2\|\mathbf{z}_i^{k+1} - \mathbf{z}_i^k\|^2 + 3m\bar{L}^2\|\mathbf{z}_i^k - \mathbf{z}_i^{k-1}\|^2$$
$$\leq 18m\bar{L}^2\|\mathbf{z}_i^{k+1} - \bar{\mathbf{z}}^{k+1}\|^2 + 18m\bar{L}^2\|\mathbf{z}_i^k - \bar{\mathbf{z}}^k\|^2 + 18m\bar{L}^2\|\bar{\mathbf{z}}^{k+1} - \bar{\mathbf{z}}^k\|^2$$
$$+ 9m\bar{L}^2\|\mathbf{z}_i^k - \bar{\mathbf{z}}^k\|^2 + 9m\bar{L}^2\|\mathbf{z}_i^{k-1} - \bar{\mathbf{z}}^{k-1}\|^2 + 9m\bar{L}^2\|\bar{\mathbf{z}}^k - \bar{\mathbf{z}}^{k-1}\|^2,$$

where the first and third inequalities hold due to Young's inequality, while the second inequality follows from equation (6).

The term $\|\boldsymbol{\Delta}^{k+1} - \boldsymbol{\Delta}^k\|^2$ can be upper bounded by

$$\|\boldsymbol{\Delta}^{k+1} - \boldsymbol{\Delta}^k\|^2$$
$$= \sum_{i=1}^m \left\| \mathbf{g}_i(\mathbf{z}_i^{k+1}) - \alpha \left( \mathbf{g}_i(\mathbf{z}_i^{k+1}) - \mathbf{g}_i(\mathbf{z}_i^k) \right) - \left( \mathbf{g}_i(\mathbf{z}_i^k) - \alpha \left( \mathbf{g}_i(\mathbf{z}_i^k) - \mathbf{g}_i(\mathbf{z}_i^{k-1}) \right) \right) \right\|^2$$
$$\leq 18m\bar{L}^2\|\mathbf{Z}^{k+1} - \mathbf{1}\bar{\mathbf{z}}^{k+1}\|^2 + 18m\bar{L}^2\|\mathbf{Z}^k - \mathbf{1}\bar{\mathbf{z}}^k\|^2 + 18m^2\bar{L}^2\|\bar{\mathbf{z}}^{k+1} - \bar{\mathbf{z}}^k\|^2$$
$$+ 9m\bar{L}^2\|\mathbf{Z}^k - \mathbf{1}\bar{\mathbf{z}}^k\|^2 + 9m\bar{L}^2\|\mathbf{Z}^{k-1} - \mathbf{1}\bar{\mathbf{z}}^{k-1}\|^2 + 9m^2\bar{L}^2\|\bar{\mathbf{z}}^k - \bar{\mathbf{z}}^{k-1}\|^2$$
$$\leq 18m\bar{L}^2 \left( 2\rho^2\|\mathbf{Z}^k - \mathbf{1}\bar{\mathbf{z}}^k\|^2 + 2\rho^2\eta^2\|\mathbf{S}^k - \mathbf{1}\bar{\mathbf{s}}^k\|^2 \right) + 27m\bar{L}^2\|\mathbf{Z}^k - \mathbf{1}\bar{\mathbf{z}}^k\|^2$$
$$+ 9m\bar{L}^2\|\mathbf{Z}^{k-1} - \mathbf{1}\bar{\mathbf{z}}^{k-1}\|^2 + 18m^2\bar{L}^2\|\bar{\mathbf{z}}^{k+1} - \bar{\mathbf{z}}^k\|^2 + 9m^2\bar{L}^2\|\bar{\mathbf{z}}^k - \bar{\mathbf{z}}^{k-1}\|^2$$
$$\leq 63m\bar{L}^2\|\mathbf{Z}^k - \mathbf{1}\bar{\mathbf{z}}^k\|^2 + 9m\bar{L}^2\|\mathbf{Z}^{k-1} - \mathbf{1}\bar{\mathbf{z}}^{k-1}\|^2 + 36\eta^2m\bar{L}^2\|\mathbf{S}^k - \mathbf{1}\bar{\mathbf{s}}^k\|^2$$
$$+ 18m^2\bar{L}^2\|\bar{\mathbf{z}}^{k+1} - \bar{\mathbf{z}}^k\|^2 + 9m^2\bar{L}^2\|\bar{\mathbf{z}}^k - \bar{\mathbf{z}}^{k-1}\|^2,$$

where the second inequality is based on equation (37). Recalling the definition of $\Psi$, it holds that $\Psi^k \geq 1/(2\eta)\|\bar{\mathbf{z}}^k - \bar{\mathbf{z}}^{k-1}\|^2$.

Thus, the consensus error $\|\mathbf{S}^{k+1} - \mathbf{1}\bar{\mathbf{s}}^{k+1}\|^2$ has the following bound:

$$\|\mathbf{S}^{k+1} - \mathbf{1}\bar{\mathbf{s}}^{k+1}\|^2 \leq 2\rho^2\|\mathbf{S}^k - \mathbf{1}\bar{\mathbf{s}}^k\|^2 + 2\rho^2\|\boldsymbol{\Delta}^{k+1} - \boldsymbol{\Delta}^k\|^2$$
$$\leq 126m\bar{L}^2\rho^2 \left( \|\mathbf{Z}^k - \mathbf{1}\bar{\mathbf{z}}^k\|^2 + \|\mathbf{Z}^{k-1} - \mathbf{1}\bar{\mathbf{z}}^{k-1}\|^2 \right) \tag{38}$$
$$+ \left( 72\eta^2 m\bar{L}^2 + 2 \right) \rho^2\|\mathbf{S}^k - \mathbf{1}\bar{\mathbf{s}}^k\|^2 + 72\eta m^2\bar{L}^2\rho^2 \left( \Psi^{k+1} + \Psi^k \right).$$

Similar to Lemma 3.5, we proceed to prove the conclusion of Lemma 3.6 using induction. Letting

$$R = \left\lceil \frac{2+\sqrt{2}}{2\sqrt{1-\lambda_2(\mathbf{W})}} \log\left( 14\max\left\{ \frac{3\left(72\eta^2 m\bar{L}^2 + 2\right)}{1 - \frac{\mu\eta}{2}}, \frac{20736\eta^2 m^2 n\bar{L}^4}{\mu^2\left(1 - \frac{\mu\eta}{2}\right)^2}, \frac{4}{1 - \frac{\mu\eta}{2}}, \right.\right.\right.$$

$$\left.\left.\left. \frac{756\eta^2 m\bar{L}^2}{\left(1 - \frac{\mu\eta}{2}\right)^2}, \frac{96\eta n\bar{L}^2\left(m^2\bar{L}^2\|\mathbf{z}^0 - \mathbf{z}^*\|^2 + \sum_{i=1}^{m}\|\mathbf{g}_i(\mathbf{z}^*)\|^2\right)}{\mu^2\left(1 - \frac{\mu\eta}{2}\right)\Psi^0} \right\}\right)\right\rceil,$$

(39)

it holds that

$$\rho^2 \leq \min\left\{ \frac{1 - \frac{\mu\eta}{2}}{3\left(72\eta^2 m\bar{L}^2 + 2\right)}, \frac{\mu^2\left(1 - \frac{\mu\eta}{2}\right)^2}{20736\eta^2 m^2 n\bar{L}^4}, \frac{1 - \frac{\mu\eta}{2}}{4}, \right.$$

$$\left. \frac{\left(1 - \frac{\mu\eta}{2}\right)^2}{756\eta^2 m\bar{L}^2}, \frac{\mu^2\left(1 - \frac{\mu\eta}{2}\right)\Psi^0}{96\eta n\bar{L}^2\left(m^2\bar{L}^2\|\mathbf{z}^0 - \mathbf{z}^*\|^2 + \sum_{i=1}^{m}\|\mathbf{g}_i(\mathbf{z}^*)\|^2\right)} \right\}.$$

(40)

For $k = 0$, we have

$$\|\mathbf{S}^0 - \mathbf{1}\bar{\mathbf{s}}^0\|^2 \leq \rho^2\|\boldsymbol{\Delta}^0\|^2 \leq 2\rho^2 m^2\bar{L}^2\|\mathbf{z}^0 - \mathbf{z}^*\|^2 + 2\rho^2\sum_{i=1}^{m}\|\mathbf{g}_i(\mathbf{z}^*)\|^2 \leq \frac{\mu^2}{48\eta n\bar{L}^2}\left(1 - \frac{\mu\eta}{2}\right)\Psi^0.$$

For $k = 1$, we have

$$\Psi^1 \leq (1 - \mu\eta)\Psi^0 \leq \left(1 - \frac{\mu\eta}{2}\right)\Psi^0,$$

$$\|\mathbf{Z}^1 - \mathbf{1}\bar{\mathbf{z}}^1\|^2 \leq 2\rho^2\eta^2\|\mathbf{S}^0 - \mathbf{1}\bar{\mathbf{s}}^0\|^2 \leq \frac{\mu^2\eta}{48n\bar{L}^2}\left(1 - \frac{\mu\eta}{2}\right)^2\Psi^0,$$

and

$$\|\mathbf{S}^1 - \mathbf{1}\bar{\mathbf{s}}^1\|^2 \leq \left(72\eta^2 m\bar{L}^2 + 2\right)\rho^2\|\mathbf{S}^0 - \mathbf{1}\bar{\mathbf{s}}^0\|^2 + 72\eta m^2\bar{L}^2\rho^2\left(\Psi^1 + \Psi^0\right)$$

$$\leq \left(72\eta^2 m\bar{L}^2 + 2\right)\cdot\frac{\left(1 - \frac{\mu\eta}{2}\right)}{3\left(72\eta^2 m\bar{L}^2 + 2\right)}\cdot\frac{\mu^2}{48\eta n\bar{L}^2}\left(1 - \frac{\mu\eta}{2}\right)\Psi^0$$

$$+ 72\eta m^2\bar{L}^2\cdot\frac{\mu^2\left(1 - \frac{\mu\eta}{2}\right)^2}{20736\eta^2 m^2 n\bar{L}^4}\cdot 2\Psi^0$$

$$\leq \frac{\mu^2}{48\eta n\bar{L}^2}\left(1 - \frac{\mu\eta}{2}\right)^2\Psi^0.$$

Assume that

$$\Psi^k \leq \left(1 - \frac{\mu\eta}{2}\right)^k\Psi^0,$$

$$\|\mathbf{Z}^k - \mathbf{1}\bar{\mathbf{z}}^k\|^2 \leq \frac{\mu^2\eta}{48n\bar{L}^2}\left(1 - \frac{\mu\eta}{2}\right)^{k+1}\Psi^0,$$

$$\|\mathbf{S}^k - \mathbf{1}\bar{\mathbf{s}}^k\|^2 \leq \frac{\mu^2}{48\eta n\bar{L}^2}\left(1 - \frac{\mu\eta}{2}\right)^{k+1}\Psi^0,$$

holds for all $k \leq t$. Then for $k = t + 1$, we have

$$\Psi^{t+1} \leq (1 - \mu\eta)\Psi^t + \frac{12n\bar{L}^2}{\mu}\left(\|\mathbf{Z}^t - \mathbf{1}\bar{\mathbf{z}}^t\|^2 + \|\mathbf{Z}^{t-1} - \mathbf{1}\bar{\mathbf{z}}^{t-1}\|^2\right)$$

$$\leq (1 - \mu\eta)\left(1 - \frac{\mu\eta}{2}\right)^t\Psi^0 + \frac{12n\bar{L}^2}{\mu}\cdot 2\cdot\frac{\mu^2\eta}{48n\bar{L}^2}\left(1 - \frac{\mu\eta}{2}\right)^t\Psi^0$$

$$= \left(1 - \frac{\mu\eta}{2}\right)^{t+1}\Psi^0,$$

where the first inequality is based on equation (36) and the second inequality is due to the induction hypothesis.

For $\|\mathbf{Z}^k - \mathbf{1}\bar{\mathbf{z}}^k\|^2$, it holds that

$$
\begin{aligned}
&\|\mathbf{Z}^{k+1} - \mathbf{1}\bar{\mathbf{z}}^{k+1}\|^2 \\
&\leq 2\rho^2\|\mathbf{Z}^k - \mathbf{1}\bar{\mathbf{z}}^k\|^2 + 2\rho^2\eta^2\|\mathbf{S}^k - \mathbf{1}\bar{\mathbf{s}}^k\|^2 \\
&\leq 2 \cdot \frac{\left(1 - \frac{\mu\eta}{2}\right)}{4} \cdot \frac{\mu^2\eta}{48n\bar{L}^2}\left(1 - \frac{\mu\eta}{2}\right)^{t+1}\Psi^0 + 2 \cdot \frac{\left(1 - \frac{\mu\eta}{2}\right)}{4} \cdot \eta^2 \cdot \frac{\mu^2}{48\eta n\bar{L}^2}\left(1 - \frac{\mu\eta}{2}\right)^{t+1}\Psi^0 \\
&= \frac{\mu^2\eta}{48n\bar{L}^2}\left(1 - \frac{\mu\eta}{2}\right)^{t+2}\Psi^0,
\end{aligned}
$$

where the first inequality is based on equation (37) and the second inequality is due to the induction hypothesis and equation (40).

For $\|\mathbf{S}^{t+1} - \mathbf{1}\bar{\mathbf{s}}^{t+1}\|^2$, it holds that

$$
\begin{aligned}
&\|\mathbf{S}^{t+1} - \mathbf{1}\bar{\mathbf{s}}^{t+1}\|^2 \\
&\leq 126m\bar{L}^2\rho^2\left(\|\mathbf{Z}^k - \mathbf{1}\bar{\mathbf{z}}^k\|^2 + \|\mathbf{Z}^{k-1} - \mathbf{1}\bar{\mathbf{z}}^{k-1}\|^2\right) \\
&\quad + \left(72\eta^2 m\bar{L}^2 + 2\right)\rho^2\|\mathbf{S}^k - \mathbf{1}\bar{\mathbf{s}}^k\|^2 + 72\eta m^2\bar{L}^2\rho^2\left(\Psi^{k+1} + \Psi^k\right) \\
&\leq 126m\bar{L}^2 \cdot \frac{\left(1 - \frac{\mu\eta}{2}\right)^2}{756\eta^2 m\bar{L}^2} \cdot 2 \cdot \frac{\mu^2\eta}{48n\bar{L}^2}\left(1 - \frac{\mu\eta}{2}\right)^t\Psi^0 \\
&\quad + \left(72\eta^2 m\bar{L}^2 + 2\right) \cdot \frac{\left(1 - \frac{\mu\eta}{2}\right)}{3\left(72\eta^2 m\bar{L}^2 + 2\right)} \cdot \frac{\mu^2}{48\eta n\bar{L}^2}\left(1 - \frac{\mu\eta}{2}\right)^{t+1}\Psi^0 \\
&\quad + 72\eta m^2\bar{L}^2 \cdot \frac{\mu^2\left(1 - \frac{\mu\eta}{2}\right)^2}{20736\eta^2 m^2 n\bar{L}^4} \cdot 2 \cdot \left(1 - \frac{\mu\eta}{2}\right)^t\Psi^0 \\
&= \frac{\mu^2}{48\eta n\bar{L}^2}\left(1 - \frac{\mu\eta}{2}\right)^{t+2}\Psi^0,
\end{aligned}
$$

where the first inequality is based on equation (38) and the second inequality is due to the induction hypothesis and equation (40).

Thus, the conclusion of Lemma 3.6 holds by induction.

## C.6 The proof of Theorem 3.7

In this section, the upper bounds in Theorem 3.7 are proved based on the linear convergence of the Lyapunov functions and the consensus errors established in Lemmas 3.5 and 3.6.

**Case I:** $\bar{L} \leq \sqrt{mn}L$.

By Young's inequality, for any $i \in [m]$, it holds that

$$\|\mathbf{z}_i^K - \mathbf{z}^*\|^2 \leq 2\|\mathbf{z}_i^K - \bar{\mathbf{z}}^K\|^2 + 2\|\bar{\mathbf{z}}^K - \mathbf{z}^*\|^2.$$

According to Lemma 3.5, we have

$$
\begin{aligned}
\frac{1}{2\eta}\mathbb{E}\left[\|\bar{\mathbf{z}}^K - \mathbf{z}^*\|^2\right] &\leq \mathbb{E}\left[\Phi^K\right] \\
&\leq \left(\max\left\{1 - \frac{\mu\eta}{8}, 1 - \frac{p\eta\mu}{2(\beta + \eta\mu)}\right\}\right)^K\left(\frac{1}{\eta} + \frac{3\mu}{2} + \frac{\beta + \eta\mu}{p\eta}\right)\|\bar{\mathbf{z}}^0 - \mathbf{z}^*\|^2,
\end{aligned}
$$

and

$$
\begin{aligned}
\mathbb{E}\left[\|\mathbf{z}_i^K - \bar{\mathbf{z}}^K\|^2\right] &\leq \mathbb{E}\left[\|\mathbf{Z}^K - \mathbf{1}\bar{\mathbf{z}}^K\|^2\right] \\
&\leq \frac{1 - \tilde{\alpha}}{4C_1}\tilde{\alpha}^{K+1}\Phi^0 \\
&\leq \frac{2\eta\Phi^0}{3}\left(\max\left\{1 - \frac{\mu\eta}{8}, 1 - \frac{p\eta\mu}{2(\beta + \eta\mu)}\right\}\right)^K,
\end{aligned}
$$

where the last inequality holds by $C_1 \geq 3/(8\eta)$.

Therefore, for any $i \in [m]$,

$$\mathbb{E}\left[\|\mathbf{z}_i^K - \mathbf{z}^*\|^2\right] = \mathcal{O}\left(\left(\max\left\{1 - \frac{\mu\eta}{8}, 1 - \frac{p\eta\mu}{2(\beta + \eta\mu)}\right\}\right)^K\right).$$

By the fact that $1 - x \leq e^{-x}$, the number of iterations for $\mathbb{E}\left[\|\mathbf{z}_i^K - \mathbf{z}^*\|^2\right] \leq \epsilon$ is

$$\begin{aligned}
K &= \mathcal{O}\left(\left(\frac{1}{\mu\eta} + \frac{\beta + \mu\eta}{p\mu\eta}\right)\log\left(\frac{1}{\epsilon}\right)\right) \\
&= \mathcal{O}\left(\left(\frac{1}{\mu\eta} + \frac{1}{p}\right)\log\left(\frac{1}{\epsilon}\right)\right) \\
&= \mathcal{O}\left(\frac{L}{\mu}\log\left(\frac{1}{\epsilon}\right)\right).
\end{aligned} \tag{41}$$

**LIFO Calls:** Under the parameter settings of Lemma 3.1, the expected LIFO calls have the upper bound:

$$\begin{aligned}
&\mathcal{O}\left(mn + (pmn + (1 - p)b)K\right) \\
&= \mathcal{O}\left(mn + (pmn + b)K\right) \\
&= \mathcal{O}\left(mn + \left(mn\frac{\bar{L}}{L}\max\left\{\frac{\mu}{\bar{L}}, \frac{1}{\sqrt{mn}}\right\} + \left\lceil\frac{\bar{L}}{L}\min\left\{\frac{\bar{L}}{\mu}, \sqrt{mn}\right\}\right\rceil\right)\frac{L}{\mu}\log\left(\frac{1}{\epsilon}\right)\right) \\
&= \mathcal{O}\left(\left(mn + \sqrt{mn}\frac{\bar{L}}{\mu}\right)\log\left(\frac{1}{\epsilon}\right)\right).
\end{aligned} \tag{42}$$

**Computation Rounds:** In each iteration, the number of computation rounds depends on the node with the maximum computation rounds, that is, $\max_{i \in [m]} \sum_{j=1}^n \xi_{i,j}^k$. Therefore, we first bound $\mathbb{E}[\max_{i \in [m]} \sum_{j=1}^n \xi_{i,j}^k]$.

**Lemma C.1.** *Let $Y_i = \sum_{j=1}^n \xi_{ij}$, where $\xi_{ij} \sim$ Bernoulli$(q)$ are i.i.d. random variables, so that $Y_i \sim$ Binomial$(n, q)$. It holds that*

$$\mathbb{E}\left[\max_{i \in [m]} Y_i\right] \leq 2nq + \log m.$$

*Proof.* Firstly, we show that $Y_i$ is locally sub-Gaussian [15]. For all $t \in [-1, 1]$, the moment generating function of $Y_i$ satisfies

$$\begin{aligned}
\mathbb{E}\left[e^{tY_i}\right] &= \mathbb{E}\left[e^{t\sum_{j=1}^n \xi_{ij}}\right] \\
&= \left((1 - q) + qe^t\right)^n \\
&\leq \left((1 - q) + q(1 + t + t^2)\right)^n \\
&= \left(1 + qt + qt^2\right)^n \\
&\leq e^{nqt + nqt^2}.
\end{aligned}$$

where the first inequality holds due to $e^x \leq 1 + x + x^2, x \in [-1, 1]$, and the second inequality follows $1 + x \leq e^x$.

Then we have

$$\exp\left(\mathbb{E}\left[t\max_{i \in [m]} Y_i\right]\right) \leq \mathbb{E}\left[\exp\left(t\max_{i \in [m]} Y_i\right)\right] = \mathbb{E}\left[\max_{i \in [m]} e^{tY_i}\right] \leq \sum_{i=1}^m \mathbb{E}\left[e^{tY_i}\right] \leq me^{nqt + nqt^2},$$

where the first inequality is based on Jensen's inequality.

Letting $t = 1$ and taking the logarithm of the inequality, we have

$$\mathbb{E}\left[\max_{i \in [m]} Y_i\right] \leq 2nq + \log m.$$

This completes the proof. $\qquad\square$

We now derive the upper bound for the expected computation rounds:

$$\mathcal{O}\left(n + \left(pn + (1-p)\mathbb{E}\left[\max_{i \in [m]} \sum_{j=1}^{n} \xi_{i,j}^{k}\right]\right)K\right)$$

$$= \mathcal{O}\left(n + (pn + 2nq + \log m)K\right)$$

$$= \mathcal{O}\left(n + \left(pn + \frac{b}{m} + \log m\right)K\right)$$

$$= \mathcal{O}\left(n + \left(n\frac{\bar{L}}{L}\max\left\{\frac{\mu}{\bar{L}}, \frac{1}{\sqrt{mn}}\right\} + \frac{1}{m}\left\lceil\frac{\bar{L}}{L}\min\left\{\frac{\bar{L}}{\mu}, \sqrt{mn}\right\}\right\rceil + \log m\right)\frac{L}{\mu}\log\left(\frac{1}{\epsilon}\right)\right) \tag{43}$$

$$= \mathcal{O}\left(\left(n + \sqrt{\frac{n}{m}}\frac{\bar{L}}{\mu} + \frac{L}{\mu}\log m\right)\log\left(\frac{1}{\epsilon}\right)\right)$$

$$= \tilde{\mathcal{O}}\left(\left(n + \sqrt{\frac{n}{m}}\frac{\bar{L}}{\mu} + \frac{L}{\mu}\right)\log\left(\frac{1}{\epsilon}\right)\right),$$

where the first step follows from Lemma C.1 and $1 - p \leq 1$, the second step from $q = b/mn$, and the third step from the choice of parameters $p$ and $b$.

**Communication Rounds:** Based on the value of $R$ in equation (31), we have

$$R = \tilde{\mathcal{O}}\left(\frac{1}{\sqrt{1 - \lambda_2(\mathbf{W})}}\right) = \tilde{\mathcal{O}}\left(\sqrt{\chi}\right).$$

Therefore, the communication rounds can be upper bounded by

$$KR = \tilde{\mathcal{O}}\left(\sqrt{\chi}K\right) = \tilde{\mathcal{O}}\left(\frac{\sqrt{\chi}L}{\mu}\log\left(\frac{1}{\epsilon}\right)\right). \tag{44}$$

**Case II:** $\bar{L} \geq \sqrt{mn}L$.

When $\bar{L} \geq \sqrt{mn}L$, similar to the proof of equation (41), we can upper bound the number of iterations based on Lemma 3.6. Specifically, the number of iterations for $\|\mathbf{z}_i^K - \mathbf{z}^*\|^2 \leq \epsilon$ is

$$K = \mathcal{O}\left(\left(\frac{1}{\mu\eta}\right)\log\left(\frac{1}{\epsilon}\right)\right)$$

$$= \mathcal{O}\left(\frac{L}{\mu}\log\left(\frac{1}{\epsilon}\right)\right). \tag{45}$$

**LIFO Calls:** Under the parameter settings of Lemma 3.6, the LIFO calls required for each iteration are $\mathcal{O}(mn)$. Therefore, the total LIFO calls can be upper bounded by

$$\mathcal{O}\left(mnK\right) = \mathcal{O}\left(\frac{mnL}{\mu}\log\left(\frac{1}{\epsilon}\right)\right). \tag{46}$$

**Computation Rounds:** For each iteration, the computation rounds per node are $\mathcal{O}(n)$, and thus the total computation rounds are bounded by

$$\mathcal{O}\left(nK\right) = \mathcal{O}\left(\frac{nL}{\mu}\log\left(\frac{1}{\epsilon}\right)\right). \tag{47}$$

**Communication Rounds:** Under the parameter settings of Lemma 3.6, it holds that $R = \tilde{\mathcal{O}}\left(\sqrt{\chi}\right)$. The communication rounds are bounded by

$$KR = \tilde{\mathcal{O}}\left(\sqrt{\chi}K\right) = \tilde{\mathcal{O}}\left(\frac{\sqrt{\chi}L}{\mu}\log\left(\frac{1}{\epsilon}\right)\right). \tag{48}$$

Finally, combining the results of Case I and Case II, we obtain that for any $0 < L \leq \bar{L}$, running Algorithm 2 with appropriate parameter settings can find an $\epsilon$-suboptimal solution at each node, with the expected LIFO complexity of

$$\mathcal{O}\left(\left(mn + \frac{\min\{mnL, \sqrt{mn}\bar{L}\}}{\mu}\right)\log\left(\frac{1}{\epsilon}\right)\right),$$

the expected computation rounds of

$$\tilde{\mathcal{O}}\left(\left(n + \frac{L}{\mu} + \frac{\min\{nL, \sqrt{n/m}\bar{L}\}}{\mu}\right)\log\left(\frac{1}{\epsilon}\right)\right),$$

and the communication rounds of

$$\tilde{\mathcal{O}}\left(\frac{\sqrt{\chi}L}{\mu}\log\left(\frac{1}{\epsilon}\right)\right).$$

## D   The Proofs for Section 4

In this section, we will establish the complexity lower bounds presented in Section 4. The proofs of Theorems 4.2, 4.3 and 4.4 are provided in Appendix D.1, D.2 and D.3, respectively.

### D.1   The Proof of Theorem 4.2

Luo et al. [47] established the LIFO complexity lower bound for the single-machine finite-sum problem. We extend this result to the decentralized setting and derive lower bounds with respect to the smoothness parameters $L$ and $\bar{L}$.

Let $d_x = d_y = d$ and $N = mn$. Without loss of generality, the algorithm can be assumed to start at $(\mathbf{x}^0, \mathbf{y}^0) = (\mathbf{0}, \mathbf{0})$. Following Luo et al. [47], we consider the function $H : \mathbb{R}^d \times \mathbb{R}^d \to \mathbb{R}$ defined as

$$H(\mathbf{x}, \mathbf{y}; \gamma, d) = \frac{\gamma}{2}\|\mathbf{x}\|^2 + \mathbf{x}^\top(\mathbf{B}\mathbf{y} - \mathbf{c}) - \frac{\gamma}{2}\|\mathbf{y}\|^2, \tag{49}$$

where

$$\mathbf{B} = \begin{bmatrix} 1 & & & & \\ -1 & 1 & & & \\ & \ddots & \ddots & & \\ & & -1 & 1 & \\ & & & -1 & \sqrt{\gamma\omega} \end{bmatrix} \in \mathbb{R}^{d\times d},$$

$\mathbf{c} = (\omega, 0, 0, \ldots, 0)^\top$ and $\omega = (\sqrt{\gamma^2 + 4} - \gamma)/2$.

To characterize the zero-chain property [58] of $H$, we define subspaces $\mathcal{F}_k$ as

$$\mathcal{F}_k = \begin{cases} \text{span}\{\mathbf{e}_1, \mathbf{e}_2, \ldots, \mathbf{e}_k\}, & k = 1, \ldots, d, \\ \{\mathbf{0}_d\}, & k = 0, \end{cases}$$

where $\{\mathbf{e}_1, \ldots, \mathbf{e}_d\}$ is the standard basis of $\mathbb{R}^d$.

Following lemma characterizes the properties of $H$.

**Lemma D.1** (Luo et al. [47, Lemma 13]). *The function $H$ has following properties:*

1. *$H$ is $\sqrt{8 + 2\gamma^2}$-smooth.*

2. *For $k < d$, if $(\mathbf{x}, \mathbf{y}) \in \mathcal{F}_k \times \mathcal{F}_k$, then $(\nabla_\mathbf{x} H(\mathbf{x}, \mathbf{y}), \nabla_\mathbf{y} H(\mathbf{x}, \mathbf{y})) \in \mathcal{F}_{k+1} \times \mathcal{F}_{k+1}$.*

3. *Let $r = (2 + \gamma^2 - \gamma\sqrt{\gamma^2 + 4})/2$. The saddle point of $H$ is*

$$\begin{cases} \mathbf{x}^* = (r, r^2, \ldots, r^d)^\top, \\ \mathbf{y}^* = \omega\left(r, r^2, \ldots, r^{d-1}, \frac{1}{\sqrt{1-r}}r^d\right)^\top. \end{cases}$$

4. *For $k \leq d/2$ and $(\mathbf{x}, \mathbf{y}) \in \mathcal{F}_k \times \mathcal{F}_k$, it holds that*

$$\frac{\|\mathbf{x} - \mathbf{x}^*\|^2 + \|\mathbf{y} - \mathbf{y}^*\|^2}{\|\mathbf{x}^*\|^2 + \|\mathbf{y}^*\|^2} \geq \frac{1}{2}r^{2k}.$$

**Lemma D.2.** *For the parameters $\bar{L} \geq L$, $\min\{\bar{L}^2, mnL^2\}/\mu^2 > 10mn$, and $\epsilon < e^{-5}/2$, there exists a hard instance satisfying Assumptions 2.1–2.4. In order to find an $\epsilon$-suboptimal solution, the LIFO calls of any LIFO algorithm is lower bounded by*

$$\Omega\left(\frac{\min\{mnL, \sqrt{mn}\bar{L}\}}{\mu} \log\left(\frac{1}{\epsilon}\right)\right).$$

*Proof.* Recalling that $N = mn$, we define a matrix sequence $\{\mathbf{U}_i\}_{i=1}^N$ such that $\mathbf{U}_i \in \mathbb{R}^{d \times Nd}$, $\mathbf{U}_i\mathbf{U}_i^\top = \mathbf{I}$, and $\mathbf{U}_i\mathbf{U}_j^\top = \mathbf{0}$ for any $1 \leq i \neq j \leq N$. we set parameters as

$$\hat{L}^2 = \min\{\bar{L}^2, NL^2\}, \quad \gamma = \sqrt{\frac{8N}{\hat{L}^2/\mu^2 - 2N}}, \quad \hat{\lambda} = \frac{N\mu}{\gamma}, \quad d = \left\lfloor \frac{1}{\gamma}\log\left(\frac{1}{2\epsilon}\right)\right\rfloor - 4.$$

A hard instance $f : \mathbb{R}^{Nd} \times \mathbb{R}^{Nd} \to \mathbb{R}$ can be constructed as $f_{i,j}(\mathbf{x}, \mathbf{y}) = \tilde{f}_{(i-1)\times n+j}(\mathbf{x}, \mathbf{y})$, where

$$\tilde{f}_i(\mathbf{x}, \mathbf{y}) = \hat{\lambda}H(\mathbf{U}_i\mathbf{x}, \mathbf{U}_i\mathbf{y}).$$

The global objective function $f(\mathbf{x}, \mathbf{y})$ is given by

$$f(\mathbf{x}, \mathbf{y}) = \frac{1}{mn}\sum_{i=1}^m \sum_{j=1}^n f_{i,j}(\mathbf{x}, \mathbf{y}) = \frac{1}{N}\sum_{i=1}^N \hat{\lambda}H(\mathbf{U}_i\mathbf{x}, \mathbf{U}_i\mathbf{y})$$

$$= \frac{\hat{\lambda}\gamma}{2N}\|\mathbf{x}\|^2 - \frac{\hat{\lambda}}{N}\mathbf{x}^\top\left(\sum_{i=1}^N \mathbf{U}_i^\top \mathbf{c}\right) + \frac{\hat{\lambda}}{N}\mathbf{x}^\top\left(\sum_{i=1}^N \mathbf{U}_i^\top \mathbf{B}\mathbf{U}_i\right)\mathbf{y} - \frac{\hat{\lambda}\gamma}{2N}\|\mathbf{y}\|^2.$$

Based on $\hat{\lambda} = N\mu/\gamma$, it is straightforward to see that $f$ is $\mu$-strongly-convex-$\mu$-strongly-concave.

Then, we verify that the function satisfies the smoothness Assumptions 2.2 and 2.3.
It follows that, for any $(\mathbf{x}_1, \mathbf{y}_1), (\mathbf{x}_2, \mathbf{y}_2) \in \mathbb{R}^{Nd} \times \mathbb{R}^{Nd}$,

$$\|\nabla_{\mathbf{x}}f(\mathbf{x}_1, \mathbf{y}_1) - \nabla_{\mathbf{x}}f(\mathbf{x}_2, \mathbf{y}_2)\|^2 + \|\nabla_{\mathbf{y}}f(\mathbf{x}_1, \mathbf{y}_1) - \nabla_{\mathbf{y}}f(\mathbf{x}_2, \mathbf{y}_2)\|^2$$

$$= \left\|\frac{1}{N}\sum_{i=1}^N\left(\nabla_{\mathbf{x}}\tilde{f}_i(\mathbf{x}_1, \mathbf{y}_1) - \nabla_{\mathbf{x}}\tilde{f}_i(\mathbf{x}_2, \mathbf{y}_2)\right)\right\|^2 + \left\|\frac{1}{N}\sum_{i=1}^N\left(\nabla_{\mathbf{y}}\tilde{f}_i(\mathbf{x}_1, \mathbf{y}_1) - \nabla_{\mathbf{y}}\tilde{f}_i(\mathbf{x}_2, \mathbf{y}_2)\right)\right\|^2$$

$$= \frac{\hat{\lambda}^2}{N^2}\left(\left\|\sum_{i=1}^N \mathbf{U}_i^\top\left(\nabla_{\mathbf{x}}H(\mathbf{U}_i\mathbf{x}_1, \mathbf{U}_i\mathbf{y}_1) - \nabla_{\mathbf{x}}H(\mathbf{U}_i\mathbf{x}_2, \mathbf{U}_i\mathbf{y}_2)\right)\right\|^2\right.$$

$$\left. + \left\|\sum_{i=1}^N \mathbf{U}_i^\top\left(\nabla_{\mathbf{y}}H(\mathbf{U}_i\mathbf{x}_1, \mathbf{U}_i\mathbf{y}_1) - \nabla_{\mathbf{y}}H(\mathbf{U}_i\mathbf{x}_2, \mathbf{U}_i\mathbf{y}_2)\right)\right\|^2\right)$$

$$= \frac{\hat{\lambda}^2}{N^2}\sum_{i=1}^N\left(\|\nabla_{\mathbf{x}}H(\mathbf{U}_i\mathbf{x}_1, \mathbf{U}_i\mathbf{y}_1) - \nabla_{\mathbf{x}}H(\mathbf{U}_i\mathbf{x}_2, \mathbf{U}_i\mathbf{y}_2)\|^2\right.$$

$$\left. + \|\nabla_{\mathbf{y}}H(\mathbf{U}_i\mathbf{x}_1, \mathbf{U}_i\mathbf{y}_1) - \nabla_{\mathbf{y}}H(\mathbf{U}_i\mathbf{x}_2, \mathbf{U}_i\mathbf{y}_2)\|^2\right)$$

$$\leq \frac{\hat{\lambda}^2(8 + 2\gamma^2)}{N^2}\sum_{i=1}^N\left(\|\mathbf{U}_i(\mathbf{x}_1 - \mathbf{x}_2)\|^2 + \|\mathbf{U}_i(\mathbf{y}_1 - \mathbf{y}_2)\|^2\right)$$

$$= \frac{\hat{\lambda}^2(8 + 2\gamma^2)}{N^2}\left(\|\mathbf{x}_1 - \mathbf{x}_2\|^2 + \|\mathbf{y}_1 - \mathbf{y}_2\|^2\right)$$

$$= \frac{\hat{L}^2}{N^2}\left(\|\mathbf{x}_1 - \mathbf{x}_2\|^2 + \|\mathbf{y}_1 - \mathbf{y}_2\|^2\right)$$

$$\leq L^2\left(\|\mathbf{x}_1 - \mathbf{x}_2\|^2 + \|\mathbf{y}_1 - \mathbf{y}_2\|^2\right),$$

where the first inequality holds by Property 1 in Lemma D.1 and the second inequality follows from the setting of $\hat{L}$.

Meanwhile, for any $(\mathbf{x}_1, \mathbf{y}_1), (\mathbf{x}_2, \mathbf{y}_2) \in \mathbb{R}^{Nd} \times \mathbb{R}^{Nd}$, it holds that

$$\frac{1}{mn} \sum_{i=1}^{m} \sum_{j=1}^{n} \left( \|\nabla_{\mathbf{x}} f_{i,j}(\mathbf{x}_1, \mathbf{y}_1) - \nabla_{\mathbf{x}} f_{i,j}(\mathbf{x}_2, \mathbf{y}_2)\|^2 + \|\nabla_{\mathbf{y}} f_{i,j}(\mathbf{x}_1, \mathbf{y}_1) - \nabla_{\mathbf{y}} f_{i,j}(\mathbf{x}_2, \mathbf{y}_2)\|^2 \right)$$

$$= \frac{1}{N} \sum_{i=1}^{N} \left( \left\|\nabla_{\mathbf{x}} \tilde{f}_i(\mathbf{x}_1, \mathbf{y}_1) - \nabla_{\mathbf{x}} \tilde{f}_i(\mathbf{x}_2, \mathbf{y}_2)\right\|^2 + \left\|\nabla_{\mathbf{y}} \tilde{f}_i(\mathbf{x}_1, \mathbf{y}_1) - \nabla_{\mathbf{y}} \tilde{f}_i(\mathbf{x}_2, \mathbf{y}_2)\right\|^2 \right)$$

$$= \frac{\hat{\lambda}^2}{N} \sum_{i=1}^{N} \left( \left\|\mathbf{U}_i^\top \left(\nabla_{\mathbf{x}} H(\mathbf{U}_i \mathbf{x}_1, \mathbf{U}_i \mathbf{y}_1) - \nabla_{\mathbf{x}} H(\mathbf{U}_i \mathbf{x}_2, \mathbf{U}_i \mathbf{y}_2)\right)\right\|^2 \right.$$

$$\left. + \left\|\mathbf{U}_i^\top \left(\nabla_{\mathbf{y}} H(\mathbf{U}_i \mathbf{x}_1, \mathbf{U}_i \mathbf{y}_1) - \nabla_{\mathbf{y}} H(\mathbf{U}_i \mathbf{x}_2, \mathbf{U}_i \mathbf{y}_2)\right)\right\|^2 \right)$$

$$= \frac{\hat{\lambda}^2}{N} \sum_{i=1}^{N} \left( \left\|\nabla_{\mathbf{x}} H(\mathbf{U}_i \mathbf{x}_1, \mathbf{U}_i \mathbf{y}_1) - \nabla_{\mathbf{x}} H(\mathbf{U}_i \mathbf{x}_2, \mathbf{U}_i \mathbf{y}_2)\right\|^2 \right.$$

$$\left. + \left\|\nabla_{\mathbf{y}} H(\mathbf{U}_i \mathbf{x}_1, \mathbf{U}_i \mathbf{y}_1) - \nabla_{\mathbf{y}} H(\mathbf{U}_i \mathbf{x}_2, \mathbf{U}_i \mathbf{y}_2)\right\|^2 \right)$$

$$\leq \frac{\hat{\lambda}^2 (8 + 2\gamma^2)}{N} \sum_{i=1}^{N} \left( \|\mathbf{U}_i(\mathbf{x}_1 - \mathbf{x}_2)\|^2 + \|\mathbf{U}_i(\mathbf{y}_1 - \mathbf{y}_2)\|^2 \right)$$

$$= \frac{\hat{\lambda}^2 (8 + 2\gamma^2)}{N} \left( \|\mathbf{x}_1 - \mathbf{x}_2\|^2 + \|\mathbf{y}_1 - \mathbf{y}_2\|^2 \right)$$

$$= \hat{L}^2 \left( \|\mathbf{x}_1 - \mathbf{x}_2\|^2 + \|\mathbf{y}_1 - \mathbf{y}_2\|^2 \right)$$

$$\leq \bar{L}^2 \left( \|\mathbf{x}_1 - \mathbf{x}_2\|^2 + \|\mathbf{y}_1 - \mathbf{y}_2\|^2 \right).$$

For any LIFO algorithm with at most $\lfloor Nd/2 \rfloor$ LIFO calls, by Property 2 in Lemma D.1, the variable $\mathbf{x}$ has at most $\lfloor Nd/2 \rfloor$ non-zero coordinates. Therefore, there exist an index $i_0$ such that $\mathbf{U}_{i_0}\mathbf{x} \in \mathcal{F}_{d/2}$. Then by Property 4 in Lemma D.1, we have

$$\mathbb{E}\left[\|\mathbf{x} - \mathbf{x}^*\|^2 + \|\mathbf{y} - \mathbf{y}^*\|^2\right] \geq \mathbb{E}\left[\|\mathbf{U}_{i_0}(\mathbf{x} - \mathbf{x}^*)\|^2 + \|\mathbf{U}_{i_0}(\mathbf{y} - \mathbf{y}^*)\|^2\right]$$

$$\geq \frac{r^{d/2}}{2} \left(\|\mathbf{U}_{i_0}\mathbf{x}^*\|^2 + \|\mathbf{U}_{i_0}\mathbf{y}^*\|^2\right)$$

$$\geq \frac{r^{d/2}}{2} r^2 \frac{1 - r^{2d}}{1 - r^2}$$

$$\geq \frac{r^{d/2+2}}{2},$$

where the third inequality holds by the Property 3 in Lemma D.1, i.e., $\mathbf{U}_{i_0}\mathbf{x}^* = (r, r^2, \ldots, r^d)^\top$.

Note that

$$(\frac{d}{2} + 2)\log(\frac{1}{r}) = (\frac{d}{2} + 2)\log(1 + \frac{\gamma(\gamma + \sqrt{\gamma^2 + 4})}{2})$$

$$\leq (\frac{d}{2} + 2)\frac{\gamma(\gamma + \sqrt{\gamma^2 + 4})}{2}$$

$$< (\frac{d}{2} + 2)\gamma(\gamma + 1),$$

where the first inequality follows the fact that $\log(1 + x) \leq x$ and the second inequality holds by $\sqrt{a+b} < \sqrt{a} + \sqrt{b}$ for $a, b > 0$. Since $\min\{\bar{L}^2, NL^2\}/\mu^2 > 10N$, we have $\gamma = \sqrt{8N/(\hat{L}^2/\mu^2 - 2N)} \leq 1$, which further implies

$$(\frac{d}{2} + 2)\log(1/r) < (d + 4)\gamma \leq \log(1/2\epsilon).$$

Thus, when the number of LIFO calls is less than $\lfloor Nd/2 \rfloor$, we have $\mathbb{E}\left[\|\mathbf{x} - \mathbf{x}^*\|^2 + \|\mathbf{y} - \mathbf{y}^*\|^2\right] > \epsilon$. Then the LIFO calls to find an $\epsilon$-suboptimal solution is lower bounded by

$$\frac{Nd}{2} = \Omega\left(\frac{\sqrt{N}\hat{L}}{\mu}\log\left(\frac{1}{\epsilon}\right)\right) = \Omega\left(\frac{\min\{mnL, \sqrt{mn}\bar{L}\}}{\mu}\log\left(\frac{1}{\epsilon}\right)\right).$$

This completes the proof of Lemma D.2. $\qquad\square$

For the case $\min\{\bar{L}^2, mnL^2\}/\mu^2 = \mathcal{O}(\sqrt{mn})$, we have the following lemma.

**Lemma D.3.** *For the parameters $\bar{L} \geq L$, $L/\mu > 2$, and $\epsilon < 1/8$, there exists a hard instance satisfying Assumptions 2.1–2.4. In order to find an $\epsilon$-suboptimal solution, the LIFO calls of any LIFO algorithm is lower bounded by $\Omega(mn)$.*

*Proof.* Let $x_i$ and $y_i$ denote the $i$-th coordinates of the vectors $\mathbf{x}$ and $\mathbf{y}$, respectively. Consider the functions $f_{i,j}(\mathbf{x}, \mathbf{y}) = \tilde{f}_{(i-1)\times n+j}(\mathbf{x}, \mathbf{y})$, where

$$\tilde{f}_i(\mathbf{x}, \mathbf{y}) = \frac{\mu}{2}\|\mathbf{x}\|^2 + \frac{\sqrt{mn}\hat{L}}{2}(x_i - 1)^2 - \frac{\mu}{2}\|\mathbf{y}\|^2 - \frac{\sqrt{mn}\hat{L}}{2}(y_i - 1)^2$$

for $i \in [mn]$ and $\hat{L} = \sqrt{L^2/2 - \mu^2}$. It follows that the global objective $f(\mathbf{x}, \mathbf{y}) : \mathbb{R}^{mn} \times \mathbb{R}^{mn} \to \mathbb{R}$ takes the form

$$f(\mathbf{x}, \mathbf{y}) = \frac{\mu}{2}\|\mathbf{x}\|^2 + \frac{\hat{L}}{2\sqrt{mn}}\|\mathbf{x} - \mathbf{1}\|^2 - \frac{\mu}{2}\|\mathbf{y}\|^2 - \frac{\hat{L}}{2\sqrt{mn}}\|\mathbf{y} - \mathbf{1}\|^2.$$

It is clear that $f$ is $\mu$-strongly-convex-$\mu$-strongly concave and the saddle point $(\mathbf{x}^*, \mathbf{y}^*)$ satisfies

$$\mathbf{x}^* = \mathbf{y}^* = \frac{\hat{L}}{\hat{L} + \sqrt{mn}\mu}\mathbf{1}.$$

For any $(\mathbf{x}_1, \mathbf{y}_1), (\mathbf{x}_2, \mathbf{y}_2) \in \mathbb{R}^{mn} \times \mathbb{R}^{mn}$, we have

$$\frac{1}{mn}\sum_{i=1}^{m}\sum_{j=1}^{n}\left(\|\nabla_{\mathbf{x}}f_{i,j}(\mathbf{x}_1, \mathbf{y}_1) - \nabla_{\mathbf{x}}f_{i,j}(\mathbf{x}_2, \mathbf{y}_2)\|^2 + \|\nabla_{\mathbf{y}}f_{i,j}(\mathbf{x}_1, \mathbf{y}_1) - \nabla_{\mathbf{y}}f_{i,j}(\mathbf{x}_2, \mathbf{y}_2)\|^2\right)$$

$$= \frac{1}{mn}\sum_{i=1}^{mn}\left(\|\nabla_{\mathbf{x}}\tilde{f}_i(\mathbf{x}_1, \mathbf{y}_1) - \nabla_{\mathbf{x}}\tilde{f}_i(\mathbf{x}_1, \mathbf{y}_1)\|^2 + \|\nabla_{\mathbf{y}}\tilde{f}_i(\mathbf{x}_1, \mathbf{y}_1) - \nabla_{\mathbf{y}}\tilde{f}_i(\mathbf{x}_1, \mathbf{y}_1)\|^2\right)$$

$$= \frac{1}{mn}\sum_{i=1}^{mn}\left(\|\mu(\mathbf{x}_1 - \mathbf{x}_2) + \sqrt{mn}\hat{L}\mathbf{e}_i\mathbf{e}_i^\top(\mathbf{x}_1 - \mathbf{x}_2)\|^2 + \|\mu(\mathbf{y}_1 - \mathbf{y}_2) + \sqrt{mn}\hat{L}\mathbf{e}_i\mathbf{e}_i^\top(\mathbf{y}_1 - \mathbf{y}_2)\|^2\right)$$

$$\leq \frac{1}{mn}\sum_{i=1}^{mn}\left(2\mu^2\|\mathbf{x}_1 - \mathbf{x}_2\|^2 + 2mn\hat{L}^2\|\mathbf{e}_i\mathbf{e}_i^\top(\mathbf{x}_1 - \mathbf{x}_2)\|^2\right.$$

$$\left. + 2\mu^2\|\mathbf{y}_1 - \mathbf{y}_2\|^2 + 2mn\hat{L}^2\|\mathbf{e}_i\mathbf{e}_i^\top(\mathbf{y}_1 - \mathbf{y}_2)\|^2\right)$$

$$= 2(\hat{L}^2 + \mu^2)\left(\|\mathbf{x}_1 - \mathbf{x}_2\|^2 + \|\mathbf{y}_1 - \mathbf{y}_2\|^2\right)$$

$$= L^2\left(\|\mathbf{x}_1 - \mathbf{x}_2\|^2 + \|\mathbf{y}_1 - \mathbf{y}_2\|\right).$$

Thus, the function set $\{f_{i,j}\}_{i,j=1}^{m,n}$ is $\bar{L}$-mean-squared smooth ($L \leq \bar{L}$) and $f$ is $L$-smooth.

By the zero-chain property of $f$, if the LIFO calls are less than $mn/2$, it holds that

$$\mathbb{E}\left[\|\mathbf{x} - \mathbf{x}^*\|^2 + \|\mathbf{y} - \mathbf{y}^*\|^2\right]$$

$$\geq \frac{mn}{2} \cdot \frac{\hat{L}^2}{(\hat{L} + \sqrt{mn}\mu)^2}$$

$$\geq \frac{mn}{2} \cdot \frac{\hat{L}^2}{2\hat{L}^2 + 2mn\mu^2}$$

$$= \frac{mn}{2} \cdot \frac{L^2/2 - \mu^2}{L^2 - 2\mu^2 + 2mn\mu^2}$$

$$= \frac{mn}{4} \left( 1 - \frac{2mn}{L^2/\mu^2 + 2mn - 2} \right)$$

$$\geq \frac{mn}{4(mn+1)} \geq \frac{1}{8} > \epsilon.$$

Therefore, the LIFO calls to find an $\epsilon$-suboptimal solution is lower bounded by $\Omega(mn)$. $\square$

Combing the results of Lemmas D.2 and D.3, for the parameters $\bar{L} \geq L$, $L/\mu > 2$, and $\epsilon < 0.003$, the LIFO calls of any LIFO algorithm is lower bounded by

$$\Omega \left( mn + \frac{\min\{mnL, \sqrt{mn}\bar{L}\}}{\mu} \log\left(\frac{1}{\epsilon}\right) \right).$$

### D.2 The proof of Theorem 4.3

For any decentralized LIFO algorithm, it can perform at most $m$ LIFO calls in each computation round. From the conclusion of Theorem 4.2, it follows that the LIFO calls are lower bounded by

$$\Omega \left( mn + \frac{\min\left\{mnL, \sqrt{mn}\bar{L}\right\}}{\mu} \log\left(\frac{1}{\epsilon}\right) \right).$$

Therefore, the computation rounds have the lower bound

$$\Omega \left( \frac{1}{m} \left( mn + \frac{\min\left\{mnL, \sqrt{mn}\bar{L}\right\}}{\mu} \log\left(\frac{1}{\epsilon}\right) \right) \right) = \Omega \left( n + \frac{\min\left\{nL, \sqrt{n/m}\bar{L}\right\}}{\mu} \log\left(\frac{1}{\epsilon}\right) \right). \tag{50}$$

Another instance is a direct extension of the single-machine setting [89]. For all $i \in [m]$ and $j \in [n]$, define

$$f_{i,j}(\mathbf{x}, \mathbf{y}) = f(\mathbf{x}, \mathbf{y}) = \frac{\sqrt{L^2 - \mu^2}}{2} \mathbf{x}^\top \mathbf{A} \mathbf{y} + \frac{L^2 - \mu^2}{4\mu} \mathbf{e}_1^\top \mathbf{y} + \frac{\mu}{2} \|\mathbf{x}\|^2 - \frac{\mu}{2} \|\mathbf{y}\|^2, \tag{51}$$

where

$$\mathbf{A} = \begin{bmatrix} 1 & -1 & & & \\ & 1 & -1 & & \\ & & \ddots & \ddots & \\ & & & 1 & -1 \\ & & & & 1 \end{bmatrix} \in \mathbb{R}^{d \times d}.$$

It can be verified that the function defined in (51) satisfies Assumptions 2.1, 2.2 and 2.3, as all $f_{i,j}$ are $L$-smooth. From the single-machine case [89], it is straightforward to deduce that the algorithm requires at least $\Omega(L/\mu \log(1/\epsilon))$ iterations to achieve an $\epsilon$-sub-optimal solution. Since each iteration requires at least one computation round, this implies a lower bound on the computation rounds of

$$\Omega \left( \frac{L}{\mu} \log\left(\frac{1}{\epsilon}\right) \right). \tag{52}$$

Combining the results of equations (50) and (52), it follows that the computation rounds have a lower bound of

$$\Omega \left( n + \left( \frac{L}{\mu} + \frac{\min\{nL, \sqrt{n/m}\bar{L}\}}{\mu} \right) \log\left(\frac{1}{\epsilon}\right) \right).$$

## D.3 The proof of Theorem 4.4

The main idea of proving the lower bound on communication complexity is to extend the difficult examples constructed by Zhang et al. [89] to the distributed setting. First, we divide all nodes into three sets $\mathcal{V}_1$, $\mathcal{V}_2$, and $\mathcal{V}_3$, where $|\mathcal{V}_1| = |\mathcal{V}_2| = \lfloor m/3 \rfloor$, and $|\mathcal{V}_3| = m - |\mathcal{V}_1| - |\mathcal{V}_2|$. Let $d_x = d_y = d$, $\hat{L}^2 = (L^2 - \mu^2)|\mathcal{V}_1|/(2m)$,

$$
\mathbf{A}_1 = \begin{bmatrix} 1 & 0 & & & \\ & 1 & -2 & & \\ & & \ddots & \ddots & \\ & & & 1 & 0 \\ & & & & 1 \end{bmatrix}, \quad \mathbf{A}_2 = \begin{bmatrix} 1 & -2 & & & \\ & 1 & 0 & & \\ & & \ddots & \ddots & \\ & & & 1 & -2 \\ & & & & 1 \end{bmatrix} \in \mathbb{R}^{d \times d},
$$

and $\mathbf{A} = (\mathbf{A}_1 + \mathbf{A}_2)/2$. For $i = 1, 2, \dots, m$, the functions are constructed as $f_{i,1}(\mathbf{x}, \mathbf{y}) = \cdots = f_{i,n}(\mathbf{x}, \mathbf{y}) = f_i(\mathbf{x}, \mathbf{y})$, where

$$
f_i(\mathbf{x}, \mathbf{y}) = \begin{cases} \dfrac{m}{|\mathcal{V}_1|}\left(\dfrac{\hat{L}}{4}\mathbf{x}^\top \mathbf{A}_1 \mathbf{y} + \dfrac{\hat{L}^2}{4\mu}\mathbf{e}_1^\top \mathbf{y}\right) + \dfrac{\mu}{2}\|\mathbf{x}\|^2 - \dfrac{\mu}{2}\|\mathbf{y}\|^2, & i \in \mathcal{V}_1, \\[3mm] \dfrac{m}{|\mathcal{V}_2|}\left(\dfrac{\hat{L}}{4}\mathbf{x}^\top \mathbf{A}_2 \mathbf{y}\right) + \dfrac{\mu}{2}\|\mathbf{x}\|^2 - \dfrac{\mu}{2}\|\mathbf{y}\|^2, & i \in \mathcal{V}_2, \\[3mm] \dfrac{\mu}{2}\|\mathbf{x}\|^2 - \dfrac{\mu}{2}\|\mathbf{y}\|^2, & i \in \mathcal{V}_3. \end{cases} \tag{53}
$$

Then the global objective function takes the form

$$
f(\mathbf{x}, \mathbf{y}) = \frac{\hat{L}}{2}\mathbf{x}^\top \mathbf{A}\mathbf{y} + \frac{\hat{L}^2}{4\mu}\mathbf{e}_1^\top \mathbf{y} + \frac{\mu}{2}\|\mathbf{x}\|^2 - \frac{\mu}{2}\|\mathbf{y}\|^2.
$$

We will prove that the constructed functions are $L$-mean-squared smooth, thereby satisfying Assumptions 2.2 and 2.3. For any $\mathbf{x}_1, \mathbf{x}_2, \mathbf{y}_1, \mathbf{y}_2 \in \mathbb{R}^d$,

$$
\frac{1}{mn}\sum_{i=1}^m \sum_{j=1}^n \left(\|\nabla_\mathbf{x} f_{i,j}(\mathbf{x}_1, \mathbf{y}_1) - \nabla_\mathbf{x} f_{i,j}(\mathbf{x}_2, \mathbf{y}_2)\|^2 + \|\nabla_\mathbf{y} f_{i,j}(\mathbf{x}_1, \mathbf{y}_1) - \nabla_\mathbf{y} f_{i,j}(\mathbf{x}_2, \mathbf{y}_2)\|^2\right)
$$

$$
= \frac{1}{m}\sum_{i=1}^m \left(\|\nabla_\mathbf{x} f_i(\mathbf{x}_1, \mathbf{y}_1) - \nabla_\mathbf{x} f_i(\mathbf{x}_2, \mathbf{y}_2)\|^2 + \|\nabla_\mathbf{y} f_i(\mathbf{x}_1, \mathbf{y}_1) - \nabla_\mathbf{y} f_i(\mathbf{x}_2, \mathbf{y}_2)\|^2\right)
$$

$$
= \frac{1}{m}\sum_{i \in \mathcal{V}_1} \left(\left\|\frac{m}{|\mathcal{V}_1|}\frac{\hat{L}}{4}\mathbf{A}_1(\mathbf{y}_1 - \mathbf{y}_2) + \mu(\mathbf{x}_1 - \mathbf{x}_2)\right\|^2 + \left\|\frac{m}{|\mathcal{V}_1|}\frac{\hat{L}}{4}\mathbf{A}_1^\top(\mathbf{x}_1 - \mathbf{x}_2) - \mu(\mathbf{y}_1 - \mathbf{y}_2)\right\|^2\right)
$$

$$
+ \frac{1}{m}\sum_{i \in \mathcal{V}_2} \left(\left\|\frac{m}{|\mathcal{V}_2|}\frac{\hat{L}}{4}\mathbf{A}_2(\mathbf{y}_1 - \mathbf{y}_2) + \mu(\mathbf{x}_1 - \mathbf{x}_2)\right\|^2 + \left\|\frac{m}{|\mathcal{V}_2|}\frac{\hat{L}}{4}\mathbf{A}_2^\top(\mathbf{x}_1 - \mathbf{x}_2) - \mu(\mathbf{y}_1 - \mathbf{y}_2)\right\|^2\right)
$$

$$
+ \frac{|\mathcal{V}_3|}{m}\mu^2\left(\|\mathbf{x}_1 - \mathbf{x}_2\|^2 + \|\mathbf{y}_1 - \mathbf{y}_2\|^2\right)
$$

$$
= \frac{m}{|\mathcal{V}_1|}\frac{\hat{L}^2}{16}\left(\|\mathbf{A}_1(\mathbf{y}_1 - \mathbf{y}_2)\|^2 + \|\mathbf{A}_1^\top(\mathbf{x}_1 - \mathbf{x}_2)\|^2\right)
$$

$$
+ \frac{m}{|\mathcal{V}_2|}\frac{\hat{L}^2}{16}\left(\|\mathbf{A}_2(\mathbf{y}_1 - \mathbf{y}_2)\|^2 + \|\mathbf{A}_2^\top(\mathbf{x}_1 - \mathbf{x}_2)\|^2\right) + \mu^2\left(\|\mathbf{x}_1 - \mathbf{x}_2\|^2 + \|\mathbf{y}_1 - \mathbf{y}_2\|^2\right)
$$

$$
\leq \left(\frac{2m}{|\mathcal{V}_1|}\hat{L}^2 + \mu^2\right)\left(\|\mathbf{x}_1 - \mathbf{x}_2\|^2 + \|\mathbf{y}_1 - \mathbf{y}_2\|^2\right)
$$

$$
= L^2\left(\|\mathbf{x}_1 - \mathbf{x}_2\|^2 + \|\mathbf{y}_1 - \mathbf{y}_2\|^2\right),
$$

where the inequality holds because $\|\mathbf{A}_1\|_2 \leq 4$, $\|\mathbf{A}_2\|_2 \leq 4$ and $|\mathcal{V}_1| = |\mathcal{V}_2|$.

For a decentralized algorithm starting at $(\mathbf{x}^0, \mathbf{y}^0) = (\mathbf{0}, \mathbf{0})$, its communication complexity depends on the distance between $\mathcal{V}_1$ and $V_2$ on the graph, which is denoted by $D(\mathcal{V}_1, \mathcal{V}_2)$. Based on the zero-chain property of the constructed function, the following lemma holds.

**Lemma D.4.** *Consider the minimax problem with the objective functions (53). For any algorithm satisfying Definition 4.1, after $R$ communication rounds, the algorithm's output has only the first $\lfloor R/D(\mathcal{V}_1, \mathcal{V}_2) \rfloor$ coordinates non-zero, while the remaining $d - \lfloor R/D(\mathcal{V}_1, \mathcal{V}_2) \rfloor$ coordinates are zero.*

*Proof.* Initially, for a variable in $\mathcal{V}_1$, transmitting it to $\mathcal{V}_2$ requires at least $D(\mathcal{V}_1, \mathcal{V}_2)$ communication rounds. According to the constructed function (53), nodes in $\mathcal{V}_2$ will maintain $\mathbf{x} = \mathbf{y} = \mathbf{0}$ until receiving values from $\mathcal{V}_1$. Similarly, the nodes in $\mathcal{V}_1$ will have at most the first coordinate non-zero until they receive the non-zero values returned from $\mathcal{V}_2$. (Nodes in $\mathcal{V}_3$ do not contribute to increasing the number of non-zero coordinates.) In subsequent steps, each additional non-zero coordinate needs at least $D(\mathcal{V}_1, \mathcal{V}_2)$ communication rounds. After $R$ communication rounds, the output of the algorithm will have only the first $\lfloor R/D(\mathcal{V}_1, \mathcal{V}_2) \rfloor$ coordinates non-zero, while the remaining $\lfloor R/D(\mathcal{V}_1, \mathcal{V}_2) \rfloor$ coordinates are zero. $\qquad\square$

The following lemma establishes a lower bound on the distance between the current point and the optimal solution.

**Lemma D.5** (Zhang et al. [89, Theorem 3.5]). *For $d \geq \max\{4k, 2\log(\iota/4\sqrt{2})\}$ and $(\mathbf{x}, \mathbf{y}) \in \mathcal{F}_k \times \mathcal{F}_k$, it holds that*

$$\mathbb{E}\left[\|\mathbf{x} - \mathbf{x}^*\|^2 + \|\mathbf{y} - \mathbf{y}^*\|^2\right] \geq \tau^k \frac{\|\mathbf{y}^0 - \mathbf{y}^*\|^2}{16},$$

*where $\tau = ((2 + \iota) - \sqrt{(2 + \iota)^2 - 4})/2 \in (0, 1)$ and $\iota = 4\mu^2/\hat{L}^2$.*

Combing the result of Lemma D.4, it holds that for any decentralized first-order algorithm satisfying Definition 4.1, the output $(\mathbf{x}_{\text{out}}^R, \mathbf{y}_{\text{out}}^R)$ after $R$ communication rounds will satisfy

$$\mathbb{E}\left[\|\mathbf{x}_{\text{out}}^R - \mathbf{x}^*\|^2 + \|\mathbf{y}_{\text{out}}^R - \mathbf{y}^*\|^2\right] \geq \tau^{\frac{R}{D(\mathcal{V}_1, \mathcal{V}_2)}} \frac{\|\mathbf{y}^0 - \mathbf{y}^*\|^2}{16}. \tag{54}$$

**Lemma D.6** (Yuan et al. [88, Theorem 1]). *For any $m \geq 2$ and $\lambda_2 \in [0, \cos(\pi/m)]$, we can always construct a ring-lattice graph so that the mixing matrix $\mathbf{W}$ satisfies Assumption 2.4 and $\lambda_2(\mathbf{W}) = \lambda_2$, and the diameter of the graph satisfies*

$$D(\mathcal{V}_1, \mathcal{V}_2) = \Omega\left(\sqrt{\chi}\right).$$

With the above lemmas, we proceed to prove Theorem 4.4.

*Proof of Theorem 4.4.* By equation (54), to find an $\epsilon$-suboptimal solution, communication rounds $R$ are lower bounded by

$$R \geq D(\mathcal{V}_1, \mathcal{V}_2) \log\left(\frac{\|\mathbf{y}^0 - \mathbf{y}^*\|^2}{16\epsilon}\right) / \log \frac{1}{\tau}.$$

For any $x > 0$, it holds that $(\log(1 + x))^{-1} \geq x^{-1}$. Thus, we have

$$\begin{aligned}
\left(\log \frac{1}{\tau}\right)^{-1} &= \left(\log\left(1 + \left(\frac{1}{\tau} - 1\right)\right)\right)^{-1} \\
&\geq \frac{\tau}{1 - \tau} \\
&= \frac{1 + \frac{\iota}{2} - \frac{1}{2}\sqrt{(2 + \iota)^2 - 4}}{\frac{1}{2}\sqrt{(2 + \iota)^2 - 4} - \frac{\iota}{2}} \\
&= \frac{\sqrt{\iota(4 + \iota)}}{2\iota} - \frac{1}{2} \\
&= \frac{1}{2}\sqrt{\frac{\hat{L}^2}{4\mu^2}\left(4 + \frac{4\mu^2}{\hat{L}^2}\right)} - \frac{1}{2}
\end{aligned}$$

$$= \frac{1}{2}\sqrt{\frac{\hat{L}^2}{\mu^2} + 1} - \frac{1}{2}$$

$$= \Omega\left(\frac{\hat{L}}{\mu}\right).$$

Based on the fact that $\hat{L}^2 = (L^2 - \mu^2)|\mathcal{V}_1|/(2m)$, where $|\mathcal{V}_1| = \lfloor m/3 \rfloor$ and $L \geq 2\mu$, it follows that $\hat{L} = \Omega(L)$. Therefore, for any decentralized LIFO algorithm, the communication rounds have the lower bound:

$$R \geq D(\mathcal{V}_1, \mathcal{V}_2) \log\left(\frac{\|\mathbf{y}^0 - \mathbf{y}^*\|^2}{16\epsilon}\right) / \log\frac{1}{\tau}$$

$$= \Omega\left(D(\mathcal{V}_1, \mathcal{V}_2)\frac{L}{\mu}\log\left(\frac{1}{\epsilon}\right)\right)$$

$$= \Omega\left(\frac{\sqrt{\chi}L}{\mu}\log\left(\frac{1}{\epsilon}\right)\right),$$

where the last step holds by Lemma D.6.

