# OpenReview forum: "A Near-Optimal Algorithm for Decentralized Convex-Concave Finite-Sum Minimax Optimization"
_NeurIPS.cc/2025/Conference — NeurIPS 2025 spotlight_

### Official Review · Reviewer_96So · 2025-07-01

**Clarity:** 2
**Significance:** 3
**Originality:** 2
**Rating:** 2
**Confidence:** 4

**Summary:**

This paper addresses decentralized convex-concave finite-sum min-max optimization over networks and proposes the DIVERSE algorithm (Decentralized Variance-Reduced Optimistic Gradient with Stochastic mini-batch sizes). While the theoretical analysis is solid and the claimed complexity results improve upon existing works under strongly-convex-strongly-concave settings, the submission falls short in several key areas. In particular, the algorithmic novelty over prior methods is not sufficiently emphasized or clearly explained.  The experimental section is limited and does not adequately support the theoretical claims or demonstrate practical advantages.  Furthermore, the overall presentation lacks clarity, especially in conveying intuition and key technical contributions.

**Questions:**

1. Insufficient Clarification of Algorithmic Novelty​
The core contribution—stochastic mini-batch sampling—lacks thorough motivation. The authors should explicitly clarify:
(a) How the Bernoulli sampling scheme overcomes the "identical mini-batch size" limitation (Lines 46–47) and enables partial participation across nodes;
(b) How this design shifts complexity dependence from local smoothness parameters s ( $L_l, \bar{L}_l$ ) to global ones ( $L, \bar{L}$ ), with direct comparisons to methods like OADSVI [37]. This should be emphasized in Sections 1 and 3.1.
​
2.Inadequate Experimental Validation​
Experiments are limited to robust linear regression (quadratic costs, Eq. 52) on a ring network. To demonstrate generality:
(a) Include complex minimax benchmarks (e.g., GAN training or adversarial learning);
(b) Test under data-heterogeneity scenarios (Example A.1) to validate superiority over GT-EG/MC-SVRE in practical settings where $\bar{L} \gg L$.

3. Ambiguous Parameter Selection Rationale
Critical parameters in Algorithm 2 $(p, b, \alpha)$ are presented without derivation (Lemma 3.1). The authors must:
(a) Justify design choices (e.g., $p=\frac{\bar{L}}{8 L} \max  \left( \frac{\mu}{L}, \frac{1}{\sqrt{m n}} \right)$ ) in the main text or appendix;
(b) Discuss practical implications of the $\bar{L} \leq \sqrt{m n} L$ vs. $\bar{L}>\sqrt{m n} L$ cases on parameter tuning.

4. Experiments Overlook Theoretical Highlights
Experiments omit key innovations:
(a) Performance under varying $\frac{\bar{L}}{L}$ ratios (to showcase global-smoothness dependency);
(b) Communication rounds across topologies with different $\chi$ values (Fig. 1's ring network inadequately validates O $\left(\frac{\sqrt{x} L}{\mu}\right)$ complexity).
(c) The authors fail to provide additional datasets, case studies, and code to support reproducibility and further validation of their approach.

5. Writing and Presentation Issues
(a) Definition gaps: Asymptotic notation $\Omega(\cdot)$ is undefined in Theorem 4.3;
(b) Formatting inconsistencies: Indentation in Algorithms 1-2 and line breaks in Table 1 (e.g., DIVERSE computation rounds) require alignment;
(c) Motivation clarity: The introduction inadequately explains the practical significance of global-smoothness dependency (e.g., acceleration in $\frac{\bar{L}}{L} \gg 1$ regimes).

**Ethical Concerns:**

["NO or VERY MINOR ethics concerns only"]

**Final Justification:**

Thank you very much for your detailed feedback. You have answered all my questions and opinions. I will improve my score.

**Limitations:**

Acknowledged but not mitigated: The SC-SC assumption (Sec 2.1) is clearly stated but lacks:
- Extension plans to nonconvex settings (e.g., GANs)
- Discussion of relaxations (e.g., weak convexity)

**Paper Formatting Concerns:**

I didn't find any major formatting issues in this article.

**Quality:**

3

**Strengths And Weaknesses:**

Strengths
1. Near-Optimal Complexity with Global Smoothness Dependence
- Achieves tight computation complexity $\tilde{O}\left(m n+\frac{\min \{m n L, \sqrt{m n} \bar{L}\}}{\mu} \log \frac{1}{\epsilon}\right)$
and communication complexity $\tilde{\mathrm{O}}\left(\sqrt{\chi} \cdot \frac{L}{\mu} \log \frac{1}{\epsilon}\right)$.  First decentralized minimax method whose complexity depends solely on global smoothness parameters ( $L, \bar{L}$ ) instead of conservative local bounds ( $L_l, \bar{L}_l$ ).Matches lower bounds (Theorems 4.2-4.4), outperforming prior art by orders of magnitude under data heterogeneity (e.g., $10 \times$ speedup when $\bar{L} / L=\Theta(\sqrt{m})$ in Ex. A.1).
2. Breakthrough Sampling Strategy: Stochastic Mini-Batches
- Bernoulli-based batch sizing ( $b_{i, j}^k \sim \operatorname{Bernoulli}(q)$ ) enables:
- Node partial participation: Weak nodes can skip iterations dynamically.
- No synchronized batch sizes: Eliminates computation bottlenecks in heterogeneous systems.
- Reduces expected per-iteration cost to $\mathrm{O}(m n p+(1-p) b)$ vs. $\mathrm{O}(m n)$ in existing methods.

Weaknesses
- Variance reduction (Section 2.2) and multi-consensus (Algorithm 1) are repurposed techniques
- Lower bounds (Theorems 4.2-4.4) extend single-machine results vs. introducing new proof techniques
- Insufficient justification for global smoothness focus (no real-data examples)
- Parameter choices $p=\frac{\bar{L}}{8 L} \max  \left( \frac{\mu}{L}, \frac{1}{\sqrt{m n}} \right)$ lack intuition
- Table 1 formatting obscures complexity comparisons

---

> ### Author Rebuttal · Authors · 2025-07-29
>
> Thank you for your careful review and suggestions.
>
> ## Weaknesses
> > Variance reduction (Section 2.2) and multi-consensus (Algorithm 1) are repurposed techniques
>
> **A1:** We emphasize that our novelty is the stochastic mini-batch sampling in the algorithm design, rather than the variance reduction and multi-consensus. Please see A6 for our detailed response to the stochastic mini-batch sampling.
>
> > Lower bounds (Theorems 4.2-4.4) extend single-machine results vs. introducing new proof techniques.
>
> **A2:** Recall that existing the lower bound on single-machine only considers mean-squared smoothness parameter $\bar L$ [45], while our distributed setting additionally consider the global smoothness parameter $L$. Therefore, our construction requires a carefully scaling on the zero-chain function (48) to guarantee our instance satisfies the smoothness conditions with respect to both $L$ and $\bar{L}$ (e.g., lines 788, 790, and 814), which is more challenging than the single-machine case.
>
> > Insufficient justification for global smoothness focus (no real-data examples).
>
> **A3:** See A9.
>
> > Parameter choices $p=\frac{\bar{L}}{8L}\max\\{ \frac{\mu}{\bar{L}},\frac{1}{\sqrt{mn}}\\}$ lack intuition.
>
> **A4:** See A8.
>
> > Table 1 formatting obscures complexity comparisons.
>
> **A5:** See A10.
>
> ## Questions
> > Insufficient Clarification of Algorithmic Novelty.
>
> **A6:** Thanks for your suggestion.
>
> (a.1) Since $\xi_{i,j}^k$ has Bernoulli distribution, we only need to compute $g_{i,j}$ when $\xi_{i,j}^k=1$ (line 6 of Algorithm 2).
> Therefore, the batch size on the $i$-th node is $\sum_{j=1}^m\xi_{i,j}^k$, which is not fixed for different $i$ and overcomes the limitation of identical mini-batch sizes in previous works.
>
> (a.2) We explain the intuition for partial participation in the case of $n=1$. In this case, our method requires only a subset of nodes to compute their local gradients, since line 6 of Algorithm 2 means node $i$ does not compute its local gradient when $\xi_{i,1}^k=0$.
> In contrast, all existing methods [37, 44, 53] require all  the nodes to compute their local gradients because the mini-batch sizes on all the nodes are identical and must be 1 (because $n=1$).
> For the general $n$, node $i$ still does not compute its local (stochastic) gradient when $\xi_{i,j}^k=0$ occurs for all $j=1,\dots,n$, enabling partial participation across nodes.
>
> (b) The design of stochastic mini-batch sizes allows our algorithm to behave more like solving the finite-sum problem with $mn$ component functions on a single machine.
> As a result, our complexity bounds depend on the global smoothness parameters $L$ and $\bar{L}$.
> In contrast, the fixed mini-batch size used in exiting methods yield a complexity bound that depends on the worst local function,  and thus the corresponding results depend on the local smoothness parameters $L_l$ and $\bar{L}_l$.
>
> > Inadequate Experimental Validation.
>
> **A7:** Thanks for you suggestion.
>
> (a) We conduct adversarial training on the MNIST dataset using the following robust logistic regression model:
> \begin{align*}
> \min_{x\in\mathbb{R}^d}\max_{\substack{\\|y\\|\_{\infty}\le M}}\frac{1}{N}\sum_{i=1}^N\log\left(1+\exp(-b\_i(x + y)^\top a\_i)\right)+\frac{r\_1}{2}\\|x\\|^2 - \frac{r\_2}{2}\\|y\\|^2,
> \end{align*}
> where $x$ is the model parameter, $y$ is an adversarial perturbation constrained by $\\|y\\|\_{\infty}\le M$, and $a_i\in\mathbb{R}^d$ and $b_i\in\\{\pm 1\\}$ denote the feature and label of the $i$‑th sample.
> The experimental results are shown in the following three tables, where "Std" denotes the standard test accuracy and "Adv" denotes the adversarial accuracy under $L_\infty$ perturbations of radius 0.2.
>
> |LIFO calls|3e5||5e5| |1e6||
> |-|-|-|-|-|-|-|
> |Accuracy (%)|Std|Adv|Std|Adv|Std|Adv|
> |GT-EG|46.85|35.85|47.53|45.35|47.54|46.85|
> |MC-SVRE|96.26|19.16|98.50|53.36|98.61|78.01|
> |OADSVI|94.72|21.02|98.50|50.96|98.61|75.08|
> |DIVERSE|98.45|52.08|98.61|77.59|98.56|78.12|
>
> |Computation rounds|2e4||3e4| |5e4||
> |-|-|-|-|-|-|-|
> |Accuracy (%)|Std|Adv|Std|Adv|Std|Adv|
> |GT-EG|47.51|42.53|47.54|46.15|47.54|46.85|
> |MC-SVRE|98.61|30.78|98.61|67.34|98.61|78.01|
> |OADSVI|98.53|40.71|98.58|72.09|98.56|78.06|
> |DIVERSE|98.55|74.81|98.56|78.12|98.56|78.12|
>
> |Communication rounds|2e4||3e4| |5e4||
> |-|-|-|-|-|-|-|
> |Accuracy (%)|Std|Adv|Std|Adv|Std|Adv|
> |GT-EG|47.53|44.71|47.54|46.69|47.54|46.80|
> |MC-SVRE|47.86|44.61|55.76|38.53|87.24|24.55|
> |OADSVI|94.39|20.92|98.56|64.78|98.61|77.21|
> |DIVERSE|98.45|45.89|98.56|69.80|98.56|78.12|
>
> DIVERSE consistently achieves higher adversarial accuracy than all baselines while maintaining comparable standard accuracy, highlighting its superiority on this adversarial logistic regression task.
>
> (b) For the scenario where $\bar{L}\gg L$,  see the experiments in A9 (a).
>
> > Ambiguous Parameter Selection Rationale
>
> **A8:** (a) In the single machine finite-sum problem, if the probability for computing the full gradient is $\Theta(1/\sqrt{mn})$ and the batch size is $\Theta(\sqrt{mn})$, then the LIFO calls optimality can be achieved [A]. Therefore, intuitively, if $p=\Theta(1/\sqrt{mn})$ and $b=\Theta(\sqrt{mn})$, the LIFO calls should also be optimal for the distributed setting. Furthermore, if we enlarge $p$ and $b$ properly, each iteration will include more LIFO calls, thereby reducing the total number of iterations (also the communication and computation rounds) while the total LIFO calls remain unchanged.
>
> Technically  speaking, we first establish the lower bounds (Theorems 4.2-4.4) and then attempt to show that the proposed algorithm can achieve this bound. To this end, we work out these specific parameter settings and rigorously prove that the resulting upper bound nearly matches the lower bound.
>
> (b) When $\bar{L}>\sqrt{mn}L$, as discussed in lines 199-201 and Lemma 3.6, the algorithm computes the full gradient at each iteration, corresponding to $b=mn$ and $p=0$. When $\bar{L}\le\sqrt{mn}L$, the parameter tuning follows the setting in Lemma 3.1, as explained in (a).
>
> > Experiments Overlook Theoretical Highlights
>
> **A9:** (a) We adjust the ratio $\bar{L}/L$ on the a9a dataset following the method in Examples A.1 and A.3, where larger ratios correspond to increased heterogeneity among local functions. This adjustment effectively enlarges $\bar{L}$ while keeping $L$ fixed, thereby allowing us to examine the algorithm’s dependence on the global smoothness parameter. The numbers of LIFO calls and computation rounds required to achieve an accuracy of 1e-3 (i.e., $\\|\mathbf{z}-\mathbf{z}^\star\\|^2\le 10^{-3}$) are reported below.
>
> |$\bar{L}/L$|2.19|10.54|21.16|95.06|189.04|
> |-|-|-|-|-|-|
> |LIFO calls|5.61e5|2.25e6|5.47e6|2.78e7|3.60e7|
> |Computation rounds|2.23e4|6.24e4|1.69e5|5.87e5|7.21e5|
>
> The results show that for $\bar{L}\le\sqrt{mn}L$ (i.e., $\bar{L}/L=2.19,10.54,21.16,95.06$), the numbers of LIFO calls and computation rounds increase nearly linearly with $\bar{L}$, demonstrating our algorithm’s dependence on the global smoothness parameter $\bar{L}$. When $\bar{L}\ge\sqrt{mn},L$ (i.e., $\bar{L}/L=189.04$), the parameter setting of DIVERSE reduces to the case described in Lemma 3.6. In this regime, the algorithm no longer scales with $\mathcal{O}(\sqrt{mn}\bar{L})$ but instead achieves the improved dependence of $\mathcal{O}(mnL)$.
>
> (b) We additionally conduct experiments on randomly generated communication networks with $\chi$ values of 4.72 and 48.11. The following table reports the number of communication rounds required to reach an accuracy of 1e-5 under different network topologies and datasets.
>
> |$\chi$|a9a|w8a|ijcnn1|cod-rna|
> |-|-|-|-|-|
> |4.72|6.83e3|5.57e3|3.41e3|7.49e3|
> |48.11|2.42e4|1.48e4|1.21e4|1.95e4|
> |190.23|4.24e4|2.56e4|2.58e4|3.92e4|
>
> These results are consistent with the theoretical $\sqrt{\chi}$‑dependence of the communication complexity.
>
> (c) We have provided the code in the supplementary materials. Due to limitation of file size, we only include dataset a9a. All of datasets are public and can be accessed on the LibSVM website.
>
> > Writing and Presentation Issues
>
> **A10:** Thank you for the suggestions.
>
> (a) We will explitly define $\Omega(\cdot)$ in revision. For two non-negative functions $f(x)$ and $g(x)$, we write $f(x) =\Omega(g(x))$ if there exist constants $c>0$ and $x_0$ such that $f(x)\ge c , g(x)$ for all $x\ge x_0$.
>
> (b) We will improve the presentation of Algorithms 1–2 and Table 1 in revision.
>
> (c) We will emphasize the practical significance of global-smoothness in revision.
>
> ## Limitations
>
> **A11:** Thanks for your suggestion.
> Since the general nonconvex-nonconcave problem is intractable [B], we focus on the weakly-convex-weakly-concave (WC-WC) setting.
> We can extend proximal point iteration (PPI) [C] to establish the following update
> $$
> \begin{cases}
> \\{(\hat x_i^k,\hat y_i^k)\\}\_{i=1}^m={\rm FastMix}(\\{x_i^k\\},\\{y_i^k\\},\hat{R}) \\\\
> \\{(x_i^{k+1},y_i^{k+1})\\}\_{i=1}^m={\rm DIVERSE}(\\{f_{i,j}(x,y)+\frac{\zeta}{2}\\|x-\\hat x_i^k\\|^2-\frac{\zeta}{2}\\|y-\hat y_i^k\\|^2\\}\_{i,j=1}^{m,n},\\{(\hat x_i^k,\hat y_i^k)\\}\_{i=1}^m,\alpha,\beta,p,b,\eta,R,K),
> \end{cases}
> $$
> where first line encourages each $(\hat x_i^k,\hat y^k)$ be close to their average
> and the second line extends PPI [C] to decentralized setting by using DIVERSE to solve the following sub-problem
> $$
> \min_{x\in{\mathbb R}^{d_x}}\max_{y\in{\mathbb R}^{d_y}}\frac{1}{mn}\sum_{i=1}^m\sum_{j=1}^n f_{i,j}(x,y)+\frac{\zeta}{2}||x-\hat x_i^k||^2-\frac{\zeta}{2}||y-\hat y_i^k||^2,
> $$
> which is SC-SC by taking appropriate $\zeta$.
> We think by appropriate parameter settings and careful analysis, the above scheme can solve WC-WC problems.
>
> ---
> References
>
> [A] Li et al. PAGE: A simple and optimal probabilistic gradient estimator for nonconvex optimization. ICML 2021.
>
> [B] Daskalakis et al. The complexity of constrained min-max optimization. STOC 2021.
>
> [C] Grimmer et al. The landscape of the proximal point method for nonconvex–nonconcave minimax optimization. Mathematical Programming, 2023.

---

> > ### Author Response · Authors · 2025-08-04
> >
> > Dear Reviewer 96So,
> >
> > In our rebuttal, we have highlighted the novelty of our paper, the intuition behind the parameter settings for the proposed method, as well as the additional experimental results following your suggestions.
> >
> > We would like to know if our response has addressed your questions. We welcome any further discussions if there are additional questions.
> >
> > Best Regards,
> >
> > Authors

---

> > > ### Comment · Area_Chair_nHqF · 2025-08-08
> > >
> > > Dear **Reviewer 96So**,
> > >
> > > I kindly request that you submit a response to the authors regarding their rebuttal. As per the conference policies, if no response is provided, I will need to retain the *Insufficient Review* flag, which may result in the **desk rejection of the papers you have submitted**.
> > >
> > > Thank you for your understanding and cooperation.
> > >
> > > Best regards,
> > >
> > > AC

---

> > ### Comment · Reviewer_96So · 2025-08-09
> >
> > Thank you very much for your detailed feedback. You have answered all my questions and opinions. I will improve my score.

---

> > > ### Author Response · Authors · 2025-08-09
> > >
> > > Dear Reviewer 96So,
> > >
> > > Thanks a lot for your positive response and improving the score.
> > >
> > > Best regards,
> > >
> > > Authors

---

### Official Review · Reviewer_TwzD · 2025-07-02

**Clarity:** 3
**Significance:** 3
**Originality:** 2
**Rating:** 4
**Confidence:** 3

**Summary:**

This paper designs a variance-reduced optimistic gradient method with stochastic mini-batch sizes for distributed convex-concave finite-sum minimax problem. And the method could achieve a linear convergence rate depends on global smoothness(not local smoothness) in strongly-convex-strongly-concave situation. In addition, experiments show that performance of proposed method is better than existing methods.

**Questions:**

1 Could you share additional experimental results, such as accuracy with respect to LIFO, communication rounds, and computation rounds for the four datasets already included? If more datasets have been evaluated, including those results would further strengthen the empirical section and support the generality of the proposed method.

2 In the convergence rate analysis, what specific challenges arise from the change in assumptions, and how are they addressed in your analysis? While I believe the modification of the assumptions is meaningful, it is unclear to me whether it introduces significant technical complications or fundamentally alters the existing analytical framework.

3 I have some concerns regarding the novelty of the algorithm. The integration of variance-reduction techniques alone may not substantially deviate from existing methods in terms of analysis. Could you elaborate on the statement at the end of Section 3.1, where you mention that the change to the mini-batch size for the local gradient estimator is one of the key differences from existing work? Specifically, how is this change handled in the convergence rate analysis, and does it pose any unique analytical challenges?

**Ethical Concerns:**

["NO or VERY MINOR ethics concerns only"]

**Final Justification:**

Considering the additional experiments provided by the authors and the theoretical proof of the proposed method in the paper, I believe the methodology is solid and its feasibility has been demonstrated experimentally. However, as I am not deeply familiar with this specific subfield, I am unable to strongly advocate for the acceptance of this paper.

**Limitations:**

Yes

**Quality:**

3

**Strengths And Weaknesses:**

Strengths

1 The paper makes a reasonable adjustment to the assumptions, allowing the convergence rate to be analyzed in terms of both the upper and lower bounds with respect to the global smoothness parameter and the mean-squared smoothness parameter. This differs significantly from existing work that relies on local smoothness assumptions, which may be less realistic in heterogeneous data scenarios.

2 The structure of the paper is clear, and both the algorithmic design and the theoretical contributions are well explained.

3 The experiments include a wide range of baselines under comparable settings, and the reported results are convincing.

Weaknesses

1 The experimental section is relatively limited in terms of dataset diversity, which may restrict the generalizability of the results. Additionally, the absence of commonly reported evaluation metrics such as accuracy makes it difficult to comprehensively assess the practical effectiveness of the proposed method. As a result, the experimental validation appears somewhat narrow and could benefit from more thorough benchmarking.

2 The main algorithmic contributions seem to lie in the incorporation of variance-reduction techniques and a modification to the minibatch size used for local gradient estimators, which is fixed in prior methods. While these changes are reasonable, they appear to be incremental improvements rather than fundamental innovations. The overall level of novelty is therefore somewhat limited.

---

> ### Author Rebuttal · Authors · 2025-07-29
>
> Thank you for the positive evaluation and suggestions.
>
> > The experimental section is relatively limited in terms of dataset diversity, which may restrict the generalizability of the results. Additionally, the absence of commonly reported evaluation metrics such as accuracy makes it difficult to comprehensively assess the practical effectiveness of the proposed method. As a result, the experimental validation appears somewhat narrow and could benefit from more thorough benchmarking.
>
> > Could you share additional experimental results, such as accuracy with respect to LIFO, communication rounds, and computation rounds for the four datasets already included? If more datasets have been evaluated, including those results would further strengthen the empirical section and support the generality of the proposed method.
>
> **A1:** Thank you for your suggestion. The following three tables report the classification accuracy at 1e6 LIFO calls, 4e4 computation rounds, and 5e4 communication rounds, respectively.
>
> - Classification accuracy at 1e6 LIFO calls
> |Accuracy (%) |a9a|w8a|ijcnn1|cod-rna|
> |-|-|-|-|-|
> |GT-EG|71.66|38.83|42.77|63.65|
> |MC-SVRE|77.43|69.61|79.14|72.13|
> |OADSVI|77.40|69.78|78.25|75.20|
> |DIVERSE|82.53|84.94|80.26|80.41|
>
> - Classification accuracy at 4e4 computation rounds
> |Accuracy (%) |a9a|w8a|ijcnn1|cod-rna|
> |-|-|-|-|-|
> |GT-EG|71.74|39.01|42.77|63.65|
> |MC-SVRE|81.62|75.36|78.89|72.69|
> |OADSVI|81.89|75.52|78.19|78.86|
> |DIVERSE|82.67|88.49|81.16|81.70|
>
> - Classification accuracy at 5e4 communication rounds
> |Accuracy (%) |a9a|w8a|ijcnn1|cod-rna|
> |-|-|-|-|-|
> |GT-EG|71.80|39.33|42.81|63.68|
> |MC-SVRE|76.41|62.73|75.85|72.53|
> |OADSVI|82.64|87.70|90.26|82.84|
> |DIVERSE|82.67|88.51|90.26|82.85|
>
> The results indicate that DIVERSE consistently yields the best performance in terms of classification accuracy.
>
> In addition, we conducted adversarial training on the MNIST dataset using the following robust logistic regression model:
> \begin{align*}
> \min_{x \in \mathbb{R}^d} \max_{\substack{\\|y\\|\_{\infty} \le M}} \frac{1}{N} \sum_{i=1}^N \log \left(1 + \exp (-b\_i (x + y)^\top a\_i)\right) + \frac{r\_1}{2}\\|x\\|^2 - \frac{r\_2}{2}\\|y\\|^2,
> \end{align*}
> where $x$ is the model parameter, $y$ is an adversarial perturbation constrained by $\\|y\\|\_{\infty}\le M$, and $a_i \in \mathbb{R}^d$ and $b_i \in \\{\pm 1\\}$ denote the feature and label of the $i$‑th sample.
> The experimental results are shown in the following three tables, where where "Std" denotes the standard test accuracy and "Adv" denotes the adversarial accuracy under $L_\infty$ perturbations of radius 0.2.
>
> |LIFO calls|3e5| |5e5| |1e6| |
> |-|-|-|-|-|-|-|
> |Accuracy (%)|Std|Adv|Std|Adv|Std|Adv|
> |GT-EG|46.85|35.85|47.53|45.35|47.54|46.85|
> |MC-SVRE|96.26|19.16|98.50|53.36|98.61|78.01|
> |OADSVI|94.72|21.02|98.50|50.96|98.61|75.08|
> |DIVERSE|98.45|52.08|98.61|77.59|98.56|78.12|
>
> |Computation rounds|2e4| |3e4| |5e4| |
> |-|-|-|-|-|-|-|
> |Accuracy (%)|Std|Adv|Std|Adv|Std|Adv|
> |GT-EG|47.51|42.53|47.54|46.15|47.54|46.85|
> |MC-SVRE|98.61|30.78|98.61|67.34|98.61|78.01|
> |OADSVI|98.53|40.71|98.58|72.09|98.56|78.06|
> |DIVERSE|98.55|74.81|98.56|78.12|98.56|78.12|
>
> |Communication rounds|2e4| |3e4| |5e4| |
> |-|-|-|-|-|-|-|
> |Accuracy (%)|Std|Adv|Std|Adv|Std|Adv|
> |GT-EG|47.53|44.71|47.54|46.69|47.54|46.80|
> |MC-SVRE|47.86|44.61|55.76|38.53|87.24|24.55|
> |OADSVI|94.39|20.92|98.56|64.78|98.61|77.21|
> |DIVERSE|98.45|45.89|98.56|69.80|98.56|78.12|
>
> These additional results on adversarial logistic regression with the MNIST dataset further support the generality of DIVERSE.
>
> > In the convergence rate analysis, what specific challenges arise from the change in assumptions, and how are they addressed in your analysis? While I believe the modification of the assumptions is meaningful, it is unclear to me whether it introduces significant technical complications or fundamentally alters the existing analytical framework.
>
> **A2:** Compared to previous works [37, 44, 53], the main assumption change is that only the global smoothness rather than local smoothness is required. To obtain the  tighter complexity bounds that depend on the global parameters $L$ and $\bar{L}$, we have introduced the random variables $\\{\xi\_{i,j}^k\\}$ to construct the gradient estimators $\\{\delta\_i^k\\}\_{i=1}^m$ with stochastic mini-batch sizes.
> This scheme makes the algorithm behave more like in the finite-sum optimization on single-machine case (see the example in A3 to Reviewer H3m2 for intuition). The analysis becomes more difficult after the introduction of the random variables $\\{\xi_{i,j}^k\\}$ since  the related local gradient estimators may have different batch-sizes.
> Specifically, all of our analysis has to be conducted base on the global smoothness parameters $L$ and $\bar{L}$ (see pages 19, 20, 24, and 25) rather than the larger local parameters in related works.
>
> Moreover, different with previous works [37, 44, 53] which only consider computation rounds and communication rounds, we also consider the LIFO calls and prove our algorithm is (nearly) optimal with respect to all the three criteria. We emphasize that the LIFO calls are not always proportional to the computation rounds in distributed setting due to the partial participated computation.
>
> > The main algorithmic contributions seem to lie in the incorporation of variance-reduction techniques and a modification to the minibatch size used for local gradient estimators, which is fixed in prior methods. While these changes are reasonable, they appear to be incremental improvements rather than fundamental innovations. The overall level of novelty is therefore somewhat limited.
>
> > I have some concerns regarding the novelty of the algorithm. The integration of variance-reduction techniques alone may not substantially deviate from existing methods in terms of analysis. Could you elaborate on the statement at the end of Section 3.1, where you mention that the change to the mini-batch size for the local gradient estimator is one of the key differences from existing work? Specifically, how is this change handled in the convergence rate analysis, and does it pose any unique analytical challenges?
>
> **A3:** We emphasize that the main difference between our algorithm and existing methods lies in the introduction of the random variables $\\{\xi_{i,j}^k\\}$ (yielding stochastic mini-batch sizes), rather than the variance-reduction technique.
>
> - The stochastic mini-batch sizes enable the algorithm to behave more like the finite-sum optimization on single-machine, leading to the better dependency on smoothness parameters and $m$. Please see A2 for details.
>
> - Since the the gradient estimators $\\{\delta_i^k\\}_{i=1}^m$ may have different mini-batch sizes, the existing analysis for the fixed batch-size does not work. Therefore, our analysis for the gradient estimators is quite different from the counterpart in existing work. Specifically, the analysis on the page 19, 20, 24, and 25 is based on the global smoothness parameter $L$ and $\bar L$, while the analysis in exiting work [37, 44, 53] are based on the larger local parameters.
>
> ---
> References
>
> [37] Dmitry Kovalev, Aleksandr Beznosikov, Abdurakhmon Sadiev, Michael Persiianov, Peter Richtárik, and Alexander Gasnikov. Optimal algorithms for decentralized stochastic variational inequalities. In Advances in Neural Information Processing Systems, pages 31073–31088, 2022.
>
> [44] Luo Luo and Haishan Ye. Decentralized stochastic variance reduced extragradient method. arXiv preprint arXiv:2202.00509, 2022.
>
> [53] Soham Mukherjee and Mrityunjoy Chakraborty. A decentralized algorithm for large scale min-max problems. In 2020 59th IEEE Conference on Decision and Control (CDC), pages 2967–2972. IEEE, 2020.

---

> > ### Comment · Reviewer_TwzD · 2025-08-07
> >
> > Thank you for the additional experimental results and the clarification regarding the novelty of the proof. The supplementary experiments have addressed my concerns about the applicability of the method. I will maintain my supportive stance toward the paper and keep my score unchanged for now.

---

### Official Review · Reviewer_H3m2 · 2025-07-03

**Clarity:** 4
**Significance:** 2
**Originality:** 2
**Rating:** 4
**Confidence:** 4

**Summary:**

This paper studies the decentralized finite-sum minimax optimization problem under the strongly-convex-strongly-concave setting. The authors propose a DIVERSE algorithm, which achieves near-optimal complexities for local oracle calls, computation rounds, and communication rounds. The primary contribution is that the derived complexity bounds depend on the *global* smoothness parameters of the objective function, rather than the *local* (and often much larger) smoothness parameters that prior works rely on. The authors provide a rigorous theoretical analysis, establishing both upper bounds for their algorithm and nearly matching lower bounds for a broad class of decentralized first-order methods. The theoretical findings are validated by numerical experiments on a robust linear regression problem, where DIVERSE outperforms existing state-of-the-art algorithms.

**Questions:**

1. What do we use a stochastic gradient batch-size? The authors claim, “It is worth noting that existing decentralized minimax optimization methods require identical mini-batch size for all the nodes when constructing the local gradient estimator, which is sample inefficient.” Authors need to justify it.

2. What are the key techniques to enable a sharper bound?

**Ethical Concerns:**

["NO or VERY MINOR ethics concerns only"]

**Final Justification:**

I have carefully read the author's responses to my comments and the feedback from the other reviewers. The author has addressed all of my questions and concerns effectively. I am now confident in supporting the publication of this manuscript and will maintain my initial high score.

**Limitations:**

yes

**Quality:**

3

**Strengths And Weaknesses:**

**Strengths:**

- The paper makes a contribution by providing a sharper bound in decentralized minimax optimization.


**Weaknesses:**

- This paper does not provide insight into the algorithm design, rendering it to seem like a simple combination of several existing methods.

- Even though the theoretical analysis is correct and solid, it is better to highlight the key steps in the proof to provide more information to the audience. For example, the idea of establishing the lower bound.

- Some technical claims have not been fully justified. See **Question.**

- The experiments do not report error bars or any other measure of statistical significance.

---

> ### Author Rebuttal · Authors · 2025-07-29
>
> Thank you for the positive evaluation and comments.
>
> > This paper does not provide insight into the algorithm design, rendering it to seem like a simple combination of several existing methods.
>
> **A1:** We have introduced the insight of our algorithm in Section 3.1. As mentioned in lines 159-165, the main difference between our DIVERSE and exiting methods [37, 44, 53] is that the mini-batch sizes of our  local gradient estimators $\\{\delta_i^k\\}\_{i=1}^m$ in equation (3) are not identical since the variables $\\{\xi_{i,j}^k\\}_{i,j=1}^{m,n}$ are random. Therefore, the behaviors of all $m$ nodes are similar to that of the stochastic gradient estimator with large mini-batch $b$ on a single machine (see examples in A3 for intuition), which is the core idea to achieve the sharper complexity bounds that depend on the global smoothness. The above ideas and results have not appeared in existing works [37, 44, 53], since they only consider the fixed mini-batch size for all nodes.
>
> > Even though the theoretical analysis is correct and solid, it is better to highlight the key steps in the proof to provide more information to the audience. For example, the idea of establishing the lower bound.
>
> **A2:** Thank you for your suggestion. We sketch and highlight the key steps for the proof of our lower bound on LIFO calls (Theorem 4.2) as follows.
>
> - We first introduce Lemma D.1 to construct the function defined equation (48) and present its properties of smoothness, zero-chain, and saddle point.
>
> - We then introduce Lemma D.2 to achieve the second term $\Omega( \min \\{ mn L, \sqrt{mn}\bar{L}\\}/\mu \log ( 1/\epsilon))$ by scaling and spiting the function defined equation (48), which leads to the construction of local functions $f_{i,j}$ in line 790.
>
> - Consequently, Lemma D.3 is introduced to achieve the first term $\Omega(mn)$ in our lower bound by constructing the decoupled function defined in line 814.
>
> - Combining Lemmas D.2 and D.3 yields the final lower bound in Theorem 4.2.
>
> We are happy to incorporate a sketched proof for Theorem 4.2 and other main results in our revision.
>
> > Some technical claims have not been fully justified. See Question.
>
> > What do we use a stochastic gradient batch-size? The authors claim, “It is worth noting that existing decentralized minimax optimization methods require identical mini-batch size for all the nodes when constructing the local gradient estimator, which is sample inefficient.” Authors need to justify it.
>
> **A3:** We first provide the intuition in the case of $n=1$. In this case, our method requires only a subset of nodes to compute their local gradients, since line 6 of Algorithm 2 involves the random variable $\xi_{i,j}^k$. In contrast, all existing methods [37, 44, 53]  require all the nodes to compute their local gradients because the mini-batch sizes on all the nodes are identical and must be 1 (because $n=1$). This means existing methods have to compute the full gradient of the objective and they are sample inefficient.
>
> For the general $n$, the stochastic batch-size allows our method to behave similar to solving the finite-sum problem with $mn$ components on a single machine, leading to the dependency on the global mean-squared smoothness parameter $\bar{L}$ in the complexity. In contrast, the fixed batch-size in existing methods leads to the dependency on the local smoothness parameter $\bar{L}_l$. Recall that $\bar L$ is typically smaller than $\bar{L}_l$, which means our method is more efficient.
>
> If we ignore the difference among smoothness parameters (i.e., $\bar L={\bar L}_\ell=L$), our method achieves the LIFO complexity of $\mathcal{O}(mn + \sqrt{mn}L/\mu)$ while the state-of-the-art OADSVI [37] requires $\mathcal{O}(mn + m\sqrt{n}L/\mu)$. The tighter dependence on $m$ is also attributed to the novel stochastic mini-batch sizes we have used.
>
> > What are the key techniques to enable a sharper bound?
>
> **A4:** The key technique is introducing the stochastic mini-batch sizes to achive the better dependence on the smoothness parameter. Please see A1 and A3 for the detailed explanation.
>
> > The experiments do not report error bars or any other measure of statistical significance.
>
> **A5:** Thanks for your suggestion. We have run our experiments by 10 times and present the the mean and standard deviation as follows. The three tables report the LIFO calls, computation rounds, and communication rounds required to achieve an accuracy of 1e-5 (i.e., $\\| \mathbf{z} - \mathbf{z}^\star \\|^2 \leq 10^{-5}$).
>
> - LIFO calls:
> | |a9a|w8a|ijcnn1|cod-rna|
> |-|-|-|-|-|
> |MC-SVRE|2.1e6 ±7.1e4|1.5e6 ±9.0e4|1.5e6 ±6.9e4|7.3e6 ±3.8e5|
> |OADSVI|2.5e6 ±1.3e5|2.9e6 ±2.5e5|1.8e6 ±2.1e5|5.6e6 ±6.8e5|
> |DIVERSE|8.1e5 ±6.7e4|6.1e5 ±7.2e4|4.6e5 ±1.3e5|2.1e6 ±1.8e5|
>
>
> - Computation rounds:
> | |a9a|w8a|ijcnn1|cod-rna|
> |-|-|-|-|-|
> |MC-SVRE|4.2e4 ±1.4e3|3.1e4 ±1.8e3|3.0e4 ±1.4e3|1.5e5 ±7.7e3|
> |OADSVI|4.9e4 ±2.7e3|5.8e4 ±5.0e3|3.6e4 ±4.2e3|1.1e5 ±1.4e4|
> |DIVERSE|2.9e4 ±3.7e3|2.0e4 ±4.0e3|1.7e4 ±6.0e3|5.4e4 ±4.2e3|
>
>
> - Communication rounds:
> | |a9a|w8a|ijcnn1|cod-rna|
> |-|-|-|-|-|
> |MC-SVRE|2.2e5 ±2.3e4|1.1e5 ±1.1e4|1.2e5 ±1.9e4|1.9e5 ±2.9e4|
> |OADSVI|4.6e4 ±1.5e3|4.7e4 ±3.1e3|3.1e4 ±2.7e3|4.7e4 ±6.1e3|
> |DIVERSE|4.2e4 ±8.3e3|2.5e4 ±8.7e3|2.5e4 ±9.1e3|3.9e4 ±7.1e3|
>
> Experimental results show that DIVERSE has lower variance in LIFO calls and communication rounds than existing state-of-the-art algorithms, indicating its stability.
>
> ---
> References
>
> [37] Dmitry Kovalev, Aleksandr Beznosikov, Abdurakhmon Sadiev, Michael Persiianov, Peter Richtárik, and Alexander Gasnikov. Optimal algorithms for decentralized stochastic variational inequalities. In Advances in Neural Information Processing Systems, pages 31073–31088, 2022.
>
> [44] Luo Luo and Haishan Ye. Decentralized stochastic variance reduced extragradient method. arXiv preprint arXiv:2202.00509, 2022.
>
> [53] Soham Mukherjee and Mrityunjoy Chakraborty. A decentralized algorithm for large scale min-max problems. In 2020 59th IEEE Conference on Decision and Control (CDC), pages 2967–2972. IEEE, 2020.

---

> > ### Comment · Reviewer_H3m2 · 2025-08-03
> >
> > I would like to thank the authors for their detailed reply to my comments, and the concerns have been addressed. It is suggested that the authors discuss them in the revision. Since I have supported the paper in the initial stage, I will keep the score.

---

### Official Review · Reviewer_nrCq · 2025-07-09

**Clarity:** 3
**Significance:** 3
**Originality:** 3
**Rating:** 5
**Confidence:** 4

**Summary:**

In this paper, the authors consider decentralized finite-sum strongly convex–strongly concave saddle-point problems. To solve this problem, they propose a new variance-reduced algorithm called __DIVERSE__. The algorithm is a non-trivial combination of OGDA and Snapshot Gradient Tracking. They provide convergence guarantees along with new theoretical lower bounds, which match their upper bounds. To support their theoretical findings, they present experimental results demonstrating that the proposed method outperforms previous approaches.

**Questions:**

__Questions:__

1.On page 28, right after line 669, there is an equation:
$$\sum_{i=0}^m ||g_i(z^{0})||^2 = \sum_{i=0}^m ||g_i(z^{0}) - g_i(z^{\star})||^2.$$
However, this holds only if each $g_i(z^{\star}) = 0$. I believe this is a mistake in your proof. Could you comment on this observation?

__Recommendations:__

1.It would be better to cite the following two papers on the snapshot technique for gradient tracking:
https://arxiv.org/abs/2110.05282,
https://arxiv.org/abs/2212.05273.

2.To be more precise, DIVERSE is not based exactly on OGDA but rather on the Operator Extrapolation method (https://arxiv.org/abs/2011.02987), which shares similarities in the convergence analysis. Citing this work could help clarify the construction of the method.

3.Please introduce the definition of $\Delta^k$ in the main part of the paper.


I may consider increasing my score if all of my questions are addressed.

**Ethical Concerns:**

["NO or VERY MINOR ethics concerns only"]

**Final Justification:**

From my perspective, this is a good theoretical paper. The main algorithm is based on idea of the work (https://proceedings.neurips.cc/paper_files/paper/2022/file/c959bb2cb164d37569a17fa67494d69a-Paper-Conference.pdf). I recommend to accept this paper.

**Limitations:**

Yes

**Quality:**

3

**Strengths And Weaknesses:**

__Strengths:__

1. Well-written paper.

2. Strong theoretical findings: novel lower bounds and a new algorithm.

__Weaknesses:__

1. The complexity of DIVERSE has an additional logarithmic factor compared to OADSVI.

2. See the Questions section.

---

> ### Author Rebuttal · Authors · 2025-07-29
>
> Thank you for your careful review and constructive suggestions.
>
> > On page 28, right after line 669, there is an equation:
> $$\sum_{i=1}^m\\| g_i(z^0)\\|^2 = \sum_{i=1}^m\\|g_i(z^0)-g_i(z^\star)\\|^2.$$
> However, this holds only if each $g_i(z^\star) = 0$.  I believe this is a mistake in your proof. Could you comment on this observation?
>
> Thank you for pointing out the issue in the proof.
> It can addressed by modifying the upper bound of $\\| \mathbf{S}^0 - \mathbf{1} \bar{\mathbf{s}}^0 \\|^2$ and it does not affect our main results.
> Specifically, we can upper bound $\\| \mathbf{S}^0 - \mathbf{1} \bar{\mathbf{s}}^0 \\|^2$ as
> \begin{align*}
> \\| \mathbf{S}^0 - \mathbf{1} \bar{\mathbf{s}}^0 \\|^2
> &= \\| \mathtt{FastMix}(\mathbf{\Delta}^0, \mathbf{W}, R) - \frac{1}{m} \mathbf{1} \mathbf{1}^\top \mathtt{FastMix}(\mathbf{\Delta}^0, \mathbf{W}, R) \\|^2 \\\\
> &\le \rho^2 \\| \mathbf{\Delta}^0 - \frac{1}{m} \mathbf{1} \mathbf{1}^\top \mathbf{\Delta}^0 \\|^2 \\\\
> &\le \rho^2 \\| \mathbf{\Delta}^0 \\|^2\\\\
> & = \rho^2 \sum_{i=1}^m \\| \mathbf{g}\_i(\mathbf{z}^0) \\|^2\\\\
> & \le 2 \rho^2 \sum_{i=1}^m \\| \mathbf{g}\_i(\mathbf{z}^0) - \mathbf{g}\_i(\mathbf{z}^\star) \\|^2 + 2 \rho^2 \sum_{i=1}^m \\| \mathbf{g}\_i(\mathbf{z}^\star) \\|^2\\\\
> & \le 2 \rho^2 m^2 \bar{L}^2 \\| \mathbf{z}^0 - \mathbf{z}^\star \\|^2 + 2 \rho^2 \sum_{i=1}^m \\| \mathbf{g}\_i(\mathbf{z}^\star) \\|^2,
> \end{align*}
> where the first inequality holds by Proposition B.2, the second inequality follows from the fact that $\\| \mathbf{A} - \frac{1}{m}\mathbf{1} \mathbf{1}^\top\mathbf{A} \\| \le \\| \mathbf{A} \\|$ for any $\mathbf{A} \in \mathbb{R}^{m\times d_z}$, the third inequality is based on Young's inequality, and the last inequality holds due to Assumption 2.3.
> Then we modify the setting of $R$ (equation (31) on page 27) by replacing the last term in the max  operator (in the logarithmic term) with
> $\frac{8C\_1\eta^2 \left(m^2 \bar{L}^2 \\| \mathbf{z}^0 - \mathbf{z}^\star \\|^2 + \sum_{i=1}^m \\| \mathbf{g}\_i(\mathbf{z}^\star) \\|^2 \right)}{(1-\tilde{\alpha})\tilde{\alpha}\Phi^0},$
> which guarantees
> $\rho^2 \le \frac{(1-\tilde{\alpha})\tilde{\alpha}\Phi^0}{8C\_1\eta^2 \left(m^2 \bar{L}^2 \\| \mathbf{z}^0 - \mathbf{z}^\star \\|^2 + \sum_{i=1}^m \\| \mathbf{g}\_i(\mathbf{z}^\star) \\|^2 \right)}.$
>
> Combining the above modified upper bounds of $\\| \mathbf{S}^0 - \mathbf{1} \bar{\mathbf{s}}^0 \\|^2$ and $\rho^2$, we can achieve
> \begin{align*}
> \\| \mathbf{S}^0 - \mathbf{1} \bar{\mathbf{s}}^0 \\|^2
> \le \frac{1-\tilde{\alpha}}{4\eta^2 C_1} \tilde{\alpha} \Phi^0.
> \end{align*}
>
> Therefore, we have obtain the desired result and the remaining proof after line 669 still holds.
> We emphasize that the modification of $\rho^2$ only affect the logarithmic term in $R$.
> Hence, the overall communication complexity is still in the order of $TR=\tilde{\mathcal{O}}(\sqrt{\chi}L/\mu \log(1/\epsilon))$, which is not changed.
>
> > It would be better to cite the following two papers on the snapshot technique for gradient tracking [A, B].
>
> > To be more precise, DIVERSE is not based exactly on OGDA but rather on the Operator Extrapolation method [C], which shares similarities in the convergence analysis. Citing this work could help clarify the construction of the method.
>
> Thank you for these suggestions. We are happy to include the recommended references in  revision.
> Specifically, we will cite references [A] and [B] in the literature review of decentralized optimization.
> We will also cite reference [C] to clarify the connection between DIVERSE and the Operator Extrapolation framework.
>
> > Please introduce the definition of $\mathbf{\Delta}^k$ in the main part of the paper.
>
> Thanks for your suggestion.
> We will explicitly introduce the definition of $\mathbf{\Delta}^k$ in our revision.
> Specifically, it is the aggregated matrix of local gradient estimators $\\{\mathbf{\delta}^k_i\\}_{i=1}^m$ (defined in equation (3)), i.e.,
> \begin{align*}
> \mathbf{\Delta}^k =
> \begin{bmatrix}
> \mathbf{\delta}^k_1 \\\\
> \vdots \\\\
> \mathbf{\delta}^k_m
> \end{bmatrix}
> \in \mathbb{R}^{m \times d_z}.
> \end{align*}
>
> ---
> References
>
> [A] Zhuoqing Song, Lei Shi, Shi Pu, and Ming Yan. Optimal gradient tracking for decentralized optimization. Mathematical Programming, 207:1–53, 2024.
>
> [B] Haishan Ye and Xiangyu Chang. Snap-shot decentralized stochastic gradient tracking methods. arXiv preprint arXiv:2212.05273, 2022.
>
> [C] Georgios Kotsalis, Guanghui Lan, and Tianjiao Li. Simple and optimal methods for stochastic variational inequalities, I: operator extrapolation. SIAM Journal on Optimization, 32:2041–2073, 2022.

---

> ### Author Response · Authors · 2025-08-05
>
> Dear Reviewer nrCq,
>
> Thanks a lot for your positive feedback and raising the score.
>
> Best regards,
>
> Authors

---

### Comment · Area_Chair_nHqF · 2025-08-05

Dear Reviewers,

Thank you again for your time and efforts in reviewing papers for NeurIPS 2025.

I am writing to remind you that **active participation in the author-reviewer discussion phase is mandatory**. According to the guidelines from the NeurIPS program chairs, reviewers are **required to engage directly with the authors in the discussion thread**, especially in response to their rebuttals.

Please note the following important policy:

- Simply reading the rebuttal or internally considering it is **not sufficient** -- reviewers must **post at least one message to the authors**, even if it is only to confirm that their concerns were resolved. If they have not been addressed, please explain why.

- **Acknowledging the rebuttal without any engagement with the authors will be considered insufficient**. I am obligated to flag such cases using the *InsufficientReview* mechanism, which may **impact future reviewing invitations and result in desk rejection of your own submissions**.

If you have not yet responded to the authors in the discussion thread, I kindly ask you to do so **as soon as possible**, and **no later than August 8, 11:59pm AoE**.

Please don't hesitate to reach out to me if you have any questions or concerns.

Best regards,

AC

---

### Note · Authors · 2025-08-15

Dear area chair and reviewers,

We sincerely appreciate your continued follow-up during the rebuttal period and taking the time to handle our submission.

Based on the feedback of reviewers, their major concerns are well addressed at the rebuttal stage, and each reviewer is positive to our submission or agree to raise the score after the rebuttal. We are happy to incorporate the reviewers' suggestions into our revision accordingly.

We look forward to your decision and greatly appreciate your consideration of our paper.

Best regards,

Authors

---

### Decision · Program_Chairs · 2025-09-17

**Decision:**

Accept (spotlight)

**Comment:**

This paper studies decentralized smooth strongly-convex–strongly-concave finite-sum min-max problems and introduces a new method, DIVERSE, for solving them. The algorithm combines Optimistic Gradient Descent Ascent, stochastic mini-batch sampling, recursive variance reduction, and a fast consensus procedure. While each of these components is known, their careful integration in this work is non-trivial. The authors succeed in designing a method that is both
- provably convergent and
- optimal.
Under the stated assumptions, they establish tight complexity upper bounds for their method and complement them with matching lower bounds for the considered setting.

I view this as a strong theoretical contribution. The reviewers largely share this assessment. In particular, although Reviewer 96So did not update their score, their final justification indicates that they would have raised it, acknowledging that the authors had adequately addressed their concerns.

The paper does have some weaknesses, most notably the limited scope of numerical experiments. However, I believe this is a relatively minor issue compared to the theoretical depth and significance of the results.

Overall, I recommend **acceptance as a Spotlight**. The problem studied is fundamental, and achieving both lower bounds and an optimal method is an important milestone in optimization research.